# On the Properties of Kullback-Leibler Divergence Between Multivariate Gaussian Distributions

**Yufeng Zhang, Jialu Pan** [*]**, Kenli Li**
College of Computer Science and Electronic Engineering
Hunan University, Changsha, China
{yufengzhang, jialupan, lkl}@hnu.edu.cn

**Wanwei Liu, Zhenbang Chen, Xinwang Liu**
College of Computer
Key Laboratory of Software Engineering for Complex Systems
National University of Defense Technology, Changsha, China
{wwliu, zbchen, xinwangliu, wj}@nudt.edu.cn

**Ji Wang**
College of Computer
State Key Laboratory for High Performance Computing
Key laboratory of Software Engineering for Complex Systems
National University of Defense Technology, Changsha, China
wj@nudt.edu.cn

## Abstract

Kullback-Leibler (KL) divergence is one of the most important measures to calculate the difference between probability distributions. In this paper, we theoretically study several properties of KL divergence between multivariate Gaussian distributions. Firstly, for any two $n$-dimensional Gaussian distributions $\mathcal{N}_1$ and $\mathcal{N}_2$, we prove that when $KL(\mathcal{N}_2||\mathcal{N}_1) \leq \varepsilon$ ($\varepsilon > 0$) the supremum of $KL(\mathcal{N}_1||\mathcal{N}_2)$ is $(1/2)\left((-W_0(-e^{-(1+2\varepsilon)}))^{-1} + \log(-W_0(-e^{-(1+2\varepsilon)})) - 1\right)$, where $W_0$ is the principal branch of Lambert $W$ function. For small $\varepsilon$, the supremum is $\varepsilon + 2\varepsilon^{1.5} + O(\varepsilon^2)$. This quantifies the approximate symmetry of small KL divergence between Gaussian distributions. We further derive the infimum of $KL(\mathcal{N}_1||\mathcal{N}_2)$ when $KL(\mathcal{N}_2||\mathcal{N}_1) \geq M$ ($M > 0$). We give the conditions when the supremum and infimum can be attained. Secondly, for any three $n$-dimensional Gaussian distributions $\mathcal{N}_1$, $\mathcal{N}_2$, and $\mathcal{N}_3$, we theoretically show that an upper bound of $KL(\mathcal{N}_1||\mathcal{N}_3)$ is $3\varepsilon_1 + 3\varepsilon_2 + 2\sqrt{\varepsilon_1\varepsilon_2} + o(\varepsilon_1) + o(\varepsilon_2)$ when $KL(\mathcal{N}_1||\mathcal{N}_2) \leq \varepsilon_1$ and $KL(\mathcal{N}_2||\mathcal{N}_3) \leq \varepsilon_2$ ($\varepsilon_1, \varepsilon_2 \geq 0$). This reveals that KL divergence between Gaussian distributions follows a relaxed triangle inequality. Note that, all these bounds in the theorems presented in this work are independent of the dimension $n$. Finally, we discuss several applications of our theories in deep learning, reinforcement learning, and sample complexity research.

## 1 Introduction

A statistical divergence measures the "distance" between probability distributions. Let $X$ be a space of probability distributions with the same support. A statistical divergence $D : X \times X \to \mathbb{R}^+$ ($\mathbb{R}^+$

---

[*]Jialu Pan is the corresponding author.

37th Conference on Neural Information Processing Systems (NeurIPS 2023).

is the set of non-negative real numbers) should satisfy (a) non-negativity: $D(p, q) \geq 0$ and (b) identity of indiscernibles: $D(p, p) = 0$, where $p, q$ are probability densities. Another stricter concept, statistical distance, also measures the distance between probability distributions. A statistical distance should satisfy two additional properties, including (c) symmetry: $D(p, q) = D(q, p)$ and (d) triangle inequality: $D(p, q) \leq D(p, g) + D(g, q)$, where $p, q$ and $g$ are probability densities.

Kullback-Leibler (KL) divergence, also referred to as relative entropy [31], is applied broadly in many fields, such as machine learning [10, 23], information theory [15], and statistics [45]. The KL divergence between two continuous probability densities $p(x)$ and $q(x)$ is defined as $KL(p(x)||q(x)) = \int p(x) \log \frac{p(x)}{q(x)} \, dx$. KL divergence is not a proper distance [31]. First, KL divergence is not symmetric. It might happen that the forward KL divergence[2] $KL(p||q)$ is very small but the reverse KL divergence $KL^*(p||q) = KL(q||p)$ is very large. Second, KL divergence does not satisfy the triangle inequality. This hinders the application of KL divergence in many contexts.

KL divergence also has connections with other information measures. For example, by taking the second-order Taylor expansion, KL divergence can be approximated with fisher information matrix [31]. Therefore, KL divergence is locally approximately symmetric when two distributions are close to each other.

Meanwhile, Gaussian distribution is one of the most important distributions and is central to statistics. It is also pervasive in many fields. The probability density function of an $n$-dimensional Gaussian distribution is $\mathcal{N}(\boldsymbol{\mu}, \boldsymbol{\Sigma}) = ((2\pi)^{n/2}|\boldsymbol{\Sigma}|^{1/2})^{-1} \exp\left(-(1/2)(\boldsymbol{x} - \boldsymbol{\mu})^\top \boldsymbol{\Sigma}^{-1}(\boldsymbol{x} - \boldsymbol{\mu})\right)$. Here $\boldsymbol{\mu} \in \mathbb{R}^n$ is the mean and $\boldsymbol{\Sigma} \in \boldsymbol{S}_{++}^n$ is the covariance matrix, where $\boldsymbol{S}_{++}^n$ is the space of symmetric positive definite $n \times n$ matrices. Gaussian distribution is the basis for more complicated distributions such as Gaussian Mixture Model [10]. In this paper, we refer to Gaussian distribution as Gaussian for brevity.

The KL divergence between two $n$-dimensional Gaussians $\mathcal{N}_1$, $\mathcal{N}_2$ has the following closed form [45]

$$KL(\mathcal{N}_1(\boldsymbol{\mu}_1, \boldsymbol{\Sigma}_1)||\mathcal{N}_2(\boldsymbol{\mu}_2, \boldsymbol{\Sigma}_2)) = \frac{1}{2}\left(\log\frac{|\boldsymbol{\Sigma}_2|}{|\boldsymbol{\Sigma}_1|} + \mathrm{Tr}(\boldsymbol{\Sigma}_2^{-1}\boldsymbol{\Sigma}_1) + (\boldsymbol{\mu}_2 - \boldsymbol{\mu}_1)^\top \boldsymbol{\Sigma}_2^{-1}(\boldsymbol{\mu}_2 - \boldsymbol{\mu}_1) - n\right) \quad (1)$$

where the logarithm is taken to base $e$ and Tr is the trace of matrix. Like many other distributions, KL divergence between Gaussians is neither symmetric nor satisfies the triangle inequality.

In this work, we investigate the following two research problems, which are motivated by our research on out-of-distribution detection with flow-based model [57].

1. How to quantify the relation between forward and reverse KL divergences between Gaussian distributions?
2. Does the KL divergence between Gaussian distributions satisfy some property similar to triangle inequality?

**Contributions.** The contributions of this work are as follows. Let $\mathcal{N}_i(\boldsymbol{\mu}_i, \boldsymbol{\Sigma}_i)$ ($i \in \{1, 2, 3\}$) be any three $n$-dimensional Gaussians.

1. We prove that when $KL(\mathcal{N}_1||\mathcal{N}_2) \leq \varepsilon$ ($\varepsilon \geq 0$) the supremum of $KL(\mathcal{N}_2||\mathcal{N}_1)$ is $(1/2)\left((-W_0(-e^{-(1+2\varepsilon)}))^{-1} + \log(-W_0(-e^{-(1+2\varepsilon)})) - 1\right)$, where $W_0$ is the principal branch of Lambert $W$ function. We give the conditions when the supremum can be attained. For small $\varepsilon$, the supremum is $\varepsilon + 2\varepsilon^{1.5} + O(\varepsilon^2)$. This quantifies the approximate symmetry of small KL divergence between Gaussians.
2. We find the infimum of $KL(\mathcal{N}_1||\mathcal{N}_2)$ if $KL(\mathcal{N}_2||\mathcal{N}_1) \geq M$ ($M > 0$). We also give the conditions when the infimum can be attained.
3. We find an upper bound of $KL(\mathcal{N}_1||\mathcal{N}_3)$ if $KL(\mathcal{N}_1||\mathcal{N}_2) \leq \varepsilon_1$ and $KL(\mathcal{N}_2||\mathcal{N}_3) \leq \varepsilon_2$ for $\varepsilon_1, \varepsilon_2 \geq 0$. For small $\varepsilon_1$ and $\varepsilon_2$, the upper bound is $3\varepsilon_1 + 3\varepsilon_2 + 2\sqrt{\varepsilon_1\varepsilon_2} + o(\varepsilon_1) + o(\varepsilon_2)$ . This indicates that KL divergence between Gaussians follows a relaxed triangle inequality.
4. All the bounds in our theorems are independent of the dimension $n$. This is a critical property in contexts where dimensionality has a fundamental impact.
5. The theorems proved in this paper can extend the applications of KL divergence in many contexts. We discuss the motivation application in out-of-distribution detection with flow-based model and multiple applications in reinforcement learning and sample complexity research.

---

[2]We can choose to call $KL(p||q)$ or $KL(q||p)$ as forward KL divergence.

The rest part of this paper is organized as follows. In Section 2 we discuss related work. In Section 3 we prepare lemmas and notations. In Section 4 we investigate the supremum (infimum) of reverse KL divergence between Gaussians when forward KL divergence is bounded. In Section 5 we investigate the relaxed triangle inequality of KL divergence between Gaussians. In Section 6, we discuss applications. Finally, we conclude in Section 7. We put long proofs in Appendix.

## 2 Related work

KL divergence has a wide range of applications [15, 10, 23, 45, 20, 25, 3]. Researchers have investigated KL divergence between many distributions, including Markov sources [47], GMM models [18, 27], multivariate generalized Gaussians [11], discrete normal distributions [42], *etc*. In [45], a bound of KL divergence between Gaussians is given. As far as we know, no related work focuses on the similar properties of KL divergence between Gaussians as in this paper.

KL divergence is one member of more general divergences such as Bregman divergence [12, 7, 21, 53], $f$-divergence [5, 45, 4], Rényi divergence [48, 45, 54], and $(f, \Gamma)$-divergence [9]. Bregman divergence defines a class of divergences [7] in vector space. KL divergence between multinomial distributions is a special form of Bregman divergence when the convex function for Bregman divergence is chosen as $\sum_{i=1}^{n} p_i \log p_i$, where $p_i \geq 0$ define a multinomial distribution. Frigyik *et. al.* [21] extends vector Bregman divergence to functional Bregman divergence in $L^p$. Similarly, KL divergence is a special form of functional Bregman divergene. (Functional) Bregman divergence also satisfies generalized Pythagoras theorem [7, 21].

The asymmetry of KL divergence has restricted the application of KL divergence in practical applications. Many other divergences have been investigated [46, 16, 2, 43, 26, 17, 7, 22, 56, 55]. Pardo gives a comprehensive survey on a wide range of statistical divergences in his book [45].

## 3 Lemmas and Notations

Table 1: Notations.

| | |
|---|---|
| $f(x)$ | $x - \log x$ $(x \in \mathbb{R}^{++})$ |
| $W(x)$ | the Lambert $W$ function |
| $W_0(x)$ | the principal branch (branch 0) of $W(x)$ |
| $W_{-1}(x)$ | the branch $-1$ of $W(x)$ |
| $w_1(t)$ | the smaller solution of $f(x) = 1 + t$ $(t \geq 0)$ |
| $w_2(t)$ | the larger solution of $f(x) = 1 + t$ $(t \geq 0)$ |
| $\bar{f}(x_1, \ldots, x_n)$ | $\sum_{i=1}^{n} f(x_i)$ |
| $\lambda$ | the eigenvalue of matrix |
| $\mathcal{N}(0, I)$ | standard Gaussian distribution, dimension $n$ is eliminated for brevity |

We introduce the following Lambert $W$ function before our theoretical results.

**Definition 1** *Lambert $W$ Function[32, 14]. The inverse function of function $y = xe^x$ is called Lambert W function $y = W(x)$.*

When $x \in \mathbb{R}$, $W$ is a multivalued function with two branches $W_0, W_{-1}$, where $W_0$ is the principal branch (also called branch 0) and $W_{-1}$ is the branch $-1$. Figure A.1 in Appendix A shows the graph of $W_0$ and $W_{-1}$.

**Lemmas**. *Function $f(x) = x - \log x$ $(x \in \mathbb{R}^{++})$ ($\mathbb{R}^{++}$ is the set of positive real numbers) lies in the core of our problems.* In Lemma B.1 in Appendix B, we prove several properties of $f(x)$. We show that the inverse function of $f$ is $f^{-1}(x) = -W(-e^{-x})$ $(x \geq 1)$. Equation $f(x) = x - \log x = 1 + t$ $(t \geq 0)$ has two solutions $w_1(t) = -W_0(-e^{-(1+t)}) \in (0, 1]$ and $w_2(t) = -W_{-1}(-e^{-(1+t)}) \in [1, +\infty)$. We treat $w_1(t), w_2(t)$ as functions of $t$.

Table 1 summarizes some notations used in this paper. Please see Table A.1 in Appendix A for a full list of notations.

# 4 Bounds of Forward and Reverse KL Divergence Between Gaussians

In this section, we investigate the relation between forward and reverse KL divergences between Gaussians. These conclusions quantify the approximate symmetry of small KL divergence between Gaussians.

## 4.1 Supremum of Reverse KL Divergence Between Gaussians

The following Theorem 1 gives the supremum of reverse KL divergence when forward KL divergence is bounded by a number $\varepsilon$.

**Theorem 1** *For any two $n$-dimensional Gaussian distributions $\mathcal{N}(\boldsymbol{\mu}_1, \boldsymbol{\Sigma}_1)$ and $\mathcal{N}(\boldsymbol{\mu}_2, \boldsymbol{\Sigma}_2)$, if $KL(\mathcal{N}(\boldsymbol{\mu}_1, \boldsymbol{\Sigma}_1)||\mathcal{N}(\boldsymbol{\mu}_2, \boldsymbol{\Sigma}_2)) \leq \varepsilon \ (\varepsilon \geq 0)$, then*

$$KL(\mathcal{N}(\boldsymbol{\mu}_2, \boldsymbol{\Sigma}_2)||\mathcal{N}(\boldsymbol{\mu}_1, \boldsymbol{\Sigma}_1)) \leq \frac{1}{2}\left(\frac{1}{-W_0(-e^{-(1+2\varepsilon)})} - \log\frac{1}{-W_0(-e^{-(1+2\varepsilon)})} - 1\right)$$

*The supremum is attained when the following two conditions hold.*

(1) *There exists only one eigenvalue $\lambda_j$ of $B_2^{-1}\boldsymbol{\Sigma}_1(B_2^{-1})^\top$ or $B_1^{-1}\boldsymbol{\Sigma}_2(B_1^{-1})^\top$ equal to $-W_0(-e^{-(1+2\varepsilon)})$ and all other eigenvalues $\lambda_i \ (i \neq j)$ are equal to 1, where $B_1 = P_1 D_1^{1/2}$, $P_1$ is an orthogonal matrix whose columns are the eigenvectors of $\boldsymbol{\Sigma}_1$, $D_1 = diag(\lambda_1, \ldots, \lambda_n)$ whose diagonal elements are the corresponding eigenvalues, $B_2$ is defined in the similar way as $B_1$ except on $\boldsymbol{\Sigma}_2$.*

(2) $\boldsymbol{\mu}_1 = \boldsymbol{\mu}_2$.

**Overview of Proof of Theorem 1**. Please see Appendix D for the full proof. Here we give the overview of the proof. Theorem 1 can be proved by solving the following optimization problem $P_1$ analytically.

$$P_1 : \text{ maximize } KL(\mathcal{N}(\boldsymbol{\mu}_2, \boldsymbol{\Sigma}_2)||\mathcal{N}(\boldsymbol{\mu}_1, \boldsymbol{\Sigma}_1)) \tag{2}$$

$$s.t. \ KL(\mathcal{N}(\boldsymbol{\mu}_1, \boldsymbol{\Sigma}_1)||\mathcal{N}(\boldsymbol{\mu}_2, \boldsymbol{\Sigma}_2)) \leq \varepsilon \tag{3}$$

The proof consists of the following several steps.

1. *Invertible linear transformation.* We apply a linear transformation to $\mathcal{N}(\boldsymbol{\mu}_1, \boldsymbol{\Sigma}_1)$ and $\mathcal{N}(\boldsymbol{\mu}_2, \boldsymbol{\Sigma}_2)$ such that one of them is transformed to a standard Gaussian. In this way, we can simplify the objective function and the constraint in $P_1$. Note that diffeomorphism preserves KL divergence [41] (see Proposition D.1 in Appendix D). In the end, we use the inverse linear transformation to transform $\mathcal{N}(\boldsymbol{\mu}_1, \boldsymbol{\Sigma}_1)$ and $\mathcal{N}(\boldsymbol{\mu}_2, \boldsymbol{\Sigma}_2)$ back.

2. *Reducing to new optimization problem.* We reduce $P_1$ to the following core optimization problem $P_2$.

$$P_2 : \text{ maximize } \bar{f}(\frac{1}{x_1}, \ldots, \frac{1}{x_n}) \tag{4}$$

$$s.t. \ \bar{f}(x_1, \ldots, x_n) \leq n + \varepsilon' \tag{5}$$

   where $\bar{f}(x_1, \ldots, x_n) = \sum_{i=1}^n f(x_i) = \sum_{i=1}^n x_i - \log x_i \ (x_i \in (0, \infty))$.

3. *Investigating $f(x)$.* $f(x)$ lies in the core of the problem. We have proven several properties of $f(x)$ in Section 3. A fundamental property of $f(x)$ (stated in Lemma B.1a) is that $f(x)$ is strictly convex and takes its minimum value 1 at $x = 1$. These lemmas allow us to conduct further analysis in other parts of this paper.

4. *Allocating $\varepsilon'$.* In problem $P_2$, the supremum of $\bar{f}(\frac{1}{x_1}, \ldots, \frac{1}{x_n})$ is affected by the domain of each dimension, which is in turn determined by how $\varepsilon'$ is allocated to these dimensions. We call $(\varepsilon_1, \cdots, \varepsilon_n)$ where $\sum_1^n \varepsilon_i = \varepsilon'$ as an *allocation*. We prove that $\bar{f}(\frac{1}{x_1}, \ldots, \frac{1}{x_n})$ takes its maximum when $\varepsilon'$ is allocated to only one dimension (*i.e.*, an "extreme" allocation). In other words, there exists one $\varepsilon_j = \varepsilon'$ and $\varepsilon_i = 0$ for all $i \neq j$. The key is to prove the convexity of function $\Delta(\varepsilon) = f(\frac{1}{w_1(\varepsilon)}) - f(w_1(\varepsilon))$, where $w_1(\varepsilon) = -W_0(-e^{-(1+\varepsilon)})$.

In summary, we use a linear transformation to simplify problem $P_1$ into standard Gaussian case. Then we deal with the simplified problem in the following Lemma 1, which accomplishes the above steps $2 \sim 4$. Theorem 1 can be seen as the generalization of Lemma 1.

**Lemma 1** *Let $\mathcal{N}(0, I)$ be n-dimensional standard Gaussian distribution, $\varepsilon$ be a positive number. For any n-dimensional Gaussian distribution $\mathcal{N}(\boldsymbol{\mu}, \boldsymbol{\Sigma})$,*

*(a) If $KL(\mathcal{N}(\boldsymbol{\mu}, \boldsymbol{\Sigma})||\mathcal{N}(0, I)) \leq \varepsilon$, then*

$$KL(\mathcal{N}(0, I)||\mathcal{N}(\boldsymbol{\mu}, \boldsymbol{\Sigma})) \leq \frac{1}{2}\left(\frac{1}{-W_0(-e^{-(1+2\varepsilon)})} - \log\frac{1}{-W_0(-e^{-(1+2\varepsilon)})} - 1\right)$$

*(b) If $KL(\mathcal{N}(0, I)||\mathcal{N}(\boldsymbol{\mu}, \boldsymbol{\Sigma})) \leq \varepsilon$, then*

$$KL(\mathcal{N}(\boldsymbol{\mu}, \boldsymbol{\Sigma})||\mathcal{N}(0, I)) \leq \frac{1}{2}\left(\frac{1}{-W_0(-e^{-(1+2\varepsilon)})} - \log\frac{1}{-W_0(-e^{-(1+2\varepsilon)})} - 1\right)$$

**Proof 1** *Please see Section C in the Appendix for details.* □

To further investigate the bound in Theorem 1, we expand the Lambert $W$ function using the series presented in [19, 14] for small $\varepsilon$. This result is stated in the following Theorem 2, which can help users to apply our theorem conveniently.

**Theorem 2** *For any two n-dimensional Gaussian distributions $\mathcal{N}(\boldsymbol{\mu}_1, \boldsymbol{\Sigma}_1)$, $\mathcal{N}(\boldsymbol{\mu}_2, \boldsymbol{\Sigma}_2)$, and a small positive number $\varepsilon$, if $KL(\mathcal{N}(\boldsymbol{\mu}_1, \boldsymbol{\Sigma}_1)||\mathcal{N}(\boldsymbol{\mu}_2, \boldsymbol{\Sigma}_2)) \leq \varepsilon$, then*

$$KL(\mathcal{N}(\boldsymbol{\mu}_2, \boldsymbol{\Sigma}_2)||\mathcal{N}(\boldsymbol{\mu}_1, \boldsymbol{\Sigma}_1)) \leq \varepsilon + 2\varepsilon^{1.5} + O(\varepsilon^2)$$

**Proof 2** *Please see Appendix E for details.* □

Theorem 1 holds for any two Gaussians $\mathcal{N}(\boldsymbol{\mu}_1, \boldsymbol{\Sigma}_1)$ and $\mathcal{N}(\boldsymbol{\mu}_2, \boldsymbol{\Sigma}_2)$. According to the proof of Theorem 1 (Lemma 1), one of $\mathcal{N}(\boldsymbol{\mu}_1, \boldsymbol{\Sigma}_1)$ and $\mathcal{N}(\boldsymbol{\mu}_2, \boldsymbol{\Sigma}_2)$ can be fixed. Thus, it is not hard to extend Lemma 1 to the case where the fixed Gaussian is not standard. We can apply linear transformation on the fixed Gaussian (see Equation (D.46)) as what we have done in the main proof of Theorem 1 (see Appendix D). Other parts of the proof are the same. Therefore, we obtain the following corollary.

**Corollary 1** *Theorem 1 and Theorem 2 hold when one of $\mathcal{N}(\boldsymbol{\mu}_1, \boldsymbol{\Sigma}_1)$ and $\mathcal{N}(\boldsymbol{\mu}_2, \boldsymbol{\Sigma}_2)$ is fixed.*

**Remark 1** *The supremum in Theorem 1 is small (zero) when $\varepsilon$ is small (zero). Figure A.2 in Appendix A shows the graph of the supremum of KL divergence. Due to the strict conditions, it is hard to reach the supremum in typical applications (e.g. in machine learning). Notably, the bound is independent of the dimension n. This is critical in high-dimensional problems (see motivation application in Section 6). We also note that the condition $KL(\mathcal{N}(\boldsymbol{\mu}_1, \boldsymbol{\Sigma}_1)||\mathcal{N}(\boldsymbol{\mu}_2, \boldsymbol{\Sigma}_2)) \leq \varepsilon$ in Theorem 1 is strict in high-dimensional problems.*

**Remark 2** *Theorem 1 gives the strict conditions when the supremum can be attained. We can benefit from these strict conditions in applications. When the forward KL divergence is small, we want a guarantee that the reverse KL divergence is also small such that bounding forward KL divergence also bounds the reverse KL divergence. Theorem 1 has the following two meanings.*

1. *The supremum of reverse KL divergence $\varepsilon + 2\varepsilon^{1.5} + O(\varepsilon^2)$ is small, implying that the worst case is acceptable.*
2. *The strict conditions for reaching the supremum imply that the worst case barely happens in practice. When these strict conditions do not hold, the reverse KL divergence is smaller than the supremum, which is what we want in practice.*

**Toy Examples**.

Figure 1 shows some toy examples in one dimensional case. The black line represents standard Gaussian distribution $\mathcal{N}_0(0, 1)$. All other four Gaussian distributions $\mathcal{N}_i$ ($1 \leq i \leq 4$, in colored lines) have the same forward KL divergence $KL(\mathcal{N}_i||\mathcal{N}_0) = 0.01$. The second distribution has the maximized reverse KL divergence $KL(\mathcal{N}_0||\mathcal{N}_1(0, 0.90173^2)) \approx 0.01148$, which is equal to the supremum $\frac{1}{2}\left(\frac{1}{-W_0(-e^{-(1+2\times0.01)})} - \log\frac{1}{-W_0(-e^{-(1+2\times0.01)})} - 1\right)$. Other reverse KL divergences are $KL(\mathcal{N}_0||\mathcal{N}_2(0, 1.10161^2)) \approx 0.00879$, $KL(\mathcal{N}_0||\mathcal{N}_3(0.14143, 1)) \approx 0.01$, $KL(\mathcal{N}_0||\mathcal{N}_4(0.1, 1.07153^2)) \approx 0.00892$.

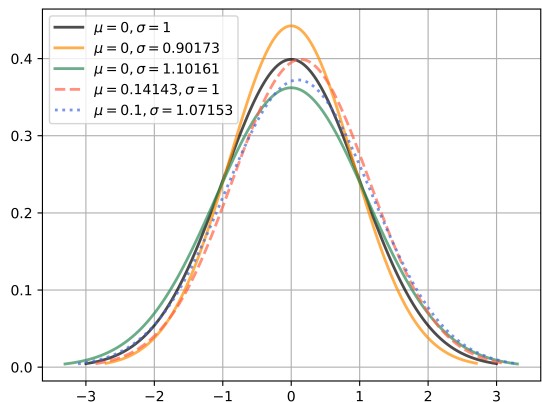

Figure 1: One dimensional toy examples.

## 4.2 Infimum of Reverse KL Divergence Between Gaussians

In this subsection, we give the infimum of $KL(\mathcal{N}_2||\mathcal{N}_1)$ when $KL(\mathcal{N}_1||\mathcal{N}_2) \geq M$ $(M > 0)$. The result is presented in Theorem 3.

**Theorem 3** *For any two n-dimensional Gaussian distributionss $\mathcal{N}(\boldsymbol{\mu}_1, \boldsymbol{\Sigma}_1)$ and $\mathcal{N}(\boldsymbol{\mu}_2, \boldsymbol{\Sigma}_2)$, if $KL(\mathcal{N}(\boldsymbol{\mu}_1, \boldsymbol{\Sigma}_1)||\mathcal{N}(\boldsymbol{\mu}_2, \boldsymbol{\Sigma}_2)) \geq M$ $(M > 0)$, then*

$$KL(\mathcal{N}(\boldsymbol{\mu}_2, \boldsymbol{\Sigma}_2)||\mathcal{N}(\boldsymbol{\mu}_1, \boldsymbol{\Sigma}_1)) \geq \frac{1}{2}\left(\frac{1}{-W_{-1}(-e^{-(1+2M)})} - \log \frac{1}{-W_{-1}(-e^{-(1+2M)})} - 1\right)$$

*The infimum is attained when the following two conditions hold.*

*(1) There exists only one eigenvalue $\lambda_j$ of $B_2^{-1}\boldsymbol{\Sigma}_1(B_2^{-1})^{\top}$ or $B_1^{-1}\boldsymbol{\Sigma}_2(B_1^{-1})^{\top}$ equal to $-W_{-1}(-e^{-(1+2M)})$ and all other eigenvalues $\lambda_i$ $(i \neq j)$ are equal to 1, where $B_1 = P_1 D_1^{1/2}$, $P_1$ is an orthogonal matrix whose columns are the eigenvectors of $\boldsymbol{\Sigma}_1$, $D_1 = diag(\lambda_1, \ldots, \lambda_n)$ whose diagonal elements are the corresponding eigenvalues, $B_2$ is defined in the similar way as $B_1$ except on $\boldsymbol{\Sigma}_2$.*

*(2) $\boldsymbol{\mu}_1 = \boldsymbol{\mu}_2$.*

**Proof 3** *Theorem 1 and Theorem 3 form a duality. These two theorems can be proved independently in a similar way and also be derived from each other. We give two proofs of Theorem 3 in Appendix F. The first proof presented in Appendix F.1 has the similar structure as that of Theorem 1, except that Theorem 3 uses $W_{-1}$. The second proof shown in Appendix F.2 derives Theorem 3 from Theorem 1. These two proofs can verify each other.* $\square$

## 5 Relaxed Triangle Inequality

In this section, we give a dimension-free bound of $KL(\mathcal{N}_1||\mathcal{N}_3)$ when $KL(\mathcal{N}_1||\mathcal{N}_2)$ and $KL(\mathcal{N}_2||\mathcal{N}_3)$ are bounded for any three Gaussians $\mathcal{N}_1$, $\mathcal{N}_2$, and $\mathcal{N}_3$. Proving the relaxed triangle inequality is much more difficult. The main result is presented in Theorem 4 and 5. We use two key Lemmas G.5 and 2 to accomplish the key steps of the proof.

**Theorem 4** *For any three n-dimensional Gaussians* $\mathcal{N}(\boldsymbol{\mu}_i, \boldsymbol{\Sigma}_i)(i \in \{1, 2, 3\})$ *such that* $KL(\mathcal{N}(\boldsymbol{\mu}_1, \boldsymbol{\Sigma}_1)||\mathcal{N}(\boldsymbol{\mu}_2, \boldsymbol{\Sigma}_2)) \le \varepsilon_1$ *and* $KL(\mathcal{N}(\boldsymbol{\mu}_2, \boldsymbol{\Sigma}_2)||\mathcal{N}(\boldsymbol{\mu}_3, \boldsymbol{\Sigma}_3)) \le \varepsilon_2$ *for* $\varepsilon_1, \varepsilon_2 \ge 0$,

$$KL(\mathcal{N}(\boldsymbol{\mu}_1, \boldsymbol{\Sigma}_1)||\boldsymbol{\Sigma}(\boldsymbol{\mu}_3, \boldsymbol{\Sigma}_3)) < \varepsilon_1 + \varepsilon_2 + \frac{1}{2}\left( W_{-1}(-e^{-(1+2\varepsilon_1)})W_{-1}(-e^{-(1+2\varepsilon_2)}) + W_{-1}(-e^{-(1+2\varepsilon_1)}) \right.$$

$$\left. + W_{-1}(-e^{-(1+2\varepsilon_2)}) + 1 - W_{-1}(-e^{-(1+2\varepsilon_2)})\left(\sqrt{2\varepsilon_1} + \sqrt{\frac{2\varepsilon_2}{-W_0(-e^{-(1+2\varepsilon_2)})}}\right)^2 \right)$$

**Overview of Proof of Theorem 4**

The proof of Theorem 4 is the most technical part of this paper. Please see Appendix I for details. Here we give the overview of the proof.

We want to solve the following optimization problem $P_3$ analytically.

$$P_3 : \text{ maximize } KL(\mathcal{N}(\boldsymbol{\mu}_1, \boldsymbol{\Sigma}_1)||\mathcal{N}(\boldsymbol{\Sigma}(\boldsymbol{\mu}_3, \boldsymbol{\Sigma}_3))$$
$$\text{s.t. } KL(\mathcal{N}(\boldsymbol{\mu}_1, \boldsymbol{\Sigma}_1)||\mathcal{N}(\boldsymbol{\mu}_2, \boldsymbol{\Sigma}_2)) \le \varepsilon_1$$
$$KL(\mathcal{N}(\boldsymbol{\mu}_2, \boldsymbol{\Sigma}_2)||\mathcal{N}(\boldsymbol{\mu}_3, \boldsymbol{\Sigma}_3)) \le \varepsilon_2$$

Unfortunately, it is hard to find the supremum due to the complexity caused by Lambert $W$ function. So we relax the constraints to simplify the problem. The proof mainly consists of the following four steps.

1. *Invertible linear transformation*. The first step is similar to that of Theorem 1. We apply a linear transformation on $\mathcal{N}_1$, $\mathcal{N}_2$, and $\mathcal{N}_3$ simultaneously such that $\mathcal{N}_2$ is converted to standard Gaussian. The transformed problem will be proved in Lemma 2 (shown below). In the end, these three Gaussians are transformed back to general case.
2. *Relaxing constraints*. In the proof of Lemma 2, we relax the constraints and finally reduce to the following core problem $P_4$.

$$P_4 : \text{ maximize } \sum_{i=1}^{n} \lambda_{1,[i]}\lambda'_{2,[i]} - \log \lambda_{1,[i]}\lambda'_{2,[i]} \tag{6}$$
$$\text{s.t. } \lambda_{1,[i]} - \log \lambda_{1,[i]} = 1 + \varepsilon_{1,[i]} \ (1 \le i \le n)$$
$$\bigwedge_{i=1}^{n} \varepsilon_{1,[i]} \ge 0 \wedge \sum_{i=1}^{n} \varepsilon_{1,[i]} = 2\varepsilon_1$$
$$\lambda'_{2,[i]} - \log \lambda'_{2,[i]} = 1 + \varepsilon_{2,[i]} \ (1 \le i \le n)$$
$$\bigwedge_{i=1}^{n} \varepsilon_{2,[i]} \ge 0 \wedge \sum_{i=1}^{n} \varepsilon_{2,[i]} = 2\varepsilon_2$$

where $\lambda_{1,[i]}$ and $\lambda'_{2,[i]}$ are the eigenvalues of $\boldsymbol{\Sigma}_1$ and $\boldsymbol{\Sigma}_2^{-1}$ arranged in decreasing order, respectively. $\varepsilon_{1,[i]}$ and $\varepsilon_{2,[i]}$ are arranged in decreasing order too.
3. *Allocating $2\varepsilon_1$ and $2\varepsilon_2$*. The value of objective function in $P_4$ is determined by $\lambda_{1,[i]}$ and $\lambda'_{2,[i]}$, which are in turn determined by how $2\varepsilon_1$ and $2\varepsilon_2$ are allocated to $(\varepsilon_{1,[1]}, \cdots, \varepsilon_{1,[n]})$ and $(\varepsilon_{2,[1]}, \cdots, \varepsilon_{2,[n]})$, respectively. We prove that an "extreme allocation" can maximize the objective function. In other words, the objective function in Equation (6) takes its maximum when $\varepsilon_{1,[1]} = 2\varepsilon_1$, $\varepsilon_{2,[1]} = 2\varepsilon_2$, and $\varepsilon_{1,[i]} = \varepsilon_{2,[i]} = 0$ for $1 < i \le n$. In the proof, we use a key Lemma G.5 to deal with the 2-dimensional case ($n = 2$). Finally, we extend to arbitrary dimensional cases.
4. *Dealing with 2-dimensional case*. The proof of Lemma G.5 is the most technical part in this work. In the proof, concentrating $\varepsilon_1$ and $\varepsilon_2$ is much harder than that in the last section for Theorem 1. $f(x) = x - \log x$ is a transcendental function whose inverse function is expressed by Lambert $W$ function. This makes even a 2-dimensional case of problem $P_4$ hard to solve. Our proof of Lemma G.5 is like coordinate descent but much difficult. We show that, for any fixed "non-extreme allocation" of $2\varepsilon_1$ (*i.e.*, $(\varepsilon_{1,[1]}, \varepsilon_{1,[2]})$ where $\varepsilon_{1,[2]} > 0$), there exists

a "more extreme" allocation of $2\varepsilon_2$ (*i.e.*, $(\varepsilon_{2,[1]}, \varepsilon_{2,[2]})$ where $\varepsilon_{2,[2]} > 0$) that maximizes the objective function. Note that it is hard to state that $(\varepsilon_{2,[1]}, \varepsilon_{2,[2]})$ is "more extreme" than $(\varepsilon_{1,[1]}, \varepsilon_{1,[2]})$ because $\varepsilon_1 \neq \varepsilon_2$. Then we fix $(\varepsilon_{2,[1]}, \varepsilon_{2,[2]})$ and find a "more extreme" allocation of $2\varepsilon_1$ $(\varepsilon'_{1,[1]}, \varepsilon'_{1,[2]})$ to lift the objective function further. Using these iterations, we can construct an infinite sequence of allocations whose limitation is an "extreme" one (*i.e.*, $\varepsilon_{1,[1]} = 2\varepsilon_1, \varepsilon_{2,[1]} = 2\varepsilon_2$). Then we prove the extreme allocation can make the objective function reach its supremum. In Appendix G we present the key Lemma G.5 and its proof. We also give its proof sketch before the long proof. Essentially, Lemma G.5 plays the most vital role in eliminating the dimension $n$ in case of $n = 2$ from the bound in 2-dimensional case. Finally, we will make the bound in high-dimensional problem dimension-free as well.

In summary, we apply a linear transformation to simplify the problem. Then we solve the simplified problem in the following Lemma 2, accomplishing the above steps $2 \sim 4$. Theorem 4 can be seen as the generalization of Lemma 2.

**Lemma 2** *For any two n-dimensional Gaussian distributions $\mathcal{N}(\boldsymbol{\mu}_1, \boldsymbol{\Sigma}_1)$ and $\mathcal{N}(\boldsymbol{\mu}_2, \boldsymbol{\Sigma}_2)$ such that $KL(\mathcal{N}(\boldsymbol{\mu}_1, \boldsymbol{\Sigma}_1)||\mathcal{N}(0, I)) \leq \varepsilon_1$, $KL(\mathcal{N}(0, I)||\mathcal{N}(\boldsymbol{\mu}_2, \boldsymbol{\Sigma}_2)) \leq \varepsilon_2$ $(\varepsilon_1, \varepsilon_2 \geq 0)$,*

$$KL(\mathcal{N}(\boldsymbol{\mu}_1, \boldsymbol{\Sigma}_1)||\mathcal{N}(\boldsymbol{\mu}_2, \boldsymbol{\Sigma}_2)) < \varepsilon_1 + \varepsilon_2 + \frac{1}{2}\left( W_{-1}(-e^{-(1+2\varepsilon_1)})W_{-1}(-e^{-(1+2\varepsilon_2)}) + W_{-1}(-e^{-(1+2\varepsilon_1)}) \right.$$

$$\left. + W_{-1}(-e^{-(1+2\varepsilon_2)}) + 1 - W_{-1}(-e^{-(1+2\varepsilon_2)}) \left( \sqrt{2\varepsilon_1} + \sqrt{\frac{2\varepsilon_2}{-W_0(-e^{-(1+2\varepsilon_2)})}} \right)^2 \right)$$

**Proof 4** *Please see Appendix H for details.* □

**Remark 3** *The bound in Lemma 2 is dimension-free and becomes 0 when $\varepsilon_1 = \varepsilon_2 = 0$.*

Similarly, we can expand Lambert $W$ function by series [19, 14] and simplify the bound in Theorem 4 as follows [36].

**Theorem 5** *For any three n-dimensional Gaussian distributions $\mathcal{N}(\boldsymbol{\mu}_i, \boldsymbol{\Sigma}_i)(i \in \{1, 2, 3\})$ such that $KL(\mathcal{N}(\boldsymbol{\mu}_1, \boldsymbol{\Sigma}_1)||\mathcal{N}(\boldsymbol{\mu}_2, \boldsymbol{\Sigma}_2)) \leq \varepsilon_1$ and $KL(\mathcal{N}(\boldsymbol{\mu}_2, \boldsymbol{\Sigma}_2)||\mathcal{N}(\boldsymbol{\mu}_3, \boldsymbol{\Sigma}_3)) \leq \varepsilon_2$ for small $\varepsilon_1, \varepsilon_2 \geq 0$,*

$$KL(\mathcal{N}(\boldsymbol{\mu}_1, \boldsymbol{\Sigma}_1)||\mathcal{N}(\boldsymbol{\mu}_3, \boldsymbol{\Sigma}_3)) < 3\varepsilon_1 + 3\varepsilon_2 + 2\sqrt{\varepsilon_1\varepsilon_2} + o(\varepsilon_1) + o(\varepsilon_2)$$

**Proof 5** *Please see Appendix J for details.* □

Finally, in the proof of Theorem 4, we use invertible linear transformation to convert $\mathcal{N}_2$ to standard Gaussian while preserving KL divergence. This proof also applies to the case when $\mathcal{N}(\boldsymbol{\mu}_2, \boldsymbol{\Sigma}_2)$ is fixed. Therefore, we obtain the following corollary.

**Corollary 2** *Theorem 4 and 5 hold when $\mathcal{N}(\boldsymbol{\mu}_2, \boldsymbol{\Sigma}_2)$ is fixed.*

**Remark 4** *Our theorem is different from existing generalized Pythagoras inequalities satisfied by KL divergence. Please see Appendix K.1 for more discussion.*

# 6 Discussion and Applications

In our theorems, we allow all parameters are unknown or one Gaussian is fixed. Therefore, our theorems are suitable for problems where the parameters can vary. This is common in deep learning where the parameters are learned from data. In this section, we discuss motivation application and other applications ranging from anomaly detection to reinforcement learning to sample complexity research.

## 6.1 Motivation Application: Anomaly Detection with Flow-based Model

The research problems in this paper are motivated by our research on deep anomaly detection using flow-based model [57] [3]. Flow-based model assigns higher likelihoods to Out-of-Distribution (OOD) data than In-Distribution (ID) data (*i.e.*, training data) [39, 51, 13, 52, 40, 30]. For example, Glow [29] assigns higher likelihoods for SVHN when trained on CIFAR-10. Furthermore, we cannot sample OOD data from the model. Existing explanation is based on the discrepancy of typical set and high probability density regions of model distribution [40]. Such typicality-based explanation and OOD detection method fail when OOD data has coinciding likelihoods with ID data [40].

We focus on two research problems in this context. (1) why we cannot sample OOD data from flow-based model with prior regardless of when OOD data have higher, lower, or coinciding likelihoods? (2) How to detect OOD data using flow-based model? We investigate these problems from a statistical divergence perspective. Let $z = f(x)$ be the flow-based model mapping data $x$ in data space to representation $z$ in latent space. Suppose the prior distribution $p_Z^r$ is the most commonly used Gaussian distribution. Let $X_1 \sim p_X(x)$, $X_2 \sim q_X(x)$ be the distributions of ID and OOD datasets, respectively. $Z_1 = f(X_1) \sim p_Z(z)$, $Z_2 = f(X_2) \sim q_Z(z)$ be the distributions of representations of ID and OOD datasets, respectively. Let $p_X^r$ be the model induced distribution such that $Z_r \sim p_Z^r$ and $X_r = f^{-1}(Z_r) \sim p_X^r$. Flow-based model is usually trained by maximum likelihood estimation, which is equal to minimizing forward KL divergence $KL(p_X||p_X^r)$ [44, 23]. We conduct generalized Shapiro-Wilk test for multivariate normality on representations. As shown in the original Table C.3 in [57] (also in supplementary material), $p_Z$ is Gaussian-like for all datasets. $q_Z$ is also Gaussian-like for OOD datasets with higher or coinciding likelihoods except for just one case. These results allow us to approximate $p_Z$ and $q_Z$ with Gaussians when possible.

The theorems proved in this paper provide solid theoretical guarantee for our analysis and algorithm. On one hand, flow-based model preserves KL divergence (see Proposition D.1 in Appendix D), so $KL(p_Z||p_Z^r)$ which equals $KL(p_X||p_X^r)$ is minimized. According to the approximate symmetry of small KL divergence (Theorem 1), we can know $KL(p_Z^r||p_Z)$ is small too. So we can assume $p_Z^r \approx p_Z$ when $KL(p_Z||p_Z^r)$ is sufficiently small. On the other hand, we can also assume that the distributions of ID and OOD data are far from each other. This implies that $KL(p_X||q_X)$ equaling $KL(p_Z||q_Z)$ can be any large. This implies $KL(p_Z^r||q_Z) \approx KL(p_Z||q_Z)$ is large too. Specially, when $q_Z$ is Gaussian-like, we can apply the relaxed triangle inequality (Theorem 4) and infer that $KL(p_Z^r||q_Z)$ must be large. Note that, when $q_Z$ is not Gaussian-like, we can still apply the theorems presented in this paper to perform analysis on the lower bound of KL divergence (see original Theorem 5 in [57]). Overall, the large KL divergence between $p_Z^r$ and $q_Z$ reveals why we cannot sample OOD data from flow-based model with prior. It is also notable that flow-based model constructs diffeomorphism between data and latent space with thousands of dimensions. Thus, it is critical that the bounds found in this paper are dimension-free. Furthermore, we decompose the KL divergence further into group-wise KL divergence and mutual information. Based on these analysis, we propose an unified OOD detection algorithm KLODS both for group (GAD) and point-wise (PAD) anomaly detection. We conduct extensive experiments to compare our method with 13 baseline methods including $t$-test, KS-test, MMD [24], KSD [35], Annulus Method [13], typicality test [40], the state-of-the-art (SOTA) GAD method GOD2KS [28], input complexity compensated likelihood [50], last-scale likelihood [49], ODIN [34], Joint confidence loss [33], and DoSE [37]. Experimental results demonstrate the superiority of our method. For example, as shown in Table 2, our method outperforms the SOTA group-wise anomaly detection method GOD2KS on flow-based model by 9.1% AUROC. Our method also outperforms the SOTA point-wise anomaly detection method DoSE with Glow by 5.2% AUROC. More details of the algorithm and experimental results on both group and point-wise anomaly detection can be refered to [57].

## 6.2 Applications of Approximate Symmetry of Small KL divergence

Theorem 1 can be also applied widely in deep reinforcement learning and sample complexity research. In many contexts, researchers are hindered by the asymmetry of KL divergence. Theorem 1 on the approximate symmetry of KL divergence between Gaussians brings the following convenience to us.

1. Minimizing one of forward and reverse KL divergences also bounds another.
2. We can exchange forward and reverse KL divergences for small $\varepsilon$.

---

[3] We append an anonymous version of our work [57] in the supplementary material for convinience.

Table 2: Group-wise anomaly detection Results (AUROC and AUPR in percentage) of our method KLODS and the SOTA method GOD2KS on Glow with batch sizes 5 and 10. We run our method for 5 times. Results of GOD2KS are referred from [28]. The higher the better.

| ID | OOD | batch size=5 | | | | batch size=10 | | | |
| | | KLODS | | GOD2KS | | KLODS | | GOD2KS | |
| | | AUROC | AUPR | AUROC | AUPR | AUROC | AUPR | AUROC | AUPR |
|---|---|---|---|---|---|---|---|---|---|
| FashionMNIST | MNIST | **99.8±0.0** | **99.8±0.0** | 98 | 98 | **100.0±0.0** | **100.0±0.0** | **100** | **100** |
| | KMNIST | **99.9±0.0** | **99.9±0.0** | 97 | 96 | **100.0±0.0** | **100.0±0.0** | **100** | **100** |
| | Omniglot | **100.0±0.0** | **100.0±0.0** | **100** | **100** | **100.0±0.0** | **100.0±0.0** | **100** | **100** |
| SVHN | CelebA | **100.0±0.0** | **100.0±0.0** | **100** | 99 | **100.0±0.0** | **100.0±0.0** | **100** | **100** |
| | CIFAR-10 | **100.0±0.0** | **100.0±0.0** | 92 | 84 | **100.0±0.0** | **100.0±0.0** | 99 | 98 |
| | CIFAR-100 | **100.0±0.0** | **100.0±0.0** | 93 | 86 | **100.0±0.0** | **100.0±0.0** | 99 | 98 |
| | LSUN | **100.0±0.0** | **100.0±0.0** | 99 | 98 | **100.0±0.0** | **100.0±0.0** | **100.0** | **100.0** |
| CIFAR-10 | CelebA | **99.2±0.1** | **99.4±0.1** | 86 | 92 | **100.0±0.0** | **100.0±0.0** | 96 | 98 |
| | SVHN | 97.6±0.2 | 97.8±0.2 | 96 | **98** | 99.8±0.0 | 99.8±0.0 | **100** | **100** |
| | LSUN | **100.0±0.0** | **100.0±0.0** | 60 | 58 | **100.0±0.0** | **100.0±0.0** | 58 | 56 |
| CelebA | CIFAR-10 | **99.6±0.0** | **99.6±0.0** | 84 | 73 | **100.0±0.0** | **100.0±0.0** | 94 | 91 |
| | CIFAR-100 | **99.8±0.0** | **99.8±0.0** | 82 | 71 | **100.0±0.0** | **100.0±0.0** | 94 | 90 |
| | SVHN | **100.0±0.0** | **100.0±0.0** | 97 | 98 | **100.0±0.0** | **100.0±0.0** | **100** | **100** |
| | LSUN | **100.0±0.0** | **100.0±0.0** | 85 | 75 | **100.0±0.0** | **100.0±0.0** | 96 | 92 |
| | **average** | **99.7** | **99.7** | 90.6 | 87.6 | **100.0** | **100.0** | 95.4 | 94.5 |

We summarize the applications of Theorem 1 briefly in the following. The details of these applications are discussed in Appendix L.1.

**Providing Theoretical Guarantee for Continuous Gaussian Policy in Reinforcement Learning**. In [38], Nair *et al.* propose AWAC method to accelerate online reinforcement learning with offline datasets. They obtain theoretical guarantee in offline reinforcement learning for discrete policies. Theorem 1 can extend their guarantee to continuous Gaussian policy.

**Bringing New Insights to Existing Reinforcement Learning Algorithm**. In [1], Abdolmaleki *et al.* propose the MPO algorithm for reinforcement learning. They use Expectation-Maximization (EM) to solve control problems and use constraints on KL terms in both E and M-steps. Theorem 1 can eliminate such a difference for continuous Gaussian policies.

**Bridging Research on Sample Complexity of Learning Gaussian Distribution**. Theorem 1 can bridge existing research on sample complexity of Gaussian distribution. Researchers have proposed algorithms for learning a multivariate Gaussian distribution with error bounds in forward and reverse KL divergence separately so far [6, 8]. Theorem 1 can eliminate the difference between forward and reverse KL divergence in this scenario.

### 6.3 Application of Relaxed Triangle Inequality

**Extending One-step Safety Guarantee to Multiple Steps in Reinforcement Learning**. The relaxed triangle inequality (Theorem 5) has been applied in safe reinforcement learning. Liu *et al.* propose an Expectation-Maximization style approach for learning safe policy in reinforcement learning [36]. They utilize our relaxed triangle inequality to extend one-step robustness guarantee to multiple steps. In the original Proposition 4 in [36], they simplify the bound in Theorem 4 in case $\varepsilon_1 = \varepsilon_2$. Please see Appendix L.2 for details.

## 7 Conclusion

In this paper, we research the properties of KL divergence between Gaussians. First, we find the supremum of reverse KL divergence $KL(\mathcal{N}_2||\mathcal{N}_1)$ if the forward KL divergence $KL(\mathcal{N}_1||\mathcal{N}_2) \leq \varepsilon$ ($\varepsilon > 0$). This conclusion quantifies the approximate symmetry of small KL divergence between Gaussians. We also find the infimum of $KL(\mathcal{N}_2||\mathcal{N}_1)$ if $KL(\mathcal{N}_1||\mathcal{N}_2) \geq M$ ($M > 0$). We give the conditions when the supremum and infimum can be attained. Second, we find a bound for $KL(\mathcal{N}_1||\mathcal{N}_3)$ when $KL(\mathcal{N}_1||\mathcal{N}_2)$ and $KL(\mathcal{N}_2||\mathcal{N}_3)$ are bounded. This indicates that KL divergence between Gaussians follows a relaxed triangle inequality. All the bounds in this paper are independent of the dimension of distributions. Finally, we discuss the applications of our theorems in deep anomaly detection, reinforcement learning, and sample complexity research.

## Acknowledgments and Disclosure of Funding

This work is supported by the National Natural Science Foundation of China (Grant No. 62002107, 62372162, 62032024, 62172429, 61872371, 62202150, 62102442) and National Key Research and Development Program of China (Grant No. 2021ZD40300).

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
