# Appendices of On the Properties of Kullback-Leibler Divergence Between Multivariate Gaussian Distributions

**Yufeng Zhang, Jialu Pan**[*]**, Kenli Li**
College of Computer Science and Electronic Engineering
Hunan University
Changsha, China
{yufengzhang, jialupan, lkl}@hnu.edu.cn

**Wanwei Liu, Zhenbang Chen, Xinwang Liu**
College of Computer
National University of Defense Technology
Changsha, China
{wwliu, zbchen, xinwangliu}@nudt.edu.cn

**Ji Wang**
College of Computer
State Key Laboratory for High Performance Computing,
Key laboratory of Software Engineering for Complex Systems
National University of Defense Technology
Changsha, China
wj@nudt.edu.cn

## Contents

[*]Jialu Pan is the corresponding author.

37th Conference on Neural Information Processing Systems (NeurIPS 2023).

## A   Figures and Notations

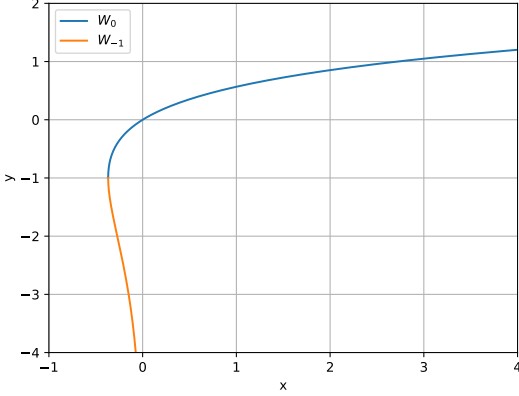

Figure A.1: The Lambert $W$ function. The upper branch (in blue) is the graph of principal branch $W_0$. The lower branch (in orange) is the graph of $W_{-1}$.

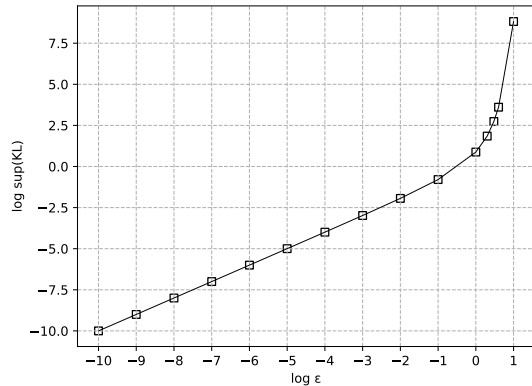

Figure A.2: Values of supremum of KL divergence shown on a logarithmic scale.

**Notations**. Table A.1 shows a full list of the notations used in this paper.

Table A.1: Notations.

| | |
|---|---|
| $f(x)$ | $x - \log x$ $(x \in \mathbb{R}^{++})$ |
| $W(x)$ | the Lambert $W$ function |
| $W_0(x)$ | the principal branch (branch 0) of $W(x)$ |
| $W_{-1}(x)$ | the branch $-1$ of $W(x)$ |
| $w_1(t)$ | the smaller solution of $f(x) = 1 + t$ $(t \geq 0)$ |
| $w_2(t)$ | the larger solution of $f(x) = 1 + t$ $(t \geq 0)$ |
| $\bar{f}(x_1, \ldots, x_n)$ | $\sum_{i=1}^{n} f(x_i)$ |
| $\lambda$ | the eigenvalue of matrix |
| $\lambda^*$ | the largest eigenvalue of matrix |
| $\lambda_*$ | the least eigenvalue of matrix |
| $f_l(x)$ | $f(1 - x) - 1$ $(0 \leq x < 1)$ |
| $f_r(x)$ | $f(x + 1) - 1$ $(x \geq 0)$ |
| $g_l(\varepsilon)$ | $f_l^{-1}(\varepsilon)$, the inverse function of $f_l$ |
| $g_r(\varepsilon)$ | $f_r^{-1}(\varepsilon)$, the inverse function of $f_r$ |
| $\mathcal{N}(0, I)$ | standard Gaussian distribution, dimension $n$ is eliminated for brevity |

# B    Lemmas Related to $f(x)$

The following Lemma B.1 will be used in all other parts of this paper. Lemma B.1 uses the Lambert $W$ function frequently, which has the following derivative.

$$W'(x) = \frac{1}{x + e^{W(x)}} = \frac{W(x)}{x(1 + W(x))} \quad (x \neq 0, -e^{-1}) \tag{B.1}$$

**Lemma B.1** *Given function $f(x) = x - \log x$ $(x \in \mathbb{R}^{++})$ ($\mathbb{R}^{++}$ is the set of positive real numbers), the following propositions hold.*

(a) *$f(x)$ is strictly convex and takes the minimum value 1 at $x = 1$.*
(b) *$f(x) > f(1/x)$ for $x > 1$ and $f(x) < f(1/x)$ for $0 < x < 1$.*
(c) *The inverse function of $f$ is $f^{-1}(x) = -W(-e^{-x})$ $(x \geq 1)$.*
(d) *The solutions of equation $x - \log x = 1 + t$ $(t \geq 0)$ are $w_1(t) = -W_0(-e^{-(1+t)}) \in (0, 1]$ and $w_2(t) = -W_{-1}(-e^{-(1+t)}) \in [1, +\infty)$. It is easy to know $w_1(0) = w_2(0) = 1$. We treat $w_1(t), w_2(t)$ as functions of $t$.*

*(e) The derivatives of $w_1(t)$ and $w_2(t)$ are*

$$w_1'(t) = \frac{-w_1(t)}{1 - w_1(t)} = \frac{W_0(-e^{-(1+t)})}{W_0(-e^{-(1+t)}) + 1} < 0 \tag{B.2}$$

$$w_2'(t) = \frac{-w_2(t)}{1 - w_2(t)} = \frac{W_1(-e^{-(1+t)})}{W_1(-e^{-(1+t)}) + 1} > 0 \tag{B.3}$$

*(f) For $t > 0$, $f(w_1(t)) < f(\frac{1}{w_1(t)})$, $f(\frac{1}{w_2(t)}) < f(w_2(t))$.*

*(g) If $f(x) \leq 1 + t$ $(t \geq 0)$, then $w_1(t) \leq x \leq w_2(t)$ and*

$$S(t) = \sup_{\substack{t \geq 0 \\ f(x) \leq 1+t}} f\left(\frac{1}{x}\right) = f\left(\frac{1}{w_1(t)}\right) \tag{B.4}$$

*(h) If $f(x) \geq 1 + t$ $(t \geq 0)$, then $0 < x \leq w_1(t) \vee x \geq w_2(t)$ and*

$$I(t) = \inf_{\substack{t \geq 0 \\ f(x) \geq 1+t}} f\left(\frac{1}{x}\right) = f\left(\frac{1}{w_2(t)}\right) \tag{B.5}$$

*(i) For $t \geq 0$, $f'(w_2(t)) \leq -f'(\frac{1}{w_2(t)})$.*

*(j) For $t_1, t_2 \geq 0$,*

$$f(w_1(t_1)w_2(t_2)) = t_1 + t_2 + 2 + w_1(t_1)w_1(t_2) - w_1(t_1) - w_1(t_2) \tag{B.6}$$

$$f(w_2(t_1)w_2(t_2)) = t_1 + t_2 + 2 + w_2(t_1)w_2(t_2) - w_2(t_1) - w_2(t_2) \tag{B.7}$$

**Proof 1**    *(a) This is because $f'(x) = 1 - \frac{1}{x}$, $f''(x) = \frac{1}{x^2} > 0$*

*(b) We note $\Delta(x) = f(\frac{1}{x}) - f(x) = \frac{1}{x} - x + 2\log x$. Then $\Delta'(x) = -(\frac{1}{x} - 1)^2 \leq 0$ and $\Delta(1) = 0$*
*So it is easy to know Lemma B.1b holds.*

*(c) We can verify this by definition.*

$$y - \log y = x \iff e^{y-x} = y \iff (-y)e^{-y} = -e^{-x} \iff y = -W(-e^{-x})$$

*(d) We can get Lemma B.1d from B.1c immediately.*

*(e) According to Equation (B.1), we can have*

$$\frac{\mathrm{d}w_1(t)}{\mathrm{d}t} = -\frac{\mathrm{d}(W_0(-e^{-(1+t)}))}{\mathrm{d}t} = \frac{-W_0(-e^{-(1+t)})}{-e^{-(1+t)}(1 + W_0(-e^{-(1+t)}))} \times \frac{\mathrm{d}(-e^{-(1+t)})}{\mathrm{d}t}$$

$$= \frac{W_0(-e^{-(1+t)})}{W_0(-e^{-(1+t)}) + 1} = \frac{-w_1(t)}{1 - w_1(t)}$$

*The derivative of $w_2(t)$ can be computed in a similar way.*

*(f) From Lemma B.1b, we can know Lemma B.1f.*

*(g) This is because*

$$f(x) \leq 1 + t \implies w_1(t) < x < w_2(t) \implies \frac{1}{w_2(t)} < \frac{1}{x} < \frac{1}{w_1(t)}$$

*Combining Lemma B.1b, we have*

$$f\left(\frac{1}{w_2(t)}\right) < f(w_2(t)) = 1 + t = f(w_1(t)) < f\left(\frac{1}{w_1(t)}\right)$$

*Thus Equation (B.4) holds. It is also easy to know that $S(t) = f(\frac{1}{w_1(t)})$ is continuous and strictly increasing with $t$.*

*(h) We have*

$$f(x) \geq 1 + t \implies x \leq w_1(t) \vee x \geq w_2(t) \implies \frac{1}{x} \leq \frac{1}{w_2(t)} \vee \frac{1}{x} \geq \frac{1}{w_1(t)} \tag{B.8}$$

*Combining Lemma B.1b, we have $f(\frac{1}{w_2(t)}) < f(\frac{1}{w_1(t)})$, so we have Lemma B.1h.*

*(i) Since $f'(x) = 1 - \frac{1}{x}$ and $w_2(t) \geq 1$ for $t \geq 0$, we have*

$$f'(w_2(t)) = 1 - \frac{1}{w_2(t)} = \frac{w_2(t) - 1}{w_2(t)} \leq w_2(t) - 1 = -\left(1 - \frac{1}{\frac{1}{w_2(t)}}\right) = -f'(\frac{1}{w_2(t)}) \quad \text{(B.9)}$$

*(j)*

$$
\begin{aligned}
&f(w_1(t_1)w_1(t_2)) \\
=&w_1(t_1)w_1(t_2) - \log w_1(t_1)w_1(t_2) \\
=&w_1(t_1)w_1(t_2) + (w_1(t_1) - \log w_1(t_1)) + (w_1(t_2) - \log w_1(t_2)) - w_1(t_1) - w_1(t_2) \\
=&w_1(t_1)w_1(t_2) + 1 + t_1 + 1 + t_2 - w_1(t_1) - w_1(t_2) \\
=&t_1 + t_2 + 2 + w_1(t_1)w_1(t_2) - w_1(t_1) - w_1(t_2)
\end{aligned}
\quad \text{(B.10)}
$$

*where the third equation follows from Lemma B.1d. Equation (B.7) can be proved in a similar way.*

$\square$

## C  Proof of Lemma 1

To prove Lemma 1 (Section 4.1 in the main context), we need to prove the following Lemma C.2 in Subsection C.1 first. Then we prove Lemma 1a and 1b in Subsection C.2 and C.3, respectively.

### C.1  Lemma

**Lemma C.2** *Given $n$-ary function $\bar{f}(x) = \bar{f}(x_1, \ldots, x_n) = \sum_{i=1}^n x_i - \log x_i$ ($x_i \in \mathbb{R}^{++}$), if $\bar{f}(x_1, \ldots, x_n) \leq n + \varepsilon(\varepsilon > 0)$, then*

$$\sup \bar{f}(\frac{1}{x_1}, \ldots, \frac{1}{x_n}) = \frac{1}{-W_0(-e^{-(1+\varepsilon)})} - \log \frac{1}{-W_0(-e^{-(1+\varepsilon)})} + n - 1 \quad \text{(C.11)}$$

*The supremum is attained when there exists only one $j$ such that $f(x_j) = 1 + \varepsilon$ and $f(x_i) = 1$ for $i \neq j$.*

**Proof 2** *We want to solve the following optimization problem analytically.*

$$\text{maximize } \bar{f}(\frac{1}{x_1}, \ldots, \frac{1}{x_n}) \quad \text{(C.12)}$$

$$\text{s.t. } \bar{f}(x_1, \ldots, x_n) \leq n + \varepsilon \quad \text{(C.13)}$$

*Since $f(x) \geq 1$, the constraint $\bar{f}(x_1, \ldots, x_n) = \sum_{i=1}^n f(x_i) = \sum_{i=1}^n x_i - \log x_i \leq n + \varepsilon$ can be replaced by the following constraints*

$$\left(\bigwedge_{i=1}^n f(x_i) = x_i - \log x_i \leq 1 + \varepsilon_i\right) \wedge \left(\bigwedge_{i=1}^n \varepsilon_i \geq 0\right) \wedge \sum_{i=1}^n \varepsilon_i \leq \varepsilon \quad \text{(C.14)}$$

*Given fixed $\varepsilon_1, \ldots, \varepsilon_n$ such that $\bigwedge_{i=1}^n \varepsilon_i \geq 0 \wedge \sum_{i=1}^n \varepsilon_i \leq \varepsilon$, we define*

$$\bar{S}(\varepsilon_1, \ldots, \varepsilon_n) = \sup_{\bigwedge_{i=1}^n f(x_i) \leq 1 + \varepsilon_i} \bar{f}(\frac{1}{x_1}, \ldots, \frac{1}{x_n}) = \sum_{i=1}^n \sup_{f(x_i) \leq 1 + \varepsilon_i} f(\frac{1}{x_i}) = \sum_{i=1}^n S(\varepsilon_i) \quad \text{(C.15)}$$

*So we have*

$$\sup \bar{f}(\frac{1}{x_1}, \ldots, \frac{1}{x_n}) = \sup_{\substack{\bigwedge_{i=1}^n \varepsilon_i \geq 0 \\ \sum_{i=1}^n \varepsilon_i \leq \varepsilon}} \bar{S}(\varepsilon_1, \ldots, \varepsilon_n) \quad \text{(C.16)}$$

*It is easy to know that $\bar{S}(\varepsilon_1, \ldots, \varepsilon_n)$ is continuous and strictly increasing with $\varepsilon_1, \ldots, \varepsilon_n$. So the condition $\sum_{i=1}^n \varepsilon_i \leq \varepsilon$ in Equation (C.16) can be changed to $\sum_{i=1}^n \varepsilon_i = \varepsilon$.*

The remaining proof consists of two steps. In step 1, we find $\bar{S}(\varepsilon_1, \ldots, \varepsilon_n)$ for fixed $\varepsilon_1, \ldots, \varepsilon_n$. In step 2, we find $\sup \bar{S}(\varepsilon_1, \ldots, \varepsilon_n)$ for any $\varepsilon_1, \ldots, \varepsilon_n$ satisfying $\bigwedge_{i=1}^{n} \varepsilon_i \geq 0 \wedge \sum_{i=1}^{n} \varepsilon_i = \varepsilon$.

**Step 1**: *According to Lemma B.1g, for fixed $\varepsilon_i$ we get*

$$S(\varepsilon_i) = \sup_{f(x) \leq 1+\varepsilon_i} f\left(\frac{1}{x}\right) = f\left(\frac{1}{w_1(\varepsilon_i)}\right) \tag{C.17}$$

*Plugging Equation (C.17) into Equation (C.15), we get*

$$\bar{S}(\varepsilon_1, \ldots, \varepsilon_n) = \sum_{i=1}^{n} f\left(\frac{1}{w_1(\varepsilon_i)}\right) \tag{C.18}$$

**Step 2**: *We define function*

$$\Delta(\varepsilon) = f\left(\frac{1}{w_1(\varepsilon)}\right) - f(w_1(\varepsilon)) = \frac{1}{w_1(\varepsilon)} - w_1(\varepsilon) + 2\log w_1(\varepsilon) \tag{C.19}$$

*Now we prove*

$$\Delta(t\varepsilon) \leq t\Delta(\varepsilon) \ (0 \leq t < 1) \tag{C.20}$$

*When $\varepsilon = 0$, it is trivial to verify that $\Delta(0) = 0$. In the following we show that $\Delta(\varepsilon)$ is strictly increasing and strictly convex. It is easy to know $\frac{d\Delta(\varepsilon)}{dw_1} = -\frac{1}{w_1^2} + \frac{2}{w_1} - 1$. Combining Lemma B.1e, the derivative of $\Delta(\varepsilon)$ is*

$$\frac{d\Delta(\varepsilon)}{d\varepsilon} = \frac{d\Delta(\varepsilon)}{dw_1} \times \frac{dw_1(\varepsilon)}{d\varepsilon} = \left(-\frac{1}{w_1(\varepsilon)^2} + \frac{2}{w_1(\varepsilon)} - 1\right) \times \frac{-w_1(\varepsilon)}{1 - w_1(\varepsilon)} = \frac{1}{w_1(\varepsilon)} - 1$$

*The second order derivative of $\Delta(\varepsilon)$ is*

$$\frac{d^2\Delta(\varepsilon)}{d\varepsilon^2} = -\frac{1}{w_1(\varepsilon)^2} \frac{-w_1(\varepsilon)}{1 - w_1(\varepsilon)} = \frac{1}{w_1(\varepsilon)(1 - w_1(\varepsilon))}$$

*Since $w_1(\varepsilon) \in (0, 1)$ for $\varepsilon > 0$, it is easy to know $\frac{d\Delta(\varepsilon)}{d\varepsilon} > 0, \frac{d^2\Delta(\varepsilon)}{d\varepsilon^2} > 0$ for $\varepsilon > 0$. This indicates that $\Delta(\varepsilon)$ is strictly increasing and strictly convex on $(0, +\infty)$. Thus, for any $\varepsilon', \varepsilon'' > 0$, we have $\Delta((1-t)\varepsilon' + t\varepsilon'') < (1-t)\Delta(\varepsilon') + t\Delta(\varepsilon'')$ for any $0 < t < 1$. Remember that we have known $\Delta(0) = 0$. Since $\Delta(\varepsilon)$ is continuous, it is easy to know*

$$\Delta(t\varepsilon'') = \lim_{\varepsilon' \to 0} \Delta((1-t)\varepsilon' + t\varepsilon'') \leq \lim_{\varepsilon' \to 0} (1-t)\Delta(\varepsilon') + t\Delta(\varepsilon'') = t\Delta(\varepsilon'') \tag{C.21}$$

*Thus, we can obtain Equation (C.20).*

*Therefore, for any $\varepsilon_1, \ldots, \varepsilon_n$ satisfying $\bigwedge_{i=1}^{n} \varepsilon_i \geq 0 \wedge \sum_{i=1}^{n} \varepsilon_i = \varepsilon$, we have*

$$\bar{\Delta}(\varepsilon_1, \ldots, \varepsilon_n) = \sum_{i=1}^{n} f\left(\frac{1}{w_1(\varepsilon_i)}\right) - f(w_1(\varepsilon_i)) = \sum_{i=1}^{n} \Delta(\varepsilon_i) = \sum_{i=1}^{n} \Delta\left(\frac{\varepsilon_i}{\varepsilon}\varepsilon\right) \leq \sum_{i=1}^{n} \frac{\varepsilon_i}{\varepsilon}\Delta(\varepsilon) = \Delta(\varepsilon) \tag{C.22}$$

*Inequality (C.22) is tight when there exists only one $j$ such that $\varepsilon_j = \varepsilon$ and $\varepsilon_i = 0$ for all $i \neq j$. This means that for any $\varepsilon_1, \ldots, \varepsilon_n$ satisfying $\bigwedge_{i=1}^{n} \varepsilon_i \geq 0 \wedge \sum_{i=1}^{n} \varepsilon_i = \varepsilon$, the following inequality holds.*

$$
\begin{aligned}
&\bar{S}(\varepsilon_1, \ldots, \varepsilon_n) \\
&= \sum_{i=1}^{n} f\left(\frac{1}{w_1(\varepsilon_i)}\right) && \text{(C.18)} \\
&\leq \Delta(\varepsilon) + \sum_{i=1}^{n} f(w_1(\varepsilon_i)) && \text{(C.22)} \\
&= \frac{1}{w_1(\varepsilon)} - \log\frac{1}{w_1(\varepsilon)} - (w_1(\varepsilon) - \log w_1(\varepsilon)) + \sum_{i=1}^{n}(1 + \varepsilon_i) && \text{\scriptsize Lemma B.1d} \\
& && \text{(C.19)} \\
&= \frac{1}{w_1(\varepsilon)} - \log\frac{1}{w_1(\varepsilon)} - (1 + \varepsilon) + n + \varepsilon \\
&= \frac{1}{w_1(\varepsilon)} - \log\frac{1}{w_1(\varepsilon)} + n - 1 \\
&= \frac{1}{-W_0(-e^{-(1+\varepsilon)})} - \log\frac{1}{-W_0(-e^{-(1+\varepsilon)})} + n - 1 && \text{(C.23)}
\end{aligned}
$$

*Finally, we have*

$$\sup \bar{f}(\frac{1}{x_1}, \ldots, \frac{1}{x_n}) = \sup_{\substack{\wedge_{i=1}^n \varepsilon_i \geq 0 \\ \Sigma_{i=1}^n \varepsilon_i \leq \varepsilon}} \bar{S}(\varepsilon_1, \ldots, \varepsilon_n) = \frac{1}{-W_0(-e^{-(1+\varepsilon)})} - \log \frac{1}{-W_0(-e^{-(1+\varepsilon)})} + n - 1$$

$$(C.24)$$

$\bar{f}(1/x_1, \ldots, 1/x_n)$ *reaches its supremum when there exists only one $j$ such that $f(x_j) = 1 + \varepsilon$ and $f(x_i) = 1$ for $i \neq j$.*

$\square$

## C.2   Proof of Lemma 1a

**Proof 3** *According to the definition of KL divergence, we have*

$$KL(\mathcal{N}(\boldsymbol{\mu}, \boldsymbol{\Sigma})||\mathcal{N}(0, I)) = \frac{1}{2}\left(-\log|\boldsymbol{\Sigma}| + \mathrm{Tr}(\boldsymbol{\Sigma}) + \boldsymbol{\mu}^\top \boldsymbol{\mu} - n\right)$$

$$KL(\mathcal{N}(0, I)||\mathcal{N}(\boldsymbol{\mu}, \boldsymbol{\Sigma})) = \frac{1}{2}\left(\log|\boldsymbol{\Sigma}| + \mathrm{Tr}(\boldsymbol{\Sigma}^{-1}) + \boldsymbol{\mu}^\top \boldsymbol{\Sigma}^{-1} \boldsymbol{\mu} - n\right)$$

*where $n$ is the dimension of the distribution. The positive definite matrix $\boldsymbol{\Sigma}$ has factorization $\boldsymbol{\Sigma} = PDP^\top$ where $P$ is an orthogonal matrix whose columns are the eigenvectors of $\boldsymbol{\Sigma}$, $D = diag(\lambda_1, \ldots, \lambda_n)$ $(\lambda_i > 0)$ whose diagonal elements are the corresponding eigenvalues. We also have*

$$|\boldsymbol{\Sigma}| = |P||D||P^\top| = |D| = \prod_{i=1}^n \lambda_i \tag{C.25}$$

$$\log|\boldsymbol{\Sigma}| = \sum_{i=1}^n \log \lambda_i, \; -\log|\boldsymbol{\Sigma}| = \sum_{i=1}^n \log \frac{1}{\lambda_i} \tag{C.26}$$

$$\mathrm{Tr}(\boldsymbol{\Sigma}) = \mathrm{Tr}(PDP^\top) = \mathrm{Tr}(P^\top PD) = \mathrm{Tr}(D) = \sum_{i=1}^n \lambda_i \tag{C.27}$$

$$\mathrm{Tr}(\boldsymbol{\Sigma}^{-1}) = \sum_{i=1}^n \lambda_i' = \sum_{i=1}^n \frac{1}{\lambda_i} \tag{C.28}$$

*where $\lambda_i' = 1/\lambda_i$ are eigenvalues of $\boldsymbol{\Sigma}^{-1}$.*

*If $KL(\mathcal{N}(\boldsymbol{\mu}, \boldsymbol{\Sigma})||\mathcal{N}(0, I)) \leq \varepsilon$, we have $-\log|\boldsymbol{\Sigma}| + \mathrm{Tr}(\boldsymbol{\Sigma}) + \boldsymbol{\mu}^\top \boldsymbol{\mu} - n \leq 2\varepsilon$. This condition is equal to the following conditions*

$$-\log|\boldsymbol{\Sigma}| + \mathrm{Tr}(\boldsymbol{\Sigma}) = \sum_{i=1}^n \lambda_i - \log \lambda_i \leq n + \varepsilon_1 \tag{C.29}$$

$$\boldsymbol{\mu}^\top \boldsymbol{\mu} \leq 2\varepsilon - \varepsilon_1 \tag{C.30}$$

$$0 \leq \varepsilon_1 \leq 2\varepsilon \tag{C.31}$$

*In the following, we find the maximum of $\log|\boldsymbol{\Sigma}| + \mathrm{Tr}(\boldsymbol{\Sigma}^{-1})$ and $\boldsymbol{\mu}^\top \boldsymbol{\Sigma}^{-1} \boldsymbol{\mu}$, respectively. From Equation (C.29), we have*

$$\sum_{i=1}^n \lambda_i - \log \lambda_i \leq n + \varepsilon_1 \tag{C.32}$$

*Applying Lemma C.2 on Inequality (C.32), we can obtain*

$$\sum_{i=1}^n \frac{1}{\lambda_i} - \log \frac{1}{\lambda_i} = \log|\boldsymbol{\Sigma}| + \mathrm{Tr}(\boldsymbol{\Sigma}^{-1}) \leq \frac{1}{-W_0(-e^{-(1+\varepsilon_1)})} - \log \frac{1}{-W_0(-e^{-(1+\varepsilon_1)})} + n - 1 \tag{C.33}$$

*Moreover, since $f(x) = x - \log x$ takes the minimum value $f(1) = 1$ at $x = 1$, it is easy to know $\lambda_i - \log \lambda_i \leq 1 + \varepsilon_1$ from Inequality (C.32). According to Lemma B.1g, we know*

$$w_1(\varepsilon_1) \leq \lambda_i \leq w_2(\varepsilon_1), \; \frac{1}{w_2(\varepsilon_1)} \leq \lambda_i' = \frac{1}{\lambda_i} \leq \frac{1}{w_1(\varepsilon_1)} \tag{C.34}$$

We also have $\mu^\top \Sigma^{-1} \mu \le \lambda'^* \mu^\top \mu$ where $\lambda'^*$ is the maximum eigenvalue of $\Sigma^{-1}$. Combining Equation (C.30) and (C.34), we can know

$$\mu^\top \Sigma^{-1} \mu \le \lambda'^*(2\varepsilon - \varepsilon_1) \le \frac{2\varepsilon - \varepsilon_1}{w_1(\varepsilon_1)} \tag{C.35}$$

Now note that Inequalities (C.33) and (C.35) are tight simultaneously when there exists one $\lambda_j = w_1(\varepsilon_1)$ and all other $\lambda_i = 1$ for $i \ne j$. Thus, we can add the two sides of Inequalities (C.33) and (C.35) and get

$$\begin{aligned}
&KL(\mathcal{N}(0, I) || \mathcal{N}(\mu, \Sigma)) \\
=& \frac{1}{2} \left( \log |\Sigma| + \mathrm{Tr}(\Sigma^{-1}) + \mu^\top \Sigma^{-1} \mu - n \right) \\
\le& \frac{1}{2} \left( \frac{1}{-W_0(-e^{-(1+\varepsilon_1)})} - \log \frac{1}{-W_0(-e^{-(1+\varepsilon_1)})} + n - 1 + \frac{2\varepsilon - \varepsilon_1}{w_1(\varepsilon_1)} - n \right) \\
=& \frac{1}{2} \left( \frac{1 + 2\varepsilon - \varepsilon_1}{w_1(\varepsilon_1)} - \log \frac{1}{w_1(\varepsilon_1)} - 1 \right) \\
=& U(\varepsilon_1) \ (0 \le \varepsilon_1 \le 2\varepsilon)
\end{aligned} \tag{C.36}$$

Notice that the derivative of $U(\varepsilon_1)$ is

$$U'(\varepsilon_1) = \frac{1}{2} \left( \frac{w_1(\varepsilon_1) + 2\varepsilon - \varepsilon_1}{w_1(\varepsilon_1)(1 - w_1(\varepsilon_1))} - \frac{1}{1 - w_1(\varepsilon_1)} \right) = \frac{1}{2} \frac{2\varepsilon - \varepsilon_1}{w_1(\varepsilon_1)(1 - w_1(\varepsilon_1))} \tag{C.37}$$

Since $w_1(\varepsilon_1) \in (0, 1)$ for $\varepsilon_1 > 0$ and $0 \le \varepsilon_1 \le 2\varepsilon$, we can know $U'(\varepsilon_1) \ge 0$ for $\varepsilon_1 > 0$. Thus, $U(\varepsilon_1)$ takes the maximum value at $\varepsilon_1 = 2\varepsilon$. Finally, we have

$$KL(\mathcal{N}(0, I) || \mathcal{N}(\mu, \Sigma)) \le U(2\varepsilon) = \frac{1}{2} \left( \frac{1}{-W_0(-e^{-(1+2\varepsilon)})} - \log \frac{1}{-W_0(-e^{-(1+2\varepsilon)})} - 1 \right) \tag{C.38}$$

Inequality (C.38) is tight only when there exists one $\lambda_j = -W_0(-e^{-(1+2\varepsilon)})$ and all other $\lambda_i = 1$ for $i \ne j$, and $|\mu| = 0$.

We can see that when $\varepsilon$ is small, the right hand side (RHS) of Equation (C.38) is also small.

$\square$

## C.3  Proof of Lemma 1b

**Proof 4** *The proof of Lemma 1b is similar as that for Lemma 1a. We list it here for clarity.*

*The condition $KL(\mathcal{N}(0, I) || \mathcal{N}(\mu, \Sigma)) \le \varepsilon$ is equal to the following conditions*

$$\log |\Sigma| + \mathrm{Tr}(\Sigma^{-1}) \le n + \varepsilon_1 \tag{C.39}$$

$$\mu^\top \Sigma^{-1} \mu \le 2\varepsilon - \varepsilon_1 \tag{C.40}$$

$$0 \le \varepsilon_1 \le 2\varepsilon \tag{C.41}$$

*We can apply Lemma C.2 on Equation (C.39) and get*

$$-\log |\Sigma| + \mathrm{Tr}(\Sigma) \le \frac{1}{-W_0(-e^{-(1+\varepsilon_1)})} - \log \frac{1}{-W_0(-e^{-(1+\varepsilon_1)})} + n - 1 \tag{C.42}$$

*Applying Lemma B.1g on Equation (C.39), we get*

$$w_1(\varepsilon_1) < \lambda' < w_2(\varepsilon_1) \tag{C.43}$$

*From Equation (C.40) we know $\mu^\top \Sigma^{-1} \mu \le 2\varepsilon - \varepsilon_1$. Since $\mu^\top \Sigma^{-1} \mu \ge \lambda'_* \mu^\top \mu$ where $\lambda'_*$ is the minimum eigenvalue of $\Sigma^{-1}$, combining Equation (C.43), we can know*

$$\mu^\top \mu \le \frac{2\varepsilon - \varepsilon_1}{\lambda'_*} \le \frac{2\varepsilon - \varepsilon_1}{w_1(\varepsilon_1)} \tag{C.44}$$

*Adding the two sides of Inequalities (C.42), and (C.44), we get the same result as Equation (C.36). Therefore, we can get the same supremum as follows.*

$$KL(\mathcal{N}(\boldsymbol{\mu}, \boldsymbol{\Sigma})||\mathcal{N}(0, I)) \leq \frac{1}{2}\left(\frac{1}{-W_0(-e^{-(1+2\varepsilon)})} - \log\frac{1}{-W_0(-e^{-(1+2\varepsilon)})} - 1\right) \qquad \text{(C.45)}$$

*Inequality (C.45) is tight only when there exists one $\lambda'_j = -W_0(-e^{-(1+2\varepsilon)})$ and all other $\lambda'_i = 1$ for $i \neq j$, and $|\boldsymbol{\mu}| = 0$.*

$\square$

# D   Proof of Theorem 1

Before our proof, we recall the following proposition which states that diffeomorphism preserves KL divergence ($f$-divergence) [14].

**Proposition D.1** *(See [14]) Let $\boldsymbol{z} = f(\boldsymbol{x})$ be a diffeomorphism, $X_1 \sim p_X$ and $X_2 \sim q_X$ be two random variables and $Z_1 = f(X_1) \sim p_Z$, $Z_2 = f(X_2) \sim q_Z$. Then $KL(p_X||q_X) = KL(p_Z||q_Z)$.*

**Proof 5** *For $X \sim \mathcal{N}(\boldsymbol{\mu}, \boldsymbol{\Sigma})$, there exists an invertible matrix $B$ such that $X' = B^{-1}(X - \boldsymbol{\mu}) \sim \mathcal{N}(0, I)$ [4]. Here $B = PD^{1/2}$, $P$ is an orthogonal matrix whose columns are the eigenvectors of $\boldsymbol{\Sigma}$, $D = diag(\lambda_1, \ldots, \lambda_n)$ whose diagonal elements are the corresponding eigenvalues. For $X_1 \sim \mathcal{N}(\boldsymbol{\mu}_1, \boldsymbol{\Sigma}_1)$ and $X_2 \sim \mathcal{N}(\boldsymbol{\mu}_2, \boldsymbol{\Sigma}_2)$, we define the following linear transformations $T_1, T_2$*

$$X_1^1 = T_1(X_1) = B_1^{-1}(X_1 - \boldsymbol{\mu}_1) \text{ such that } X_1^1 \sim \mathcal{N}(0, I) \qquad \text{(D.46)}$$

$$X_2^2 = T_2(X_2) = B_2^{-1}(X_2 - \boldsymbol{\mu}_2) \text{ such that } X_2^2 \sim \mathcal{N}(0, I) \qquad \text{(D.47)}$$

*and the reverse transformations $T_1^{-1}, T_2^{-1}$ such that $X_1 = T_1^{-1}(X_1^1) = B_1 X_1^1 + \boldsymbol{\mu}_1$ and $X_2 = T_2^{-1}(X_2^2) = B_2 X_1^2 + \boldsymbol{\mu}_2$, where $p_{X_1^1} = p_{X_2^2} = \mathcal{N}(0, I)$. Besides, it is easy to know $X_1^2 = T_2(X_1) = B_2^{-1}(X_1 - \boldsymbol{\mu}_2)$ and $X_2^1 = T_1(X_2) = B_1^{-1}(X_2 - \boldsymbol{\mu}_1)$ are both Gaussian variables. We also have*

$$X_1^2 \sim \mathcal{N}(B_2^{-1}(\boldsymbol{\mu}_1 - \boldsymbol{\mu}_2), B_2^{-1}\boldsymbol{\Sigma}_1(B_2^{-1})^\top), \; X_2^1 \sim \mathcal{N}(B_1^{-1}(\boldsymbol{\mu}_2 - \boldsymbol{\mu}_1), B_1^{-1}\boldsymbol{\Sigma}_2(B_1^{-1})^\top) \qquad \text{(D.48)}$$

*With the help of invertible linear transformations, we can convert the KL divergence between two arbitrary Gaussians into that between one Gaussian and standard Gaussian. According to Proposition D.1, diffeomorphisms preserve KL divergence. If we apply $T_2$ simultaneously on $X_1$, $X_2$, we can have*

$$KL(\mathcal{N}(\boldsymbol{\mu}_1, \boldsymbol{\Sigma}_1)||\mathcal{N}(\boldsymbol{\mu}_2, \boldsymbol{\Sigma}_2)) = KL(p_{X_1^2}||p_{X_2^2}) = KL(p_{X_1^2}||\mathcal{N}(0, I)) \qquad \text{(D.49)}$$

*Then we can apply $T_2^{-1}$ on $X_1^2$, $X_2^2$ and also have*

$$KL(\mathcal{N}(0, I)||p_{X_1^2}) = KL(p_{X_2^2}||p_{X_1^2}) = KL(\mathcal{N}(\boldsymbol{\mu}_2, \boldsymbol{\Sigma}_2)||\mathcal{N}(\boldsymbol{\mu}_1, \boldsymbol{\Sigma}_1)) \qquad \text{(D.50)}$$

*According to precondition, it is easy to know $KL(\mathcal{N}(\boldsymbol{\mu}_1, \boldsymbol{\Sigma}_1)||\mathcal{N}(\boldsymbol{\mu}_2, \boldsymbol{\Sigma}_2)) = KL(p_{X_1^2}||\mathcal{N}(0, I))$. Applying Theorem 1a on $KL(p_{X_1^2}||\mathcal{N}(0, I))$, we can prove*

$$KL(\mathcal{N}(0, I)||p_{X_1^2}) = KL(\mathcal{N}(\boldsymbol{\mu}_2, \boldsymbol{\Sigma}_2)||\mathcal{N}(\boldsymbol{\mu}_1, \boldsymbol{\Sigma}_1)) \leq \frac{1}{2}\left(\frac{1}{-W_0(-e^{-(1+2\varepsilon)})} - \log\frac{1}{-W_0(-e^{-(1+2\varepsilon)})} - 1\right)$$
$$\text{(D.51)}$$

*Similarly, if we use $T_1$ simultaneously on $X_1$ and $X_2$, we can get the same result.*

*Inequality (D.51) is tight when there exists only one eigenvalue $\lambda_j$ of $B_2^{-1}\boldsymbol{\Sigma}_1(B_2^{-1})^\top$ or $B_1^{-1}\boldsymbol{\Sigma}_2(B_1^{-1})^\top$ is equal to $-W_0(-e^{-(1+2\varepsilon)})$ and all other eigenvalues $\lambda_i$ ($i \neq j$) are equal to 1, and $\boldsymbol{\mu}_1 = \boldsymbol{\mu}_2$.*

$\square$

# E   Proof of Theorem 2

**Proof 6** *When $\varepsilon$ is small, we can use the series expanding $W_0$ (see Section III.17 in [7]) to simplify the bound in Theorem 1.*

Notice that when $\varepsilon$ is small, $-W_0(-e^{-(1+2\varepsilon)})$ is close to 1. According to the series expanding $W_0$ (see Section III.17 in [7]), we have

$$W_0(-e^{-(1+2\varepsilon)}) = -1 + 2\sqrt{\varepsilon} - \frac{4}{3}\varepsilon + \frac{2}{9}\varepsilon^{1.5} + O(\varepsilon^2) \tag{E.52}$$

Now expand the $\log$ term around $-W_0(-e^{-(1+2\varepsilon)}) = 1$ using Taylor series for small $\varepsilon$.

$$\begin{aligned}
&\log(-W_0(-e^{-(1+2\varepsilon)})) \\
&= \log(1 - W_0(-e^{-(1+2\varepsilon)}) - 1) \\
&= -W_0(-e^{-(1+2\varepsilon)}) - 1 - \frac{1}{2}\left(-W_0(-e^{-(1+2\varepsilon)}) - 1\right)^2 \\
&\quad + \frac{1}{3}\left(-W_0(-e^{-(1+2\varepsilon)}) - 1\right)^3 + O\left((-W_0(-e^{-(1+2\varepsilon)}) - 1)^4\right) \\
&= -2\sqrt{\varepsilon} + \frac{4}{3}\varepsilon - \frac{2}{9}\varepsilon^{1.5} + O(\varepsilon^2) - \frac{1}{2}\left(-2\sqrt{\varepsilon} + \frac{4}{3}\varepsilon - \frac{2}{9}\varepsilon^{1.5} + O(\varepsilon^2)\right)^2 \\
&\quad + \frac{1}{3}\left(-2\sqrt{\varepsilon} + \frac{4}{3}\varepsilon - \frac{2}{9}\varepsilon^{1.5} + O(\varepsilon^2)\right)^3 + O(\varepsilon^2) \\
&= -2\sqrt{\varepsilon} - \frac{2}{3}\varepsilon - \frac{2}{9}\varepsilon^{1.5} + O(\varepsilon^2)
\end{aligned} \tag{E.53}$$

Plugging Equation (E.52) and (E.53) into the bound in Theorem 1, we can have

$$\begin{aligned}
&KL(\mathcal{N}(\boldsymbol{\mu}_2, \boldsymbol{\Sigma}_2) || \mathcal{N}(\boldsymbol{\mu}_1, \boldsymbol{\Sigma}_1)) \\
&\leq \frac{1}{2}\left(\frac{1}{1 - 2\sqrt{\varepsilon} + \frac{4}{3}\varepsilon - \frac{2}{9}\varepsilon^{1.5} + O(\varepsilon^2)} + \left(-2\sqrt{\varepsilon} - \frac{2}{3}\varepsilon - \frac{2}{9}\varepsilon^{1.5} + O(\varepsilon^2)\right) - 1\right) \\
&= \frac{1}{2}\frac{2\varepsilon - \frac{4}{3}\varepsilon^{1.5} + O(\varepsilon^2)}{1 - 2\sqrt{\varepsilon} + \frac{4}{3}\varepsilon - \frac{2}{9}\varepsilon^{1.5} + O(\varepsilon^2)} \\
&= \varepsilon + \frac{2\varepsilon^{1.5} + O(\varepsilon^2)}{1 - 2\sqrt{\varepsilon} + \frac{4}{3}\varepsilon - \frac{2}{9}\varepsilon^{1.5} + O(\varepsilon^2)} \\
&= \varepsilon + 2\varepsilon^{1.5} + O(\varepsilon^2)
\end{aligned} \tag{E.54}$$

$\square$

# F  Proofs of Theorem 3

## F.1  The First Proof of Theorem 3

Theorem 3 can be proved using the similar method as that of Theorem 1, except that the proof uses $W_{-1}$. We put the key steps of the proof of Theorem 3 in Lemma F.3 and Lemma F.4.

**Lemma F.3** *Given $n$-ary function* $\bar{f}(\boldsymbol{x}) = \bar{f}(x_1, \ldots, x_n) = \sum_{i=1}^{n} x_i - \log x_i$ $(x_i \in \mathbb{R}^{++})$, *if* $\bar{f}(x_1, \ldots, x_n) \geq n + M(M > 0)$, *then*

$$\inf \bar{f}\left(\frac{1}{x_1}, \ldots, \frac{1}{x_n}\right) = \frac{1}{-W_{-1}(-e^{-(1+M)})} - \log \frac{1}{-W_{-1}(-e^{-(1+M)})} + n - 1$$

**Proof 7** *The structure of proof of Lemma F.3 is similar to that of Lemma C.2. The constraint in the following optimization problem*

$$\text{minimize } \bar{f}\left(\frac{1}{x_1}, \ldots, \frac{1}{x_n}\right) \tag{F.55}$$

$$\text{s.t. } \sum_{i=1}^{n} x_i - \log x_i \geq n + M \tag{F.56}$$

*can be replaced by the following constraints*

$$\bigwedge_{i=1}^{n} f(x_i) = x_i - \log x_i \geq 1 + M_i \wedge \bigwedge_{i=1}^{n} M_i \geq 0 \wedge \sum_{i=1}^{n} M_i \geq M \tag{F.57}$$

*Given fixed $M_1, \ldots, M_n$ such that $\bigwedge_{i=1}^{n} M_i \geq 0 \wedge \sum_{i=1}^{n} M_i \geq M$, we define*

$$\bar{I}(M_1, \ldots, M_n) = \inf_{\bigwedge_{i=1}^{n} f(x_i) \geq 1 + M_i} \bar{f}\left(\frac{1}{x_1}, \ldots, \frac{1}{x_n}\right) = \sum_{i=1}^{n} \inf_{f(x_i) \geq 1 + M_i} f\left(\frac{1}{x_i}\right) = \sum_{i=1}^{n} I(M_i) \tag{F.58}$$

*So we have*

$$\inf \bar{f}\left(\frac{1}{x_1}, \ldots, \frac{1}{x_n}\right) = \inf_{\substack{\bigwedge_{i=1}^{n} M_i \geq 0 \\ \sum_{i=1}^{n} M_i \geq M}} \bar{I}(M_1, \ldots, M_n) \tag{F.59}$$

*It is easy to know that $\bar{I}(M_1, \ldots, M_n)$ is continuous and strictly increasing with $M_1, \ldots, M_n$. So the condition $\sum_{i=1}^{n} M_i \geq M$ in Equation (F.59) can be changed to $\sum_{i=1}^{n} M_i = M$.*

*The remaining proof consists of two steps. We find $\bar{I}(M_1, \ldots, M_n)$ for fixed $M_1, \ldots, M_n$ in the first step, and then find $\inf \bar{I}(M_1, \ldots, M_n)$ for any $M_1, \ldots, M_n$ satisfying $\bigwedge_{i=1}^{n} M_i \geq 0 \wedge \sum_{i=1}^{n} M_i = M$ in the second step.*

**Step 1**: *According to Lemma B.1h, for fixed $M_i$, we get*

$$I(M_i) = \inf_{f(x) \geq 1 + M_i} f\left(\frac{1}{x}\right) = f\left(\frac{1}{w_2(M_i)}\right) \tag{F.60}$$

*Combining Equation (F.58), we know*

$$\bar{I}(M_1, \ldots, M_n) = \sum_{i=1}^{n} f\left(\frac{1}{w_2(M_i)}\right) \tag{F.61}$$

**Step 2**: *We define function*

$$\Delta(M) = f(w_2(M)) - f\left(\frac{1}{w_2(M)}\right) = w_2(M) - \frac{1}{w_2(M)} - 2\log w_2(M) \tag{F.62}$$

*Similarly, we can prove $\Delta(tM) \leq t\Delta(M)$ $(0 \leq t < 1)$ by showing $\Delta(0) = 0$ (apparently) and $\Delta(M)$ is strictly increasing and strictly convex. Combining Lemma B.1e, we get the derivative of $\Delta(M)$ as*

$$\frac{d\Delta(M)}{dM} = \left(1 + \frac{1}{w_2(M)^2} - \frac{2}{w_2(M)}\right) \times \frac{dw_2(M)}{dM} = 1 - \frac{1}{w_2(M)} \tag{F.63}$$

*The second order derivative of $\Delta(M)$ is*

$$\frac{d^2\Delta(M)}{dM^2} = \frac{1}{w_2(M)^2} \times \frac{w_2(M)}{w_2(M) - 1} = \frac{1}{w_2(M)(w_2(M) - 1)} \tag{F.64}$$

*Since $w_2(M) \in (1, +\infty)$ for $M > 0$, so it is easy to know $\frac{d\Delta(M)}{dM} > 0, \frac{d^2\Delta(M)}{dM^2} > 0$ for $M > 0$. This implies that $\Delta(M)$ is strictly increasing and strictly convex. We can use the similar deduction as Lemma C.2 to prove $\Delta(tM) \leq t\Delta(M)$. Thus, we have*

$$\bar{\Delta}(M_1, \ldots, M_n) = \sum_{i=1}^{n} f(w_2(M_i)) - f\left(\frac{1}{w_2(M_i)}\right) = \sum_{i=1}^{n} \Delta(M_i) = \sum_{i=1}^{n} \Delta\left(\frac{M_i}{M}M\right) \leq \sum_{i=1}^{n} \frac{M_i}{M}\Delta(M) = \Delta(M)$$
$$\tag{F.65}$$

*Inequality (F.65) is tight when there exists only one $M_j = M$ and all other $M_i = 0$ for $i \neq j$. Therefore, from Inequality (F.65), we can obtain*

$$
\begin{aligned}
\bar{I}(M_1, \ldots, M_n) & \\
&= \sum_{i=1}^{n} f\left(\frac{1}{w_2(M_i)}\right) && \text{(F.61)} \\
&\geq \sum_{i=1}^{n} f(w_2(M_i)) - \Delta(M) && \text{(F.65)} \\
&= \sum_{i=1}^{n} (1 + M_i) - \left(f(w_2(M)) - f\left(\frac{1}{w_2(M)}\right)\right) && \overset{\text{Lemma B.1d}}{\text{(F.62)}} \\
&= n + M - (1 + M) + f\left(\frac{1}{w_2(M)}\right) \\
&= \frac{1}{-W_{-1}(-e^{-(1+M)})} - \log \frac{1}{-W_{-1}(-e^{-(1+M)})} + n - 1 && \text{(F.66)}
\end{aligned}
$$

*Finally, we can conclude that*

$$
\inf \bar{f}\left(\frac{1}{x_1}, \ldots, \frac{1}{x_n}\right) = \inf_{\substack{\wedge_{i=1}^{n} M_i \geq 0 \\ \sum_{i=1}^{n} M_i = M}} \bar{I}(M_1, \ldots, M_n) = \frac{1}{-W_{-1}(-e^{-(1+M)})} - \log \frac{1}{-W_{-1}(-e^{-(1+M)})} + n - 1
$$

$$\text{(F.67)}$$

*Similarly, $\bar{f}(1/x_1, \ldots, 1/x_n)$ reaches its infimum when there exists only one $j$ such that $x_j = -W_{-1}(-e^{-(1+M)})$ and $f(x_i) = 1$ for $i \neq j$.* □

The following Lemma F.4 gives the infimum of KL divergence when one Gaussian is standard.

**Lemma F.4** *For any $n$-dimensional Gaussian distribution $\mathcal{N}(\boldsymbol{\mu}, \boldsymbol{\Sigma})$,*

*(a) If $KL(\mathcal{N}(\boldsymbol{\mu}, \boldsymbol{\Sigma}) || \mathcal{N}(0, I)) \geq M (M > 0)$, then*

$$
KL(\mathcal{N}(0, I) || \mathcal{N}(\boldsymbol{\mu}, \boldsymbol{\Sigma})) \geq \frac{1}{2}\left(\frac{1}{-W_{-1}(-e^{-(1+2M)})} - \log \frac{1}{-W_{-1}(-e^{-(1+2M)})} - 1\right) \quad \text{(F.68)}
$$

*(b) If $KL(\mathcal{N}(0, I) || \mathcal{N}(\boldsymbol{\mu}, \boldsymbol{\Sigma})) \geq M$, then*

$$
KL(\mathcal{N}(\boldsymbol{\mu}, \boldsymbol{\Sigma}) || \mathcal{N}(0, I)) \geq \frac{1}{2}\left(\frac{1}{-W_{-1}(-e^{-(1+2M)})} - \log \frac{1}{-W_{-1}(-e^{-(1+2M)})} - 1\right) \quad \text{(F.69)}
$$

**Proof 8** *(a) We first consider the case when $KL(\mathcal{N}(\boldsymbol{\mu}, \boldsymbol{\Sigma}) || \mathcal{N}(0, I)) = M$. At the end of the proof, we deal with the case when $KL(\mathcal{N}(\boldsymbol{\mu}, \boldsymbol{\Sigma}) || \mathcal{N}(0, I)) \geq M$.*

*The condition $-\log |\boldsymbol{\Sigma}| + \text{Tr}(\boldsymbol{\Sigma}) + \boldsymbol{\mu}^\top \boldsymbol{\mu} - n = 2M$ is equal to*

$$
-\log |\boldsymbol{\Sigma}| + \text{Tr}(\boldsymbol{\Sigma}) = \sum_{i=1}^{n} \lambda_i - \log \lambda_i = n + M_1 \quad \text{(F.70)}
$$

$$
\boldsymbol{\mu}^\top \boldsymbol{\mu} = 2M - M_1 \quad \text{(F.71)}
$$

*where $0 \leq M_1 \leq 2M$.*

*Applying Lemma F.3 on Equation (F.70), we can get*

$$
\log |\boldsymbol{\Sigma}| + \text{Tr}(\boldsymbol{\Sigma}^{-1}) = \sum_{i=1}^{n} \frac{1}{\lambda_i} - \log \frac{1}{\lambda_i} \geq \frac{1}{-W_{-1}(-e^{-(1+M_1)})} - \log \frac{1}{-W_{-1}(-e^{-(1+M_1)})} + n - 1
$$

$$\text{(F.72)}$$

*Inequality (F.72) is tight when all eigenvalues $\lambda_i$ of $\boldsymbol{\Sigma}$ are equal to 1 except for one $\lambda_j = -W_{-1}(-e^{-(1+M_1)})$.*

*From Equation (F.71), we know $\boldsymbol{\mu}^\top \boldsymbol{\Sigma}^{-1} \boldsymbol{\mu} \geq \lambda_*' \boldsymbol{\mu}^\top \boldsymbol{\mu} = \lambda_*'(2M - M_1)$ where $\lambda_*'$ is the smallest eigenvalue of $\boldsymbol{\Sigma}^{-1}$. Here $\lambda^* = 1/\lambda_*'$ is the largest eigenvalue of $\boldsymbol{\Sigma}$. From Equation (F.70), Lemma B.1a and B.1g, we know $\lambda^* \leq -W_{-1}(-e^{-(1+M_1)})$. So we obtain*

$$
\boldsymbol{\mu}^\top \boldsymbol{\Sigma}^{-1} \boldsymbol{\mu} \geq \frac{2M - M_1}{-W_{-1}(-e^{-(1+M_1)})} \quad \text{(F.73)}
$$

*Note that, inequalities (F.73) and (F.72) become tight simultanously when the same condition holds. Now combining Equation (F.72) and (F.73), we obtain*

$$\log|\mathbf{\Sigma}| + \text{Tr}(\mathbf{\Sigma}^{-1}) + \boldsymbol{\mu}^\top\mathbf{\Sigma}^{-1}\boldsymbol{\mu} - n$$

$$\geq \frac{1}{-W_{-1}(-e^{-(1+M_1)})} - \log\frac{1}{-W_{-1}(-e^{-(1+M_1)})} + \frac{2M - M_1}{-W_{-1}(-e^{-(1+M_1)})} - 1$$

$$= \frac{1 + 2M - M_1}{w_2(M_1)} - \log\frac{1}{w_2(M_1)} - 1$$

$$= L(M_1) \ (0 \leq M_1 \leq 2M) \tag{F.74}$$

*It is easy to know that $L'(M_1) = \frac{M_1 - 2M}{w_2(M_1)(w_2(M_1)-1)}$. Since $M_1 \leq 2M$ and $w_2(M_1) > 1$ for $M_1 > 0$, so $L'(M_1) < 0$ ($M_1 > 0$). This indicates that $L(M_1) > L(2M)$ for $0 < M_1 < 2M$. Thus, we can conclude*

$$KL(\mathcal{N}(0, I)||\mathcal{N}(\boldsymbol{\mu}, \mathbf{\Sigma})) \geq \frac{1}{2}L(2M) = \frac{1}{2}\left(\frac{1}{-W_{-1}(-e^{-(1+2M)})} - \log\frac{1}{-W_{-1}(-e^{-(1+2M)})} - 1\right) \tag{F.75}$$

*Inequality (F.75) is tight when there exist only one eigenvalue $\lambda_j$ of $\mathbf{\Sigma}$ equal to $-W_{-1}(-e^{-(1+2M)})$ and all other eigenvalues $\lambda_i (i \neq j)$ are equal to 1, and $\boldsymbol{\mu} = 0$.*

*Finally, we can consider the case when $KL(\mathcal{N}(\boldsymbol{\mu}, \mathbf{\Sigma})||\mathcal{N}(0, I)) \geq M$. The bound in Equation (F.75) is strictly increasing with $M$. Therefore, the precondition $KL(\mathcal{N}(\boldsymbol{\mu}, \mathbf{\Sigma})||\mathcal{N}(0, I)) = M$ can be changed to $KL(\mathcal{N}(\boldsymbol{\mu}, \mathbf{\Sigma})||\mathcal{N}(0, I)) \geq M$.*

*(b) The proof of Lemma 1b is the similar to that of Lemma 1a. We list it here for clarity.*

*The condition $KL(\mathcal{N}(0, I)||\mathcal{N}(\boldsymbol{\mu}, \mathbf{\Sigma})) = M$ is equal to*

$$\log|\mathbf{\Sigma}| + \text{Tr}(\mathbf{\Sigma}^{-1}) = n + M_1 \tag{F.76}$$

$$\boldsymbol{\mu}^\top\mathbf{\Sigma}^{-1}\boldsymbol{\mu} = 2M - M_1 \tag{F.77}$$

*where $0 \leq M_1 \leq 2M$. Applying Lemma F.3 on Equation (F.76), we can obtain*

$$-\log|\mathbf{\Sigma}| + \text{Tr}(\mathbf{\Sigma}) \geq \frac{1}{-W_{-1}(-e^{-(1+M_1)})} - \log\frac{1}{-W_{-1}(-e^{-(1+M_1)})} + n - 1 \tag{F.78}$$

*From Equation (F.76) and Lemma B.1a and B.1g, we have $\lambda' \leq -W_{-1}(-e^{-(1+M_1)})$ where $\lambda'$ is the eigenvalue of $\mathbf{\Sigma}^{-1}$. Now let $\lambda'^*$ be the largest eigenvalues of $\mathbf{\Sigma}^{-1}$. It is easy to know*

$$\lambda'^*\boldsymbol{\mu}^\top\boldsymbol{\mu} \geq \boldsymbol{\mu}^\top\mathbf{\Sigma}^{-1}\boldsymbol{\mu} = 2M - M_1 \implies \boldsymbol{\mu}^\top\boldsymbol{\mu} \geq \frac{2M - M_1}{-W_{-1}(-e^{-(1+M_1)})} \tag{F.79}$$

*Inequalities (F.78) and (F.79) are tight simutanously when there exist only one eigenvalue $\lambda'_j = -W_{-1}(-e^{-(1+M_1)})$ and all other eigenvalues are equal to 1, and $|\boldsymbol{\mu}| = 0$. Therefore, combining Equation (F.78) and (F.79), we obtain*

$$-\log|\mathbf{\Sigma}| + \text{Tr}(\mathbf{\Sigma}) + \boldsymbol{\mu}^\top\boldsymbol{\mu} - n \geq \frac{1}{-W_{-1}(-e^{-(1+M_1)})} - \log\frac{1}{-W_{-1}(-e^{-(1+M_1)})} + \frac{2M - M_1}{-W_{-1}(-e^{-(1+M_1)})} - 1 \tag{F.80}$$

*Finally, using the similar analysis as Equation (F.75), we can conclude that*

$$KL(\mathcal{N}(\boldsymbol{\mu}, \mathbf{\Sigma})||\mathcal{N}(0, I)) \geq \frac{1}{2}\left(\frac{1}{-W_{-1}(-e^{-(1+2M)})} - \log\frac{1}{-W_{-1}(-e^{-(1+2M)})} - 1\right) \tag{F.81}$$

*Similarly, the precondition $KL(\mathcal{N}(0, I)||\mathcal{N}(\boldsymbol{\mu}, \mathbf{\Sigma})) = M$ can be changed to $KL(\mathcal{N}(0, I)||\mathcal{N}(\boldsymbol{\mu}, \mathbf{\Sigma})) \geq M$ because the bound in Equation (F.81) is strictly increasing with $M$.*

**Notes**. *It needs strict conditions to reach the infimum in Lemma F.4.*

*Now we can also obtain Theorem 3 on two general Gaussians. We can use linear transformation on Gaussians and apply Lemma F.4 on them as what we do in the main proof of Theorem 1. The key steps have been proven in Lemma F.3 and F.4. More details are ommited.*

$\square$

## F.2 The Second Proof of Theorem 3

**Proof 9** *In the following, we give the second proof which is drawn from Theorem 1.*

*Suppose $KL(\mathcal{N}(\boldsymbol{\mu}_1, \boldsymbol{\Sigma}_1)||\mathcal{N}(\boldsymbol{\mu}_2, \boldsymbol{\Sigma}_2)) \leq t$ $(t > 0)$, according to Theorem 1, we know*

$$KL(\mathcal{N}(\boldsymbol{\mu}_2, \boldsymbol{\Sigma}_2)||\mathcal{N}(\boldsymbol{\mu}_1, \boldsymbol{\Sigma}_1))$$

$$\leq \frac{1}{2} \left( \frac{1}{-W_0(-e^{-(1+2t)})} - \log \frac{1}{-W_0(-e^{-(1+2t)})} - 1 \right) \tag{F.82}$$

$$= \frac{1}{2} \left( \frac{1}{w_1(2t)} - \log \frac{1}{w_1(2t)} - 1 \right) \tag{F.83}$$

$$= \bar{S}(t) \tag{F.84}$$

*Since $\dfrac{1}{w_1(2t)}$ is strictly increasing with t, $\bar{S}(t)$ is continuous and strictly increasing with t. Besides, the range of function $\bar{S}(t)$ for $(t > 0)$ is $(0, +\infty)$.*

*Given positive number M, according to Theorem 1, there exists $\mathcal{N}(\boldsymbol{\mu}_1, \boldsymbol{\Sigma}_1)$, $\mathcal{N}(\boldsymbol{\mu}_2, \boldsymbol{\Sigma}_2)$ and m such that*

$$\bar{S}(m) = M \tag{F.85}$$

$$KL(\mathcal{N}(\boldsymbol{\mu}_1, \boldsymbol{\Sigma}_1)||\mathcal{N}(\boldsymbol{\mu}_2, \boldsymbol{\Sigma}_2)) = M \tag{F.86}$$

$$KL(\mathcal{N}(\boldsymbol{\mu}_2, \boldsymbol{\Sigma}_2)||\mathcal{N}(\boldsymbol{\mu}_1, \boldsymbol{\Sigma}_1)) = m \tag{F.87}$$

*Thus, given the precondition $KL(\mathcal{N}(\boldsymbol{\mu}_1, \boldsymbol{\Sigma}_1)||\mathcal{N}(\boldsymbol{\mu}_2, \boldsymbol{\Sigma}_2)) \geq M$, we can know that*

$$\inf KL(\mathcal{N}(\boldsymbol{\mu}_2, \boldsymbol{\Sigma}_2)||\mathcal{N}(\boldsymbol{\mu}_1, \boldsymbol{\Sigma}_1)) \leq m \tag{F.88}$$

*In the following, we show*

$$\inf KL(\mathcal{N}(\boldsymbol{\mu}_2, \boldsymbol{\Sigma}_2)||\mathcal{N}(\boldsymbol{\mu}_1, \boldsymbol{\Sigma}_1)) = m \tag{F.89}$$

*holds. Otherwise, there exists an $m' < m$ and $\mathcal{N}(\boldsymbol{\mu}_1, \boldsymbol{\Sigma}_1)$, $\mathcal{N}(\boldsymbol{\mu}_2, \boldsymbol{\Sigma}_2)$ such that*

$$KL(\mathcal{N}(\boldsymbol{\mu}_1, \boldsymbol{\Sigma}_1)||\mathcal{N}(\boldsymbol{\mu}_2, \boldsymbol{\Sigma}_2)) \geq M \tag{F.90}$$

$$KL(\mathcal{N}(\boldsymbol{\mu}_2, \boldsymbol{\Sigma}_2)||\mathcal{N}(\boldsymbol{\mu}_1, \boldsymbol{\Sigma}_1)) = m' \tag{F.91}$$

*Applying Theorem 1 on Equation (F.91), it is easy to know*

$$\sup KL(\mathcal{N}(\boldsymbol{\mu}_1, \boldsymbol{\Sigma}_1)||\mathcal{N}(\boldsymbol{\mu}_2, \boldsymbol{\Sigma}_2)) = \bar{S}(m') \tag{F.92}$$

*This contradicts with the precondition $KL(\mathcal{N}(\boldsymbol{\mu}_1, \boldsymbol{\Sigma}_1)||\mathcal{N}(\boldsymbol{\mu}_2, \boldsymbol{\Sigma}_2)) \geq M$ because $\bar{S}(m') < \bar{S}(m) = M$. Thus, Equation (F.89) holds.*

*Now we can solve m from $\bar{S}(m) = M$ as follows.*

$$\frac{1}{2} \left( \frac{1}{-W_0(-e^{-(1+2m)})} - \log \frac{1}{-W_0(-e^{-(1+2m)})} - 1 \right) = M$$

$$\Leftrightarrow \frac{1}{-W_0(-e^{-(1+2m)})} - \log \frac{1}{-W_0(-e^{-(1+2m)})} = 1 + 2M$$

$$\Leftrightarrow \frac{1}{-W_0(-e^{-(1+2m)})} = -W_{-1}(-e^{-(1+2M)})$$

$$\Leftrightarrow \frac{1}{-W_{-1}(-e^{-(1+2M)})} = -W_0(-e^{-(1+2m)})$$

$$\Leftrightarrow \frac{1}{-W_{-1}(-e^{-(1+2M)})} - \log \frac{1}{-W_{-1}(-e^{-(1+2M)})} = 1 + 2m$$

$$\Leftrightarrow m = \frac{1}{2} \left( \frac{1}{-W_{-1}(-e^{-(1+2M)})} - \log \frac{1}{-W_{-1}(-e^{-(1+2M)})} - 1 \right) \tag{F.93}$$

*where the third and fifth equations follow from Lemma B.1d. Plugging Equation (F.93) into (F.89), we can prove Theorem 3.*

$\square$

# G Key Lemma G.5 and Its Proof

**Lemma G.5** *Given $f(x) = x - \log x$ ($x \in \mathbb{R}^{++}$), let $\varepsilon_{x,1}, \varepsilon_{x,2}, \varepsilon_{y,1}, \varepsilon_{y,2}$ be four non-negative numbers such that $\varepsilon_{x,1} \geq \varepsilon_{x,2}, \varepsilon_{y,1} \geq \varepsilon_{y,2}$. Then*

$$f(w_2(\varepsilon_{x,1})w_2(\varepsilon_{y,1})) + f(w_2(\varepsilon_{x,2})w_2(\varepsilon_{y,2})) \leq f(w_2(\varepsilon_{x,1} + \varepsilon_{x,2})w_2(\varepsilon_{y,1} + \varepsilon_{y,2})) + 1 \quad \text{(G.94)}$$

Lemma G.5 deals with 2-dimensional case of the core problem of the Theorem on relaxed triangle inequality. The proof of Lemma G.5 is the most technical part in this paper.

In the following, we give the overview of the proof. In Appendix G.1 we give some notations and Lemmas for preparation. In Appendix G.2 we give the main proof of Lemma G.5.

**Overview of proof of Lemma G.5**
In the LHS (resp. RHS) of Inequality (G.94), $\varepsilon_{x,2}$ and $\varepsilon_{y,2}$ stay in the second (resp. first) term. Intuitively, we use Inequality (G.94) to move $\varepsilon_{x,2}, \varepsilon_{y,2}$ into the first item such that $\varepsilon_{x,1}, \varepsilon_{x,2}$ and $\varepsilon_{y,1}, \varepsilon_{y,2}$ are allocated in only one dimension.

We construct a function

$$S(\theta_x, \theta_y) = f(w_2(\varepsilon_{x,1} + \theta_x \varepsilon_{x,2})w_2(\varepsilon_{y,1} + \theta_y \varepsilon_{y,2})) + f(w_2(\varepsilon_{x,2} - \theta_x \varepsilon_{x,2})w_2(\varepsilon_{y,2} - \theta_y \varepsilon_{y,2}))$$

for $-\frac{\varepsilon_{x,1}}{\varepsilon_{x,2}} \leq \theta_x \leq 1, -\frac{\varepsilon_{y,1}}{\varepsilon_{y,2}} \leq \theta_y \leq 1$ (see Equation (G.112)). When $\theta_x = \theta_y = 0$, $S(\theta_x, \theta_y) = S(0,0)$ equals the LHS of Inequality (G.94). Recall that $w_2(0) = 1$ and $f(1) = 1$. So when $\theta_x = \theta_y = 1$, $S(\theta_x, \theta_y) = S(1,1)$ equals the RHS of Inequality (G.94). When increasing $\theta_x$ and $\theta_y$ from 0 to 1, $\varepsilon_{x,2}, \varepsilon_{y,2}$ are moved to the first item gradually and the LHS approaches the RHS of Inequality (G.94) gradually. So the values of $\theta_x, \theta_y$ control how $\varepsilon_{x,2}$ and $\varepsilon_{y,2}$ are allocated among the two terms. We call $(\theta_x, \theta_y)$ and the corresponding pairs $(\varepsilon_{x,1} + \theta_x \varepsilon_{x,2}, \varepsilon_{x,2} - \theta_x \varepsilon_{x,2})$ and $(\varepsilon_{y,1} + \theta_y \varepsilon_{y,2}, \varepsilon_{y,2} - \theta_y \varepsilon_{y,2})$ indiscriminately as *allocations*. When $\theta_x = \theta_y = 1$, we call it an *extreme allocation*. To prove Inequality (G.94), it suffices to show $S(0,0) \leq S(1,1)$.

However, it is hard to prove $S(0,0) \leq S(1,1)$ directly due to the lack of conclusions relating to Lambert $W$ function. We treat the problem as an optimization problem where $S(\theta_x, \theta_y)$ is the objective function. We use an analytical and variation version of coordinate ascent to solve the optimization problem. In the beginning, we start from a well-chosen point (see Equation (G.113)) that can be arbitrarily close to point ($\theta_x = 0, \theta_y = 0$). In each iteration, we fix one of $\theta_x$ and $\theta_y$ and make another vary. In this way, we will construct an infinite sequence of allocations and finally reach the supremum of $S(\theta_x, \theta_y)$.

The proof mainly consists of the following four aspects, which are much more complex than a simple coordinate ascent algorithm.

**A1** We start optimization from a well-chosen point close to ($\theta_x = 0, \theta_y = 0$). In each iteration, once we fix one of $\theta_x$ and $\theta_y$ and make another one vary, we prove that there exists one and only one supremum. For example, once we fix $\theta_x = \theta_{x,0} > 0$ where $\theta_{x,0}$ can be arbitrarily small, we prove that there exists one and only one $-\frac{\varepsilon_{y,1}}{\varepsilon_{y,2}} < \theta_{y,1} < 1$ that maximizes $S(\theta_{x,0}, \theta_y)$ (see Proposition G.2). In the next step, we fix $\theta_y = \theta_{y,1}$ and find $\theta_{x,2}$ to lift $S(\theta_x, \theta_{y,1})$ further. In this way, we find an infinite sequence of allocations $(\theta_{x,0}, \theta_{y,0}), (\theta_{x,0}, \theta_{y,1}), (\theta_{x,2}, \theta_{y,1}), (\theta_{x,2}, \theta_{y,3}), \cdots$. In the end, we will show that in these iterations, the allocations corresponding to these local maximums are more and more *extreme* ones.

**A2** However, the condition describing the local maximum (*e.g.*, $\theta_{y,1}$) is complicated. For example, Equation (G.119) expresses the condition that $\theta_{y,1}$ should satisfy. We cannot solve the Equation analytically. Luckily, we turn to analyze the condition to study the property of these local maximums. We use a crucial transformation (in Equations (G.120) ~ (G.124)) to obtain key Equations (G.124), (G.138), (G.142) which express the property of these local maximums (corresponding to allocations) implicitly. Equations (G.124), (G.138), (G.142) also express the relation between local maximums obtained in neighboring iterations.

**A3** In the above iterations, we are lifting the objective function gradually. Furthermore, we show that these allocations are increasingly *extreme* ones. There are two problems we need to tackle. First, how to measure the *extremeness* of one allocation. We find a formula (see Equation (G.148) and (G.149)) to measure the extremeness of allocations indirectly. Second, how to compare the extremeness of two allocations. The Equations used to characterize

the property of local maximums obtained in Aspect **A2** are also used to prove that these allocations obtained in a sequence of iterations are more and more extreme.

**A4** We prove that the limit of the allocation sequence can maximize the objective function.

## G.1 Notations and Lemmas for Lemma G.5

**Notations.** We define the following helper functions based on $f(x) = x - \log x$.

$$f_l(x) = f(1 - x) - 1 \ (0 \le x < 1), \ f_r(x) = f(x + 1) - 1 \ (x \ge 0) \tag{G.95}$$

The derivatives of $f_l(x), f_r(x)$ are

$$f_l'(x) = -f'(1 - x) = \frac{1}{1 - x} - 1, \ f_r'(x) = f'(1 + x) = 1 - \frac{1}{x + 1} \tag{G.96}$$

So both $f_l(x)$ and $f_r(x)$ are strictly increasing. We note the inverse functions of $f_l, f_r$ as $g_l, g_r$, respectively. Combining Lemma B.1c, it is not hard to verify that $g_l, g_r$ are

$$g_l(\varepsilon) = f_l^{-1}(\varepsilon) = 1 - w_1(\varepsilon) = 1 + W_0(-e^{-(1+\varepsilon)}) \ (\varepsilon \ge 0) \tag{G.97}$$

$$g_r(\varepsilon) = f_r^{-1}(\varepsilon) = w_2(\varepsilon) - 1 = -W_{-1}(-e^{-(1+\varepsilon)}) - 1 \ (\varepsilon \ge 0) \tag{G.98}$$

According to Lemma B.1e, the derivatives of $g_l, g_r$ are

$$g_l'(\varepsilon) = f_l^{-1'}(\varepsilon) = \frac{w_1(\varepsilon)}{1 - w_1(\varepsilon)} = \frac{1 - f_l^{-1}(\varepsilon)}{f_l^{-1}(\varepsilon)} = \frac{1}{1 - w_1(\varepsilon)} - 1 \tag{G.99}$$

$$g_r'(\varepsilon) = f_r^{-1'}(\varepsilon) = \frac{w_2(\varepsilon)}{w_2(\varepsilon) - 1} = \frac{f_r^{-1}(\varepsilon) + 1}{f_r^{-1}(\varepsilon)} = 1 + \frac{1}{w_2(\varepsilon) - 1} \tag{G.100}$$

Specially, since $\lim\limits_{\varepsilon \to 0} w_2(\varepsilon) = w_2(0) = 1$, it is easy to know

$$\lim_{\varepsilon \to 0} g_r'(\varepsilon) = +\infty \tag{G.101}$$

In the following, we note $g_r'(0) = +\infty$ for convenience.

Lemma G.6 gives two useful conclusions for subsequent analysis. They hold apparently.

**Lemma G.6** *Let $a, b, a^+, b^-$ be positive real numbers.*

   *(a) if $a > b, a < a^+, b > b^-$, then $\frac{a+1}{b+1} < \frac{a^++1}{b^-+1}$.*
   *(b) if $a \le b$, then $\frac{a(b+1)}{b(a+1)} \le 1$.*

**Lemma G.7** *Given $f(x) = x - \log x$ and $\varepsilon \ge 0$, then $w_2(\varepsilon) - 1 \ge 1 - w_1(\varepsilon)$ where the inequality is tight when $\varepsilon = 0$;*

**Proof 10** *With the helper functions $f_l, f_r$ (Equation (G.95)), we define function*

$$\Delta_w(\varepsilon) = g_r(\varepsilon) - g_l(\varepsilon) = (w_2(\varepsilon) - 1) - (1 - w_1(\varepsilon)) \tag{G.102}$$

*It is straightforward to know*

$$\Delta_w(0) = w_2(0) - 1 - (1 - w_1(0)) = 0 \tag{G.103}$$

*In the following, we prove $\Delta_w'(\varepsilon) > 0$ for $\varepsilon > 0$. Plugging Equation (G.96), we have*

$$g_r'(\varepsilon) = f_r^{-1'}(\varepsilon) = \frac{1}{f_r'(f_r^{-1}(\varepsilon))} = \frac{1}{f_r'(w_2(\varepsilon) - 1)} = \frac{1}{1 - \frac{1}{w_2(\varepsilon)}} = \frac{1}{f'(w_2(\varepsilon))} \tag{G.104}$$

$$g_l'(\varepsilon) = f_l^{-1'}(\varepsilon) = \frac{1}{f_l'(f_l^{-1}(\varepsilon))} = \frac{1}{f_l'(1 - w_1(\varepsilon))} = \frac{1}{\frac{1}{w_1(\varepsilon)} - 1} = \frac{1}{-f'(w_1(\varepsilon))} \tag{G.105}$$

*According to Lemma B.1 and Lemma B.1f, $f(x)$ is strictly decreasing in $(0, 1)$ and $f(w_1(\varepsilon)) > f(\frac{1}{w_2(\varepsilon)})$. So we can know $w_1(\varepsilon) < \frac{1}{w_2(\varepsilon)}$. Since $f(x)$ is convex and $f'(x) < 0$ in $(0, 1)$, we*

*can know* $f'(w_1(\varepsilon)) < f'(\frac{1}{w_2(\varepsilon)})$. *Now combining Lemma B.1i, we can obtain* $0 < f'(w_2(\varepsilon)) \leq -f'(\frac{1}{w_2(\varepsilon)}) < -f'(w_1(\varepsilon))$ $(\varepsilon > 0)$. *This leads to*

$$g'_r(\varepsilon) = \frac{1}{f'(w_2(\varepsilon))} > \frac{1}{-f'(w_1(\varepsilon))} = g'_l(\varepsilon) \tag{G.106}$$

*for* $\varepsilon > 0$, *which means* $\Delta_w'(\varepsilon) = g'_r(\varepsilon) - g'_l(\varepsilon) > 0$ $(\varepsilon > 0)$. *Now combining Equation (G.103), we can conclude*

$$\Delta_w(\varepsilon) = g_r(\varepsilon) - g_l(\varepsilon) = (w_2(\varepsilon) - 1) - (1 - w_1(\varepsilon)) \geq 0 \tag{G.107}$$

□

**Lemma G.8** *Given* $f(x) = x - \log x$ *and* $\varepsilon_x, \varepsilon_y \geq 0$, *if* $f(x) \leq 1 + \varepsilon_x$ *and* $f(y) \leq 1 + \varepsilon_y$, *then*

$$f(xy) \leq f(w_2(\varepsilon_x)w_2(\varepsilon_y)) \tag{G.108}$$

**Proof 11** *From Lemma B.1g, we know* $w_1(\varepsilon_x) \leq x \leq w_2(\varepsilon_x)$ *and* $w_1(\varepsilon_y) \leq y \leq w_2(\varepsilon_y)$. *So we have* $w_1(\varepsilon_x)w_1(\varepsilon_x) \leq xy \leq w_2(\varepsilon_x)w_2(\varepsilon_y)$. *According to Lemma B.1a, it suffices to show* $f(w_1(\varepsilon_x)w_1(\varepsilon_y)) \leq f(w_2(\varepsilon_x)w_2(\varepsilon_y))$. *By the definition of* $f(x)$, *we have*

$$
\begin{aligned}
&f(w_2(\varepsilon_x)w_2(\varepsilon_y)) - f(w_1(\varepsilon_x)w_1(\varepsilon_y)) \\
=&w_2(\varepsilon_x)w_2(\varepsilon_y) - \log(w_2(\varepsilon_x)w_2(\varepsilon_y)) - (w_1(\varepsilon_x)w_1(\varepsilon_y) - \log(w_1(\varepsilon_x)w_1(\varepsilon_y))) \\
=&w_2(\varepsilon_x)w_2(\varepsilon_y) - \log w_2(\varepsilon_x) - \log w_2(\varepsilon_y) - (w_1(\varepsilon_x)w_1(\varepsilon_y) - \log w_1(\varepsilon_x) - \log w_1(\varepsilon_y)) \\
=&w_2(\varepsilon_x)w_2(\varepsilon_y) - w_2(\varepsilon_x) + w_2(\varepsilon_x) - \log w_2(\varepsilon_x) - w_2(\varepsilon_y) + w_2(\varepsilon_y) - \log w_2(\varepsilon_y) \\
&- (w_1(\varepsilon_x)w_1(\varepsilon_y) - w_1(\varepsilon_x) + w_1(\varepsilon_x) - \log w_1(\varepsilon_x) - w_1(\varepsilon_y) + w_1(\varepsilon_y) - \log w_1(\varepsilon_y)) \\
=&w_2(\varepsilon_x)w_2(\varepsilon_y) - w_2(\varepsilon_x) + \varepsilon_x - w_2(\varepsilon_y) + \varepsilon_y - (w_1(\varepsilon_x)w_1(\varepsilon_y) - w_1(\varepsilon_x) + \varepsilon_x - w_1(\varepsilon_y) + \varepsilon_y) \\
=&w_2(\varepsilon_x)w_2(\varepsilon_y) - w_2(\varepsilon_x) - w_2(\varepsilon_y) - (w_1(\varepsilon_x)w_1(\varepsilon_y) - w_1(\varepsilon_x) - w_1(\varepsilon_y)) \\
=&w_2(\varepsilon_x)w_2(\varepsilon_y) - w_2(\varepsilon_x) - w_2(\varepsilon_y) + 1 - (w_1(\varepsilon_x)w_1(\varepsilon_y) - w_1(\varepsilon_x) - w_1(\varepsilon_y) + 1) \\
=&(w_2(\varepsilon_x) - 1)(w_2(\varepsilon_y) - 1) - (w_1(\varepsilon_x) - 1)(w_1(\varepsilon_y) - 1)
\end{aligned}
\tag{G.109}
$$

*From Lemma G.7, it is easy to know* $w_2(\varepsilon_x) - 1 \geq 1 - w_1(\varepsilon_x)$ *and* $w_2(\varepsilon_y) - 1 \geq 1 - w_1(\varepsilon_y)$. *Thus we can conclude*

$$f(w_2(\varepsilon_x)w_2(\varepsilon_y)) - f(w_1(\varepsilon_x)w_1(\varepsilon_y)) \geq 0 \tag{G.110}$$

□

## G.2 The Main Proof of Key Lemma G.5

**Proof 12** *Inequality (G.94) is equal to*

$$
\begin{aligned}
&f(w_2(\varepsilon_{x,1})w_2(\varepsilon_{y,1})) + f(w_2(\varepsilon_{x,2})w_2(\varepsilon_{y,2})) \\
\leq&f(w_2(\varepsilon_{x,1} + \varepsilon_{x,2})w_2(\varepsilon_{y,1} + \varepsilon_{y,2})) + f(w_2(0)w_2(0))
\end{aligned}
\tag{G.111}
$$

*by Lemma B.1d.*

*We define function*

$$S(\theta_x, \theta_y) = f(w_2(\varepsilon_{x,1} + \theta_x\varepsilon_{x,2})w_2(\varepsilon_{y,1} + \theta_y\varepsilon_{y,2})) + f(w_2(\varepsilon_{x,2} - \theta_x\varepsilon_{x,2})w_2(\varepsilon_{y,2} - \theta_y\varepsilon_{y,2})) \tag{G.112}$$

*for* $-\frac{\varepsilon_{x,1}}{\varepsilon_{x,2}} \leq \theta_x \leq 1, -\frac{\varepsilon_{y,1}}{\varepsilon_{y,2}} \leq \theta_y \leq 1$. *Here the domains of* $\theta_x, \theta_y$ *are restricted to make* $w_2(\cdot)$ *in the definition of* $S(\theta_x, \theta_y)$ *meaningful. Inequation (G.94) states that* $S(0,0) \leq S(1,1)$.

*If we note* $\varepsilon_x = \varepsilon_{x,1} + \varepsilon_{x,2}$ *and* $\varepsilon_y = \varepsilon_{y,1} + \varepsilon_{y,2}$, $\theta_x, \theta_y$ *control how* $\varepsilon_x$ *and* $\varepsilon_y$ *are allocated between two dimensions, respectively.*

*We can prove* $S(0,0) \leq S(1,1)$ *in the following three cases.*

> **Case 1** $\varepsilon_{x,2} = \varepsilon_{y,2} = 0$.
> **Case 2** $\varepsilon_{x,2} > 0, \varepsilon_{y,2} > 0$. *In this case, we have* $\varepsilon_{x,1} \geq \varepsilon_{x,2} > 0, \varepsilon_{y,1} \geq \varepsilon_{y,2} > 0$.

**Case 3**  only one of $\varepsilon_{x,2}$ and $\varepsilon_{y,2}$ equals to 0.

It is easy to verify that $S(0,0) = S(1,1)$ for **Case 1**. In the following, we discuss **Case 2** first and deal with **Case 3** at the end of the proof.

**Case 2**:

In $S(\theta_x, \theta_y)$, $\theta_x, \theta_y$ are symmetric. Without loss of generality, we choose any $0 < \theta_{x,0} < 1$ at the beginning. The following proof consists of two steps. In **Step 1**, we prove that for any fixed $0 < \theta_{x,0} < 1$, there exists one and only one $-\frac{\varepsilon_{y,1}}{\varepsilon_{y,2}} < \theta_{y,1} < 1$ such that $S(\theta_{x,0}, \theta_y)$ takes its maximum. This accomplishes aspects **A1** and **A2** in the first iteration. In **Step 2**, we prove $S(1,1) \geq S(0,0)$. The key idea is finding a strictly increasing sequence $\{S[i]\}$ such that $S[0]$ can be arbitrarily close to $S(0,0)$ and $\lim_{i \to \infty} S[i] = S(1,1)$. **Step 2** will accomplish aspects **A1** ~ **A4** in all iterations.

**Step 1.** At the beginning, we select any [2] $0 < \theta_{x,0} < 1$. For brevity, we note

$$\tilde{\varepsilon}_{x,1}[0] = \varepsilon_{x,1} + \theta_{x,0}\varepsilon_{x,2}, \ \tilde{\varepsilon}_{x,2}[0] = \varepsilon_{x,2} - \theta_{x,0}\varepsilon_{x,2} \tag{G.113}$$

$$\tilde{\varepsilon}_{y,1} = \varepsilon_{y,1} + \theta_y\varepsilon_{y,2}, \ \tilde{\varepsilon}_{y,2} = \varepsilon_{y,2} - \theta_y\varepsilon_{y,2} \tag{G.114}$$

where we use $\tilde{\varepsilon}_{x,(\cdot)}[0]$ to denote the variable is computed with $\theta_{x,0}$.

Note that $g_r(\varepsilon)$ (defined in Equation (G.98)) is strictly increasing with $\varepsilon$. Combining the precondition $\varepsilon_{x,1} \geq \varepsilon_{x,2}$, we can know

$$\frac{g_r(\tilde{\varepsilon}_{x,1}[0])}{g_r(\tilde{\varepsilon}_{x,2}[0])} = \frac{g_r(\varepsilon_{x,1} + \theta_{x,0}\varepsilon_{x,2})}{g_r(\varepsilon_{x,2} - \theta_{x,0}\varepsilon_{x,2})} > \frac{g_r(\varepsilon_{x,1})}{g_r(\varepsilon_{x,2})} \geq 1 \tag{G.115}$$

We note this condition as $C_1[0]$ as follows.

$$C_1[0] : \frac{g_r(\tilde{\varepsilon}_{x,1}[0])}{g_r(\tilde{\varepsilon}_{x,2}[0])} > 1 \tag{G.116}$$

Now given the fixed $\theta_{x,0}$, the derivative of $S(\theta_{x,0}, \theta_y)$ is

$$\frac{dS(\theta_{x,0}, \theta_y)}{d\theta_y}$$
$$= \varepsilon_{y,2}\left(w_2(\tilde{\varepsilon}_{x,1}[0])\frac{w_2(\tilde{\varepsilon}_{y,1})}{w_2(\tilde{\varepsilon}_{y,1}) - 1} - \frac{1}{w_2(\tilde{\varepsilon}_{y,1})}\frac{w_2(\tilde{\varepsilon}_{y,1})}{w_2(\tilde{\varepsilon}_{y,1}) - 1}\right)$$
$$\quad - \varepsilon_{y,2}\left(w_2(\tilde{\varepsilon}_{x,2}[0])\frac{w_2(\tilde{\varepsilon}_{y,2})}{w_2(\tilde{\varepsilon}_{y,2}) - 1} - \frac{1}{w_2(\tilde{\varepsilon}_{y,2})}\frac{w_2(\tilde{\varepsilon}_{y,2})}{w_2(\tilde{\varepsilon}_{y,2}) - 1}\right)$$
$$= \varepsilon_{y,2}\left(\frac{w_2(\tilde{\varepsilon}_{x,1}[0])w_2(\tilde{\varepsilon}_{y,1}) - 1}{w_2(\tilde{\varepsilon}_{y,1}) - 1} - \frac{w_2(\tilde{\varepsilon}_{x,2}[0])w_2(\tilde{\varepsilon}_{y,2}) - 1}{w_2(\tilde{\varepsilon}_{y,2}) - 1}\right)$$
$$= \varepsilon_{y,2}\left(\frac{w_2(\tilde{\varepsilon}_{x,1}[0])w_2(\tilde{\varepsilon}_{y,1}) - w_2(\tilde{\varepsilon}_{x,1}[0]) + w_2(\tilde{\varepsilon}_{x,1}[0]) - 1}{w_2(\tilde{\varepsilon}_{y,1}) - 1}\right.$$
$$\left.- \frac{w_2(\tilde{\varepsilon}_{x,2}[0])w_2(\tilde{\varepsilon}_{y,2}) - w_2(\tilde{\varepsilon}_{x,2}[0]) + w_2(\tilde{\varepsilon}_{x,2}[0]) - 1}{w_2(\tilde{\varepsilon}_{y,2}) - 1}\right)$$
$$= \varepsilon_{y,2}\left(\left(w_2(\tilde{\varepsilon}_{x,1}[0]) + \frac{w_2(\tilde{\varepsilon}_{x,1}[0]) - 1}{w_2(\tilde{\varepsilon}_{y,1}) - 1}\right) - \left(w_2(\tilde{\varepsilon}_{x,2}[0]) + \frac{w_2(\tilde{\varepsilon}_{x,2}[0]) - 1}{w_2(\tilde{\varepsilon}_{y,2}) - 1}\right)\right) \tag{G.117}$$

The second order derivative is

$$\frac{d^2 S(\theta_{x,0}, \theta_y)}{d\theta_y^2}$$
$$= \frac{-(w_2(\tilde{\varepsilon}_{x,1}[0]) - 1)}{(w_2(\tilde{\varepsilon}_{y,1}) - 1)^2}\frac{w_2(\tilde{\varepsilon}_{y,1})}{w_2(\tilde{\varepsilon}_{y,1}) - 1}(\varepsilon_{y,2})^2 - \frac{-(w_2(\tilde{\varepsilon}_{x,2}[0]) - 1)}{(w_2(\tilde{\varepsilon}_{y,2}) - 1)^2}\frac{w_2(\tilde{\varepsilon}_{y,2})}{w_2(\tilde{\varepsilon}_{y,2}) - 1}(-(\varepsilon_{y,2})^2)$$
$$= -\frac{(w_2(\tilde{\varepsilon}_{x,1}[0]) - 1)w_2(\tilde{\varepsilon}_{y,1})(\varepsilon_{y,2})^2}{(w_2(\tilde{\varepsilon}_{y,1}) - 1)^3} - \frac{(w_2(\tilde{\varepsilon}_{x,2}[0]) - 1)w_2(\tilde{\varepsilon}_{y,2})(\varepsilon_{y,2})^2}{(w_2(\tilde{\varepsilon}_{y,2}) - 1)^3} \tag{G.118}$$

---

[2] We will require that $\theta_{x,0}$ can be arbitrarily close to 0 later.

*Since $w_2(\varepsilon) > 1$ for $\varepsilon > 0$, it is easy to know $\frac{\mathrm{d}^2 S(\theta_{x,0},\theta_y)}{\mathrm{d}\theta_y^2} < 0$ for $\theta_y < 1$. Thus we get the following proposition.*

**Proposition G.2** $S(\theta_{x,0}, \theta_y)$ *is strictly concave and has at most one maximum for $\theta_y < 1$.*

*Remember that we are discussing **Case 2**, so $\varepsilon_{y,2} > 0$. Now letting $\frac{\mathrm{d}S(\theta_{x,0},\theta_y)}{\mathrm{d}\theta_y} = 0$ (i.e., Equation (G.117)= 0), we can obtain*

$$\frac{\mathrm{d}S(\theta_{x,0}, \theta_y)}{\mathrm{d}\theta_y} = 0 \Leftrightarrow w_2(\tilde{\varepsilon}_{x,1}[0]) + \frac{w_2(\tilde{\varepsilon}_{x,1}[0]) - 1}{w_2(\tilde{\varepsilon}_{y,1}) - 1} = w_2(\tilde{\varepsilon}_{x,2}[0]) + \frac{w_2(\tilde{\varepsilon}_{x,2}[0]) - 1}{w_2(\tilde{\varepsilon}_{y,2}) - 1}$$

(G.119)

*Now, it is hard to continue the analysis because we can not solve Equation (G.119) analytically. However, we succeed to go further by analyzing Equation (G.119). Our analysis starts from the following transformation in Equations (G.120) ~ (G.124), which is hard to obtain but not hard to verify.*

*Using the notations of helper functions $g_r(\varepsilon) = f_r^{-1}(\varepsilon), g_r'(\varepsilon) = f_r^{-1\prime}(\varepsilon)$ in Equations (G.98) and (G.100), we can rewrite Equation (G.119) as follows.*

Equation (G.119)

$$\Leftrightarrow w_2(\tilde{\varepsilon}_{x,1}[0]) - 1 + \frac{w_2(\tilde{\varepsilon}_{x,1}[0]) - 1}{w_2(\tilde{\varepsilon}_{y,1}) - 1} = w_2(\tilde{\varepsilon}_{x,2}[0]) - 1 + \frac{w_2(\tilde{\varepsilon}_{x,2}[0]) - 1}{w_2(\tilde{\varepsilon}_{y,2}) - 1}$$

(G.120)

$$\Leftrightarrow (w_2(\tilde{\varepsilon}_{x,1}[0]) - 1)\left(1 + \frac{1}{w_2(\tilde{\varepsilon}_{y,1}) - 1}\right) = (w_2(\tilde{\varepsilon}_{x,2}[0]) - 1)\left(1 + \frac{1}{w_2(\tilde{\varepsilon}_{y,2}) - 1}\right)$$

$$\Leftrightarrow g_r(\tilde{\varepsilon}_{x,1}[0])g_r'(\tilde{\varepsilon}_{y,1}) = g_r(\tilde{\varepsilon}_{x,2}[0])g_r'(\tilde{\varepsilon}_{y,2})$$

$$\Leftrightarrow \frac{g_r(\tilde{\varepsilon}_{x,1}[0])}{g_r(\tilde{\varepsilon}_{x,2}[0])} = \frac{g_r'(\tilde{\varepsilon}_{y,2})}{g_r'(\tilde{\varepsilon}_{y,1})}$$

(G.121)

$$\Leftrightarrow \frac{g_r(\tilde{\varepsilon}_{x,1}[0])}{g_r(\tilde{\varepsilon}_{x,2}[0])} = \frac{\left(\frac{1}{g_r'(\tilde{\varepsilon}_{y,1})}\right)}{\left(\frac{1}{g_r'(\tilde{\varepsilon}_{y,2})}\right)}$$

$$\Leftrightarrow \frac{g_r(\tilde{\varepsilon}_{x,1}[0])}{g_r(\tilde{\varepsilon}_{x,2}[0])} = \frac{\left(\frac{g_r(\tilde{\varepsilon}_{y,1})}{g_r(\tilde{\varepsilon}_{y,1}) + 1}\right)}{\left(\frac{g_r(\tilde{\varepsilon}_{y,2})}{g_r(\tilde{\varepsilon}_{y,2}) + 1}\right)}$$

(G.122)

$$\Leftrightarrow \frac{g_r(\tilde{\varepsilon}_{x,1}[0])}{g_r(\tilde{\varepsilon}_{x,2}[0])} = \frac{g_r(\tilde{\varepsilon}_{y,1})}{g_r(\tilde{\varepsilon}_{y,2})}\frac{g_r(\tilde{\varepsilon}_{y,2}) + 1}{g_r(\tilde{\varepsilon}_{y,1}) + 1}$$

(G.123)

$$\Leftrightarrow \frac{g_r(\tilde{\varepsilon}_{y,1})}{g_r(\tilde{\varepsilon}_{y,2})} = \frac{g_r(\tilde{\varepsilon}_{x,1}[0])}{g_r(\tilde{\varepsilon}_{x,2}[0])}\frac{g_r(\tilde{\varepsilon}_{y,1}) + 1}{g_r(\tilde{\varepsilon}_{y,2}) + 1}$$

(G.124)

*where Equation (G.122) follows from Equation (G.100).*

*Up to now, we transform the condition $\frac{\mathrm{d}S(\theta_{x,0},\theta_y)}{\mathrm{d}\theta_y} = 0$ in Equation (G.119) to Equation (G.124). In the following, Equation (G.124) will be used to investigate the property of the maximum for $S(\theta_{x,0}, \theta_y)$. The goal is to accomplish aspect **A2** of the proof.*

*In the following **Substep 1.1**, we show that Equation (G.119) must have one and only one solution. In other words, there must be one and only one point making $\frac{\mathrm{d}S(\theta_{x,0},\theta_y)}{\mathrm{d}\theta_y} = 0$. Unfortunately, it is hard to get an analytical solution from Equation (G.119) due to the complexity brought by Lambert W function. Therefore, in **Substep 1.2**, we analyze Equations (G.120) ~ (G.124) to investigate the properties of the solution. Overall, the analysis in **Step 1** will be used as a basic step in **Step 2**.*

**Substep 1.1.** *According to the definition of $g_r'(\varepsilon)$ in Equation (G.100), $g_r'(\varepsilon)$ is strictly decreasing with $\varepsilon$. So $g_r'(\tilde{\varepsilon}_{y,2}) = g_r'(\varepsilon_{y,2} - \theta_y \varepsilon_{y,2})$ is strictly increasing and $g_r'(\tilde{\varepsilon}_{y,1}) = g_r'(\varepsilon_{y,1} + \theta_y \varepsilon_{y,2})$ is*

*strictly decreasing with $\theta_y$. Thus, the RHS of Equation (G.121) is continuous and strictly increasing with $\theta_y$. Besides, according to Equation (G.101) and the definition of $\tilde{\varepsilon}_{y,1}, \tilde{\varepsilon}_{y,2}$ in Equation (G.113), it is easy to know*

$$\lim_{\theta_y \to -\frac{\varepsilon_{y,1}}{\varepsilon_{y,2}}} \frac{g_r'(\tilde{\varepsilon}_{y,2})}{g_r'(\tilde{\varepsilon}_{y,1})} = \lim_{\theta_y \to -\frac{\varepsilon_{y,1}}{\varepsilon_{y,2}}} \frac{g_r'(\varepsilon_{y,2} - \theta_y \varepsilon_{y,2})}{g_r'(\varepsilon_{y,1} + \theta_y \varepsilon_{y,2})} = \frac{g_r'(\varepsilon_{y,2} + \varepsilon_{y,1})}{g_r'(0)} = \frac{g_r'(\varepsilon_{y,2} + \varepsilon_{y,1})}{+\infty} = 0$$

(G.125)

$$\lim_{\theta_y \to 1} \frac{g_r'(\tilde{\varepsilon}_{y,2})}{g_r'(\tilde{\varepsilon}_{y,1})} = \lim_{\theta_y \to 1} \frac{g_r'(\varepsilon_{y,2} - \theta_y \varepsilon_{y,2})}{g_r'(\varepsilon_{y,1} + \theta_y \varepsilon_{y,2})} = \frac{g_r'(0)}{g_r'(\varepsilon_{y,1} + \varepsilon_{y,2})} = \frac{+\infty}{g_r'(\varepsilon_{y,1} + \varepsilon_{y,2})} = +\infty \quad \text{(G.126)}$$

*So the range of the RHS of Equation (G.121) is $(0, +\infty)$ when $-\frac{\varepsilon_{y,1}}{\varepsilon_{y,2}} < \theta_y < 1$.*

*Remember that we start from $0 < \theta_{x,0} < 1$, combining the precondition $\varepsilon_{x,1} \geq \varepsilon_{x,2}$ and the definitions of $\tilde{\varepsilon}_{x,1}[0], \tilde{\varepsilon}_{x,2}[0]$ in Equation (G.113) and (G.113), it is easy to know that the LHS of Equation (G.121) is a positive constant number. Therefore, Equation (G.121) has one and only one solution. We note such solution as $\theta_{y,1}$. Combining with Proposition G.2, we can know that for any fixed $0 < \theta_{x,0} < 1$, there exists one and only one $-\frac{\varepsilon_{y,1}}{\varepsilon_{y,2}} < \theta_{y,1} < 1$ that maximize $S(\theta_{x,0}, \theta_y)$.*

*Here note that we still have no guarantee for $\theta_{y,1} > 0$ currently.*

**Substep 1.2.** *We can investigate the property of the solution $\theta_{y,1}$ by analyzing Equation (G.121), (G.123) and (G.124). Now we note*

$$\tilde{\varepsilon}_{y,1}[1] = \varepsilon_{y,1} + \theta_{y,1} \varepsilon_{y,2}, \ \tilde{\varepsilon}_{y,2}[1] = \varepsilon_{y,2} - \theta_{y,1} \varepsilon_{y,2}$$

*as like Equation (G.113).*

*Firstly, we can show*

$$\frac{g_r(\tilde{\varepsilon}_{y,1}[1])}{g_r(\tilde{\varepsilon}_{y,2}[1])} = \frac{g_r(\varepsilon_{y,1} + \theta_{y,1} \varepsilon_{y,2})}{g_r(\varepsilon_{y,2} - \theta_{y,1} \varepsilon_{y,2})} > 1$$

(G.127)

*by contradiction.*

*Assume to the contrary that $\frac{g_r(\tilde{\varepsilon}_{y,1}[1])}{g_r(\tilde{\varepsilon}_{y,2}[1])} \leq 1$, then $\frac{g_r(\tilde{\varepsilon}_{y,1}[1])+1}{g_r(\tilde{\varepsilon}_{y,2}[1])+1} \geq \frac{g_r(\tilde{\varepsilon}_{y,1}[1])}{g_r(\tilde{\varepsilon}_{y,2}[1])}$. Combining condition $C_1[0]$ in Equation (G.116), we can deduce*

$$\frac{g_r(\tilde{\varepsilon}_{x,1}[0])}{g_r(\tilde{\varepsilon}_{x,2}[0])} \frac{g_r(\tilde{\varepsilon}_{y,1}[1]) + 1}{g_r(\tilde{\varepsilon}_{y,2}[1]) + 1} > \frac{g_r(\tilde{\varepsilon}_{y,1}[1]) + 1}{g_r(\tilde{\varepsilon}_{y,2}[1]) + 1} \geq \frac{g_r(\tilde{\varepsilon}_{y,1}[1])}{g_r(\tilde{\varepsilon}_{y,2}[1])}$$

(G.128)

*This contradicts with Equation (G.124). Therefore, we can obtain the following condition $C_1[1]$.*

$$C_1[1] : \frac{g_r(\tilde{\varepsilon}_{y,1}[1])}{g_r(\tilde{\varepsilon}_{y,2}[1])} > 1$$

(G.129)

*Secondly, according to condition $C_1[1]$, it is easy to know*

$$\frac{g_r(\tilde{\varepsilon}_{y,1}[1])}{g_r(\tilde{\varepsilon}_{y,2}[1])} > \frac{g_r(\tilde{\varepsilon}_{y,1}[1]) + 1}{g_r(\tilde{\varepsilon}_{y,2}[1]) + 1} > 1$$

(G.130)

*Now combining Equation (G.124) and (G.130), we can know the following condition[3] $C_2[1]$ holds.*

$$C_2[1] : \frac{g_r(\tilde{\varepsilon}_{y,1}[1])}{g_r(\tilde{\varepsilon}_{y,2}[1])} > \frac{g_r(\tilde{\varepsilon}_{x,1}[0])}{g_r(\tilde{\varepsilon}_{x,2}[0])}$$

(G.131)

*In summary, we start from any fixed $0 < \theta_{x,0} < 1$ making condition $C_1[0]$ in Equation (G.116) hold. Using **Step 1**, we find the only one $-\frac{\varepsilon_{y,1}}{\varepsilon_{y,2}} < \theta_{y,1} < 1$ such that $S(\theta_{x,0}, \theta_y)$ takes its maximum at $\theta_{y,1}$ and conditions $C_1[1]$ and $C_2[1]$ hold.*

**Step 2.** *The deduction in **Step 1** can be iterated repeatedly due to the symmetry of $\theta_x$ and $\theta_y$ in $S(\theta_x, \theta_y)$. For consistency, at the begining of the iterations, we can choose any $\theta_y \neq \theta_{y,1}$ as $\theta_{y,0}$.*

---

[3]In this context, $C_2[1]$ is stronger than $C_1[1]$. We separate $C_1[1]$ and $C_2[1]$ away for clarity.

*In the following, we use notations*

$$\tilde{\varepsilon}_{x,1}[i] = \varepsilon_{x,1} + \theta_{x,i}\varepsilon_{x,2}, \ \tilde{\varepsilon}_{x,2}[i] = \varepsilon_{x,2} - \theta_{x,i}\varepsilon_{x,2}$$
$$\tilde{\varepsilon}_{y,1}[i] = \varepsilon_{y,1} + \theta_{y,i}\varepsilon_{y,2}, \ \tilde{\varepsilon}_{y,2}[i] = \varepsilon_{y,2} - \theta_{y,i}\varepsilon_{y,2} \tag{G.132}$$

*where $\tilde{\varepsilon}_{(\cdot),k}[i]$ ($k \in \{1,2\}$) is computed with $\theta_{(\cdot),i}$. Now we fix $\theta_{y,1}$ and make $\theta_x$ vary, then we repeat **Step 1** on $\theta_x$. Note that, in the second iteration the condition $C_1[1]$ plays the same role as condition $C_1[0]$ plays in the first iteration. Therefore, we can find a $\theta_{x,2}$ such that the following conditions hold*

$$S(\theta_{x,2}, \theta_{y,1}) > S(\theta_{x,0}, \theta_{y,1}) \tag{G.133}$$

$$\frac{g_r(\tilde{\varepsilon}_{x,1}[2])}{g_r(\tilde{\varepsilon}_{x,2}[2])} = \frac{g_r(\tilde{\varepsilon}_{y,1}[1])}{g_r(\tilde{\varepsilon}_{y,2}[1])}\frac{g_r(\tilde{\varepsilon}_{x,1}[2]) + 1}{g_r(\tilde{\varepsilon}_{x,2}[2]) + 1} = \frac{g_r(\tilde{\varepsilon}_{x,1}[0])}{g_r(\tilde{\varepsilon}_{x,2}[0])}\frac{g_r(\tilde{\varepsilon}_{y,1}[1]) + 1}{g_r(\tilde{\varepsilon}_{y,2}[1]) + 1}\frac{g_r(\tilde{\varepsilon}_{x,1}[2]) + 1}{g_r(\tilde{\varepsilon}_{x,2}[2]) + 1} \tag{G.134}$$

$$C_1[2] : \frac{g_r(\tilde{\varepsilon}_{x,1}[2])}{g_r(\tilde{\varepsilon}_{x,2}[2])} > 1 \tag{G.135}$$

$$C_2[2] : \frac{g_r(\tilde{\varepsilon}_{x,1}[2])}{g_r(\tilde{\varepsilon}_{x,2}[2])} > \frac{g_r(\tilde{\varepsilon}_{y,1}[1])}{g_r(\tilde{\varepsilon}_{y,2}[1])} \tag{G.136}$$

*where the first equation is by Equation (G.124). Note that, combining conditions $C_1[0]$, $C_1[1]$, $C_2[1]$, $C_1[2]$ and Equation (G.134), we know it is impossible that $\tilde{\varepsilon}_{x,1}[2] = \tilde{\varepsilon}_{x,1}[0]$ and $\theta_{x,2} = \theta_{x,0}$. So it is impossible $S(\theta_{x,2}, \theta_{y,1}) = S(\theta_{x,0}, \theta_{y,1})$.*

*We can repeat **Step 1** on $\theta_x$ and $\theta_y$ alternatively and construct a sequence $\{\theta_{x,0}, \theta_{y,1}, \theta_{x,2}, \theta_{y,3}, \dots\}$ such that the following conditions hold.*

*For $i \in \mathbb{N}$ we have*

$$S(\theta_{x,2i+2}, \theta_{y,2i+1}) > S(\theta_{x,2i}, \theta_{y,2i+1}) \tag{G.137}$$

$$\frac{g_r(\tilde{\varepsilon}_{x,1}[2i+2])}{g_r(\tilde{\varepsilon}_{x,2}[2i+2])} = \frac{g_r(\tilde{\varepsilon}_{y,1}[2i+1])}{g_r(\tilde{\varepsilon}_{y,2}[2i+1])}\frac{g_r(\tilde{\varepsilon}_{x,1}[2i+2]) + 1}{g_r(\tilde{\varepsilon}_{x,2}[2i+2]) + 1}$$
$$= \frac{g_r(\tilde{\varepsilon}_{x,1}[2i])}{g_r(\tilde{\varepsilon}_{x,2}[2i])}\frac{g_r(\tilde{\varepsilon}_{y,1}[2i+1]) + 1}{g_r(\tilde{\varepsilon}_{y,2}[2i+1]) + 1}\frac{g_r(\tilde{\varepsilon}_{x,1}[2i+2]) + 1}{g_r(\tilde{\varepsilon}_{x,2}[2i+2]) + 1} \tag{G.138}$$

$$C_1[2i+2] : \frac{g_r(\tilde{\varepsilon}_{x,1}[2i+2])}{g_r(\tilde{\varepsilon}_{x,2}[2i+2])} > 1 \tag{G.139}$$

$$C_2[2i+2] : \frac{g_r(\tilde{\varepsilon}_{x,1}[2i+2])}{g_r(\tilde{\varepsilon}_{x,2}[2i+2])} > \frac{g_r(\tilde{\varepsilon}_{y,1}[2i+1])}{g_r(\tilde{\varepsilon}_{y,2}[2i+1])} \tag{G.140}$$

*For $i \in \mathbb{N} \wedge i > 0$ we have*

$$S(\theta_{x,2i}, \theta_{y,2i+1}) > S(\theta_{x,2i}, \theta_{y,2i-1}) \tag{G.141}$$

$$\frac{g_r(\tilde{\varepsilon}_{y,1}[2i+1])}{g_r(\tilde{\varepsilon}_{y,2}[2i+1])} = \frac{g_r(\tilde{\varepsilon}_{x,1}[2i])}{g_r(\tilde{\varepsilon}_{x,2}[2i])}\frac{g_r(\tilde{\varepsilon}_{y,1}[2i+1]) + 1}{g_r(\tilde{\varepsilon}_{y,2}[2i+1]) + 1}$$
$$= \frac{g_r(\tilde{\varepsilon}_{y,1}[2i-1])}{g_r(\tilde{\varepsilon}_{y,2}[2i-1])}\frac{g_r(\tilde{\varepsilon}_{x,1}[2i]) + 1}{g_r(\tilde{\varepsilon}_{x,2}[2i]) + 1}\frac{g_r(\tilde{\varepsilon}_{y,1}[2i+1]) + 1}{g_r(\tilde{\varepsilon}_{y,2}[2i+1]) + 1} \tag{G.142}$$

$$C_1[2i+1] : \frac{g_r(\tilde{\varepsilon}_{y,1}[2i+1])}{g_r(\tilde{\varepsilon}_{y,2}[2i+1])} > 1 \tag{G.143}$$

$$C_2[2i+1] : \frac{g_r(\tilde{\varepsilon}_{y,1}[2i+1])}{g_r(\tilde{\varepsilon}_{y,2}[2i+1])} > \frac{g_r(\tilde{\varepsilon}_{x,1}[2i])}{g_r(\tilde{\varepsilon}_{x,2}[2i])} \tag{G.144}$$

*Now combining conditions $C_2[2i+2]$, $C_2[2i+1]$ and Equations (G.132), we can also obtain*

$$\theta_{x,2i+2} > \theta_{x,2i}, \ \theta_{y,2i+3} > \theta_{y,2i+1} \ (i \in \mathbb{N}) \tag{G.145}$$

*Up to now, we have constructed the following strictly increasing sequences*

$$\Theta_x[i] = \theta_{x,2i} \tag{G.146}$$

$$\Theta_y[i] = \theta_{y,2i+1} \tag{G.147}$$

$$R_x[i] = \frac{g_r(\tilde{\varepsilon}_{x,1}[2i])}{g_r(\tilde{\varepsilon}_{x,2}[2i])} \tag{G.148}$$

$$R_y[i] = \frac{g_r(\tilde{\varepsilon}_{y,1}[2i+1])}{g_r(\tilde{\varepsilon}_{y,2}[2i+1])} \tag{G.149}$$

*for $i \in \mathbb{N}$. From inequalities (G.145), we can know $\Theta_x[i]$ and $\Theta_y[i]$ are increasing. From conditions $C_2[2i+2]$ (inequality (G.140)) and $C_2[2i+1]$ (inequality (G.144)), we can know $R_x[i]$ and $R_y[i]$ are increasing. According to conditions $C_1[0], C_1[1], \cdots$, it is easy to know $R_x[i] > 1$, $R_y[i] > 1$ for $i \in \mathbb{N}$. Besides, we note*

$$R_x^+[i] = \frac{g_r(\tilde{\varepsilon}_{x,1}[2i]) + 1}{g_r(\tilde{\varepsilon}_{x,2}[2i]) + 1}, \quad R_y^+[i] = \frac{g_r(\tilde{\varepsilon}_{y,1}[2i+1]) + 1}{g_r(\tilde{\varepsilon}_{y,2}[2i+1]) + 1} \tag{G.150}$$

*for $i \in \mathbb{N}$. According to Lemma G.6a, it is easy to know both $R_x^+[i], R_y^+[i]$ are strictly increasing. Importantly, we have constructed the following strictly increasing sequence for $i \in \mathbb{N}$.*

$$S[i] = \begin{cases} S(\theta_{x,0}, \theta_{y,0}), & i = 0 \\ S(\theta_{x,i-1}, \theta_{y,i}), & i\%2 = 1 \\ S(\theta_{x,i}, \theta_{y,i-1}), & i\%2 = 0 \wedge i > 0 \end{cases} \tag{G.151}$$

*This accomplishes aspect **A3**.*

*Here we note that $C_1[i](i \geq 0)$ plays an important role in each iteration. When $C_1[i]$ holds, we let the derivative of $S$ equal to 0. Then we get the maximum of $S$ and make $C_2[i](i \geq 1)$ hold in each iteration. Importantly, $C_2[i]$ guarantees $R_x[i]$ and $R_y[i]$ are strictly increasing.*

*In the following, we prove*

$$\lim_{i \to +\infty} R_x[i] = +\infty, \quad \lim_{i \to +\infty} R_y[i] = +\infty \tag{G.152}$$

*in order to accomplish aspect **A4** finally. Now let's observe how $R_x[i]$ increases. Using the notations of $R_x[i]$ and $R_x^+[i]$, we rewrite Equation (G.138) and get the following relation*

$$R_x[i+1] = R_x[i] R_y^+[i] R_x^+[i+1] \tag{G.153}$$

*This indicates that*

$$R_x[i+1] - R_x[i] = R_x[i](R_y^+[i] R_x^+[i+1] - 1) \tag{G.154}$$

*Here $R_x[i], R_y^+[i], R_x^+[i]$ are all strictly increasing and larger than 1. The relation in Equation (G.154) indicates that the difference between neighbouring elements of $\{R_x[i]\}$ is strictly increasing. This violates the Cauchy's criterion for convergence. Thus, we can conclude $\lim_{i \to +\infty} R_x[i] = +\infty$. Similarly, we can also conclude $\lim_{i \to +\infty} R_y[i] = +\infty$.*

*Now from $\Theta_x[i] < 1$, we can know*

$$R_x[i] = \frac{g_r(\tilde{\varepsilon}_{x,1}[2i])}{g_r(\tilde{\varepsilon}_{x,2}[2i])} = \frac{w_2(\varepsilon_{x,1} + \theta_{x,2i}\varepsilon_{x,2}) - 1}{w_2(\varepsilon_{x,2} - \theta_{x,2i}\varepsilon_{x,2}) - 1} < \frac{w_2(\varepsilon_{x,1} + \varepsilon_{x,2}) - 1}{w_2(\varepsilon_{x,2} - \theta_{x,2i}\varepsilon_{x,2}) - 1} \tag{G.155}$$

*The numerator of the rightmost item of Equation (G.155) is a constant. From $\lim_{i \to +\infty} R_x[i] = +\infty$, we can conclude $\lim_{i \to +\infty} w_2(\varepsilon_{x,2} - \theta_{x,2i}\varepsilon_{x,2}) - 1 = 0$ and $\lim_{i \to +\infty} \varepsilon_{x,2} - \theta_{x,2i}\varepsilon_{x,2} = 0$. Thus, we obtain*

$$\lim_{i \to +\infty} \Theta_x[i] = \lim_{i \to +\infty} \theta_{x,2i} = 1 \tag{G.156}$$

*Similarly, we can also obtain*

$$\lim_{i \to +\infty} \Theta_y[i] = \lim_{i \to +\infty} \theta_{y,2i+1} = 1 \tag{G.157}$$

*Now combining Equations* (G.112), (G.146), (G.147), (G.151), (G.156) *and* (G.157), *we can know*

$$\lim_{i \to +\infty} S[i]$$
$$= S(1, 1) = f(w_2(\varepsilon_{x,1} + \varepsilon_{x,2})w_2(\varepsilon_{y,1} + \varepsilon_{y,2})) + f(w_2(\varepsilon_{x,2} - \varepsilon_{x,2})w_2(\varepsilon_{y,2} - \varepsilon_{y,2}))$$
$$= f(w_2(\varepsilon_{x,1} + \varepsilon_{x,2})w_2(\varepsilon_{y,1} + \varepsilon_{y,2})) + 1 \tag{G.158}$$

*Since $S[i]$ is strictly increasing, we can conclude*

$$S(\theta_{x,0}, \theta_{y,0}) < f(w_2(\varepsilon_{x,1} + \varepsilon_{x,2})w_2(\varepsilon_{y,1} + \varepsilon_{y,2})) + 1 \tag{G.159}$$

*Remember that, we can take any $\theta_{x,0} > 0$ and any $\theta_{y,0} \neq \theta_{y,1}$ as the start point of above iterations. This means that Equation (G.159) holds for any point $(\theta_{x,0}, \theta_{y,0})$ satisfying $\theta_{x,0} > 0$ in the neighborhood of $(0, 0)$ on the $\theta_x\theta_y$ plane.*

*Now we can show*

$$S(0, 0) \leq f(w_2(\varepsilon_{x,1} + \varepsilon_{x,2})w_2(\varepsilon_{y,1} + \varepsilon_{y,2})) + 1 = S(1, 1) \tag{G.160}$$

*by contradiction. Assume to the contrary that $S(0, 0) > f(w_2(\varepsilon_{x,1} + \varepsilon_{x,2})w_2(\varepsilon_{y,1} + \varepsilon_{y,2})) + 1$, due to continuity of $S(\theta_x, \theta_y)$, we can find a neighbour $(\theta'_x, \theta'_y)$ $(\theta'_x > 0)$ of $(0, 0)$ such that $S(\theta'_x, \theta'_y) > f(w_2(\varepsilon_{x,1} + \varepsilon_{x,2})w_2(\varepsilon_{y,1} + \varepsilon_{y,2})) + 1$. This contradicts with Inequality (G.159).*

***Case 3****:*

*Finally, we can discuss **Case 3** when one of $\varepsilon_{x,2}$ and $\varepsilon_{y,2}$ equals 0. Without loss of generality, we suppose that $\varepsilon_{y,2} = 0$. We can discuss this in two subcases.*

- ***Subcase 3.1****: $\varepsilon_{y,1} = \varepsilon_{y,2} = 0$. By Lemma B.1d and Equation (G.112), it is easy to know*

$$S(1, 1) = f(w_2(\varepsilon_{x,1} + \varepsilon_{x,2})w_2(0)) + f(w_2(0)w_2(0)) = 1 + \varepsilon_{x,1} + \varepsilon_{x,2} + 1$$
$$S(0, 0) = f(w_2(\varepsilon_{x,1})w_2(0)) + f(w_2(\varepsilon_{x,2})w_2(0)) = 1 + \varepsilon_{x,1} + 1 + \varepsilon_{x,2}$$

*This satisfies $S(0, 0) \leq S(1, 1)$.*

- ***Subcase 3.2****: $\varepsilon_{y,1} > \varepsilon_{y,2} = 0$.*

*Here we treat $S(\theta_x, \theta_y)$ as a function of three variables $\theta_x, \theta_y, \varepsilon_{y,2}$ as follows.*

$$S(\theta_x, \theta_y, \varepsilon_{y,2})$$
$$= f(w_2(\varepsilon_{x,1} + \theta_x\varepsilon_{x,2})w_2(\varepsilon_{y,1} + \theta_y\varepsilon_{y,2})) + f(w_2(\varepsilon_{x,2} - \theta_x\varepsilon_{x,2})w_2(\varepsilon_{y,2} - \theta_y\varepsilon_{y,2})) \tag{G.161}$$

*It is easy to know $S(\theta_x, \theta_y, \varepsilon_{y,2})$ is continuous. Note that, we have proven $S(0, 0, \varepsilon_{y,2}) \leq S(1, 1, \varepsilon_{y,2})$ for any $\varepsilon_{y,2} > 0$ in **Case 2**. Therefore, we have*

$$S(0, 0, 0) = \lim_{\varepsilon_{y,2} \to 0} S(0, 0, \varepsilon_{y,2}) \leq \lim_{\varepsilon_{y,2} \to 0} S(1, 1, \varepsilon_{y,2}) = S(1, 1, 0) \tag{G.162}$$

*so $S(0, 0) \leq S(1, 1)$ for $\varepsilon_{y,1} > \varepsilon_{y,2} = 0$.*

*This concludes the proof of Lemma G.5.*

□

# H   Proof of Key Lemma 2

The proof of key Lemma 2 needs the following Lemma H.9.

**Lemma H.9** *(See [10]) For any two Hermition positive semidefinite $n \times n$-matrices $A, B$*

$$\mathrm{Tr}(AB) \leq \sum_{i=1}^{n} \lambda_{A,[i]}\lambda_{B,[i]} \tag{H.163}$$

*where $\lambda_{A,[i]}, \lambda_{B,[i]}$ are the eigenvalues of $A, B$ arranged in decreasing order, respectively.*

**Proof of Lemma 2**

In the following proof of Lemma 2, we will use Equation (H.178) and Lemma H.9 to extend our proof from 2-dimensional to high dimensional problem.

**Proof 13** *In the proofs of Lemma C.2 (and Lemma F.3 in Appendix), we construct equivalent optimization problems by introducing new variables in the constraints. Unfortunately, in the proof of Lemma 2, we cannot use the same step. Otherwise, the bound would be too complicated to resolve. To obtain a bound independent of the dimension n, we need to relax the constraint in the beginning.*

*Our aim is to find an upper bound of $KL((\mathcal{N}(\mu_1, \Sigma_1)||\mathcal{N}(\mu_2, \Sigma_2))$ under the constraints $KL(\mathcal{N}(\mu_1, \Sigma_1)||\mathcal{N}(0, I)) \leq \varepsilon_1$, $KL(\mathcal{N}(0, I)||\mathcal{N}(\mu_2, \Sigma_2)) \leq \varepsilon_2$. In the following, we first relax the constraints and then find an upper bound under the relaxed constraints.*

*According to the definition of KL divergence, we have*

$$KL((\mathcal{N}(\mu_1, \Sigma_1)||\mathcal{N}(\mu_2, \Sigma_2)) = \frac{1}{2}\left(\log\frac{|\Sigma_2|}{|\Sigma_1|} + \text{Tr}(\Sigma_2^{-1}\Sigma_1) + (\mu_2 - \mu_1)^\top \Sigma_2^{-1}(\mu_2 - \mu_1) - n\right)$$

*In the following two steps, we first find an upper bound for the first two items, then we find an upper bound for the rest items.*

**Step 1**. *According to Lemma H.9, we have*

$$\log\frac{|\Sigma_2|}{|\Sigma_1|} + \text{Tr}(\Sigma_2^{-1}\Sigma_1)$$

$$= \text{Tr}(\Sigma_2^{-1}\Sigma_1) - \log\frac{|\Sigma_1|}{|\Sigma_2|}$$

$$= \text{Tr}(\Sigma_2^{-1}\Sigma_1) - \log(|\Sigma_2^{-1}||\Sigma_1|)$$

$$= \text{Tr}(\Sigma_2^{-1}\Sigma_1) - \log\prod_{i=1}^n \lambda_{1,i}\lambda'_{2,i}$$

$$\leq \sum_{i=1}^n \lambda_{1,[i]}\lambda'_{2,[i]} - \log\prod_{i=1}^n \lambda_{1,[i]}\lambda'_{2,[i]}$$

$$= \sum_{i=1}^n \lambda_{1,[i]}\lambda'_{2,[i]} - \log\lambda_{1,[i]}\lambda'_{2,[i]} \tag{H.164}$$

*where $\lambda_{1,i}, \lambda'_{2,i}$ are the eigenvalues of $\Sigma_1, \Sigma_2^{-1}$ arranged in decreasing order, respectively. In the following, we find an upper bound for Equation (H.164).*

*By the definition of KL divergence, the constraint $KL(\mathcal{N}(\mu_1, \Sigma_1)||\mathcal{N}(0, I)) \leq \varepsilon_1$ is equal to*

$$-\log|\Sigma_1| + \text{Tr}(\Sigma_1) + \mu_1^\top\mu_1 - n \leq 2\varepsilon_1 \tag{H.165}$$

*Combining Lemma B.1a, Equation (C.25) and (C.28), we relax the constraint in Inequality (H.165) as follows.*

$$-\log|\Sigma_1| + \text{Tr}(\Sigma_1) = \sum_{i=1}^n \lambda_{1,i} - \log\lambda_{1,i} \leq n + 2\varepsilon_1 \tag{H.166}$$

$$\mu_1^\top\mu_1 \leq 2\varepsilon_1 \tag{H.167}$$

*where $\lambda_{1,i}$ are the eigenvalues of $\Sigma_1$. For simplicity, we modify the constraint in Inequality (H.166) to the following constraint.*

$$-\log|\Sigma_1| + \text{Tr}(\Sigma_1) = \sum_{i=1}^n \lambda_{1,i} - \log\lambda_{1,i} = n + 2\varepsilon_1 \tag{H.168}$$

*In the following, we find the upper bound for Equation (H.164) under constraints (H.168). Then we will see that the upper bound is increasing with $\varepsilon_1$. So there is no difference between constraints (H.166) and (H.168).*

*Form the perspective of optimization, the constraint in Inequality (H.168) can be replaced by the following constraints*

$$\lambda_{1,i} - \log \lambda_{1,i} = 1 + \varepsilon_{1,i} \ (1 \le i \le n) \tag{H.169}$$

$$\bigwedge_{i=1}^{n} \varepsilon_{1,i} \ge 0 \wedge \sum_{i=1}^{n} \varepsilon_{1,i} = 2\varepsilon_1 \tag{H.170}$$

*Similarly, the constraint $KL(\mathcal{N}(0,I)||\mathcal{N}(\boldsymbol{\mu}_2, \boldsymbol{\Sigma}_2)) \le \varepsilon_2$ is equal to*

$$\log |\boldsymbol{\Sigma}_2| + \text{Tr}(\boldsymbol{\Sigma}_2^{-1}) + \boldsymbol{\mu}_2^\top \boldsymbol{\Sigma}_2^{-1} \boldsymbol{\mu}_2 - n \le 2\varepsilon_2 \tag{H.171}$$

*which implies the following constraints*

$$\log |\boldsymbol{\Sigma}_2| + \text{Tr}(\boldsymbol{\Sigma}_2^{-1}) = \sum_{i=1}^{n} \lambda'_{2,i} - \log \lambda'_{2,i} \le n + 2\varepsilon_2 \tag{H.172}$$

$$\boldsymbol{\mu}_2^\top \boldsymbol{\Sigma}_2^{-1} \boldsymbol{\mu}_2 \le 2\varepsilon_2 \tag{H.173}$$

*where $\lambda'_{2,i}$ are the eigenvalues of $\boldsymbol{\Sigma}_2^{-1}$. We also modify the constraint in Inequality (H.172) to the following constraint which does not affect the upper bound.*

$$\log |\boldsymbol{\Sigma}_2| + \text{Tr}(\boldsymbol{\Sigma}_2^{-1}) = \sum_{i=1}^{n} \lambda'_{2,i} - \log \lambda'_{2,i} = n + 2\varepsilon_2 \tag{H.174}$$

*Furthermore, constraint (H.174) can be replaced by the following constraints.*

$$\lambda'_{2,i} - \log \lambda'_{2,i} = 1 + \varepsilon_{2,i} \ (1 \le i \le n) \tag{H.175}$$

$$\bigwedge_{i=1}^{n} \varepsilon_{2,i} \ge 0 \wedge \sum_{i=1}^{n} \varepsilon_{2,i} = 2\varepsilon_2 \tag{H.176}$$

*In the following, we find an upper bound of Equation (H.164) under constraints (H.169), (H.170), (H.175), and (H.176).*

*Applying Lemma G.8 to Equation (H.164) with conditions (H.169) and (H.175), we can obtain*

$$\sum_{i=1}^{n} \lambda_{1,[i]} \lambda'_{2,[i]} - \log \lambda_{1,[i]} \lambda'_{2,[i]} \le \sum_{i=1}^{n} f(w_2(\varepsilon_{1,[i]})w_2(\varepsilon_{2,[i]})) \tag{H.177}$$

*where $\varepsilon_{1,[i]}$ and $\varepsilon_{2,[i]}$ are also arranged in decreasing order.*

*Now we apply Lemma G.5 to the RHS of Inequality (H.177) repeatedly on the first two dimensions as follows. Here we use notations $E_{1,k} = \sum_{i=1}^{k} \varepsilon_{1,[i]}, E_{2,k} = \sum_{i=1}^{k} \varepsilon_{2,[i]}$ for brevity.*

$$
\begin{aligned}
&\log \frac{|\boldsymbol{\Sigma_2}|}{|\boldsymbol{\Sigma_1}|} + \text{Tr}(\boldsymbol{\Sigma_2^{-1}\Sigma_1}) && \text{\textit{by (H.164)}} \\
&\le \sum_{i=1}^{n} \lambda_{1,[i]} \lambda'_{2,[i]} - \log \lambda_{1,[i]} \lambda'_{2,[i]} && \text{\textit{by (H.177)}} \\
&\le \sum_{i=1}^{n} f(w_2(\varepsilon_{1,[i]})w_2(\varepsilon_{2,[i]})) \\
&= f(w_2(\varepsilon_{1,[1]})w_2(\varepsilon_{2,[1]})) + f(w_2(\varepsilon_{1,[2]})w_2(\varepsilon_{2,[2]})) + \sum_{i=3}^{n} f(w_2(\varepsilon_{1,[i]})w_2(\varepsilon_{2,[i]})) && \text{\textit{Lemma G.5}} \\
&\le f(w_2(\varepsilon_{1,[1]} + \varepsilon_{1,[2]})w_2(\varepsilon_{2,[1]} + \varepsilon_{2,[2]})) + 1 + \sum_{i=3}^{n} f(w_2(\varepsilon_{1,[i]})w_2(\varepsilon_{2,[i]})) \\
&= f(w_2(E_{1,2})w_2(E_{2,2})) + f(w_2(\varepsilon_{1,[3]})w_2(\varepsilon_{2,[3]})) + \sum_{i=4}^{n} f(w_2(\varepsilon_{1,[i]})w_2(\varepsilon_{2,[i]})) + 1 && \text{\textit{Lemma G.5}} \\
&\le f(w_2(E_{1,2} + \varepsilon_{1,[3]})w_2(E_{2,2} + \varepsilon_{2,[3]})) + 1 + \sum_{i=4}^{n} f(w_2(\varepsilon_{1,[i]})w_2(\varepsilon_{2,[i]})) + 1 \\
&= f(w_2(E_{1,3})w_2(E_{2,3})) + \sum_{i=4}^{n} f(w_2(\varepsilon_{1,[i]})w_2(\varepsilon_{2,[i]})) + 2 \\
&\cdots \\
&\le f(w_2(E_{1,n})w_2(E_{2,n})) + n - 1 \\
&= f(w_2(\sum_{i=1}^{n} \varepsilon_{1,[i]})w_2(\sum_{i=1}^{n} \varepsilon_{2,[i]})) + n - 1 \\
&= f(w_2(2\varepsilon_1)w_2(2\varepsilon_2)) + n - 1 && \text{\textit{Lemma B.1j}} \\
&= 2\varepsilon_1 + 2\varepsilon_2 + 2 + w_2(2\varepsilon_1)w_2(2\varepsilon_2) - w_2(2\varepsilon_1) - w_2(2\varepsilon_2) + n - 1 \\
&= 2\varepsilon_1 + 2\varepsilon_2 + w_2(2\varepsilon_1)w_2(2\varepsilon_2) - w_2(2\varepsilon_1) - w_2(2\varepsilon_2) + n + 1 \tag{H.178}
\end{aligned}
$$

The bound in Equation (H.178) is increasing with $\varepsilon_1$ and $\varepsilon_2$. Therefore, the constraints (H.168) and (H.174) can be modified back to (H.166) and (H.172), respectively .

**Step 2**. from Equation (H.167), we know

$$|\boldsymbol{\mu}_1| \le \sqrt{2\varepsilon_1} \tag{H.179}$$

where $|\cdot|$ denotes the $L_2$ norm of vector. From Inequality (H.173), we also know

$$\lambda'_{2*}\boldsymbol{\mu}_2^\top \boldsymbol{\mu}_2 \le \boldsymbol{\mu}_2^\top \boldsymbol{\Sigma}_2^{-1} \boldsymbol{\mu}_2 \le 2\varepsilon_2 \implies \boldsymbol{\mu}_2^\top \boldsymbol{\mu}_2 \le \frac{2\varepsilon_2}{\lambda'_{2*}} \tag{H.180}$$

where $\lambda'_{2*}$ is the minimum eigenvalue of $\boldsymbol{\Sigma}_2^{-1}$. Combining the condition (H.172) and Lemma B.1g, we can know $\lambda'_{2*} \ge w_1(2\varepsilon_2)$. Thus we can obtain

$$\boldsymbol{\mu}_2^\top \boldsymbol{\mu}_2 \le \frac{2\varepsilon_2}{\lambda'_{2*}} \le \frac{2\varepsilon_2}{w_1(2\varepsilon_2)} \implies |\boldsymbol{\mu}_2| \le \sqrt{\frac{2\varepsilon_2}{w_1(2\varepsilon_2)}} \tag{H.181}$$

Combining Inequalities (H.179), (H.181) and using the triangle inequality for norms of vectors, we have

$$|\boldsymbol{\mu}_2 - \boldsymbol{\mu}_1| \le |\boldsymbol{\mu}_2| + |\boldsymbol{\mu}_1| \le \sqrt{2\varepsilon_1} + \sqrt{\frac{2\varepsilon_2}{w_1(2\varepsilon_2)}} \tag{H.182}$$

Again, we have $(\boldsymbol{\mu}_2 - \boldsymbol{\mu}_1)^\top \boldsymbol{\Sigma}_2^{-1} (\boldsymbol{\mu}_2 - \boldsymbol{\mu}_1) \le \lambda'^*_2 |\boldsymbol{\mu}_2 - \boldsymbol{\mu}_1|^2$, where $\lambda'^*_2$ is the maximum eigenvalue of $\boldsymbol{\Sigma}_2^{-1}$. From Lemma B.1g and condition (H.172), we know $\lambda'^*_2 \le w_2(2\varepsilon_2)$. Thus, we can conclude that

$$(\boldsymbol{\mu}_2 - \boldsymbol{\mu}_1)^\top \boldsymbol{\Sigma}_2^{-1} (\boldsymbol{\mu}_2 - \boldsymbol{\mu}_1) \le w_2(2\varepsilon_2)|\boldsymbol{\mu}_2 - \boldsymbol{\mu}_1|^2 \le w_2(2\varepsilon_2)\left(\sqrt{2\varepsilon_1} + \sqrt{\frac{2\varepsilon_2}{w_1(2\varepsilon_2)}}\right)^2 \tag{H.183}$$

Finally, combining Inequalities (H.178) and (H.183), we can conclude that

$$KL((\mathcal{N}(\boldsymbol{\mu}_1, \boldsymbol{\Sigma}_1)||\mathcal{N}(\boldsymbol{\mu}_2, \boldsymbol{\Sigma}_2))$$

$$< \frac{1}{2}\left(2\varepsilon_1 + 2\varepsilon_2 + w_2(2\varepsilon_1)w_2(2\varepsilon_2) - w_2(2\varepsilon_1) - w_2(2\varepsilon_2) + n + 1 + w_2(2\varepsilon_2)\left(\sqrt{2\varepsilon_1} + \sqrt{\frac{2\varepsilon_2}{w_1(2\varepsilon_2)}}\right)^2 - n\right)$$

$$= \varepsilon_1 + \varepsilon_2 + \frac{1}{2}\left(W_{-1}(-e^{-(1+2\varepsilon_1)})W_{-1}(-e^{-(1+2\varepsilon_2)}) + W_{-1}(-e^{-(1+2\varepsilon_1)}) + W_{-1}(-e^{-(1+2\varepsilon_2)}) + 1\right.$$

$$\left. - W_{-1}(-e^{-(1+2\varepsilon_2)})\left(\sqrt{2\varepsilon_1} + \sqrt{\frac{2\varepsilon_2}{-W_0(-e^{-(1+2\varepsilon_2)})}}\right)^2\right) \tag{H.184}$$

$\square$

# I  Proof of Theorem 4

**Proof 14** For $X_2 \sim \mathcal{N}(\boldsymbol{\mu}_2, \boldsymbol{\Sigma}_2)$, there exists an invertible matrix $B_2$ such that $X'_2 = B_2^{-1}(X_2 - \boldsymbol{\mu}_2) \sim \mathcal{N}(0, I)$ [4]. Here $B_2 = P_2 D_2^{1/2}$, $P_2$ is an orthogonal matrix whose columns are the eigenvectors of $\boldsymbol{\Sigma}_2$, $D_2 = diag(\lambda_{2,1}, \ldots, \lambda_{2,n})$ whose diagonal elements are the corresponding eigenvalues. We define the following two invertible linear transformations $T, T^{-1}$ on random vectors.

$$X' = T(X) = B_2^{-1}(X - \boldsymbol{\mu}_2), \ X = T^{-1}(X') = B_2 X' + \boldsymbol{\mu}_2 \tag{I.185}$$

Applying transformation $T$ on $X_1, X_2, X_3$, we can get three Gaussians.

$$X'_1 = T(X_1) \sim \mathcal{N}(\boldsymbol{\mu}'_1, \boldsymbol{\Sigma}'_1)$$
$$X'_2 = T(X_2) \sim \mathcal{N}(0, I)$$
$$X'_3 = T(X_3) \sim \mathcal{N}(\boldsymbol{\mu}'_3, \boldsymbol{\Sigma}'_3)$$

*According to Proposition D.1, $T$ and $T^{-1}$ preserve KL divergence. Thus, we have*

$$KL(\mathcal{N}(\boldsymbol{\mu}_1, \boldsymbol{\Sigma}_1)||\mathcal{N}(\boldsymbol{\mu}_2, \boldsymbol{\Sigma}_2)) = KL(\mathcal{N}(\boldsymbol{\mu}_1', \boldsymbol{\Sigma}_1')||\mathcal{N}(0, I)) \tag{I.186}$$

$$KL(\mathcal{N}(\boldsymbol{\mu}_2, \boldsymbol{\Sigma}_2)||\mathcal{N}(\boldsymbol{\mu}_3, \boldsymbol{\Sigma}_3)) = KL(\mathcal{N}(0, I)||\mathcal{N}(\boldsymbol{\mu}_3', \boldsymbol{\Sigma}_3')) \tag{I.187}$$

$$KL(\mathcal{N}(\boldsymbol{\mu}_1, \boldsymbol{\Sigma}_1)||\mathcal{N}(\boldsymbol{\mu}_3, \boldsymbol{\Sigma}_3)) = KL(\mathcal{N}(\boldsymbol{\mu}_1', \boldsymbol{\Sigma}_1')||\mathcal{N}(\boldsymbol{\mu}_3', \boldsymbol{\Sigma}_3')) \tag{I.188}$$

*Combining the preconditions and Equations* (I.186), (I.187), *we can know*

$$KL(\mathcal{N}(\boldsymbol{\mu}_1', \boldsymbol{\Sigma}_1')||\mathcal{N}(0, I)) \le \varepsilon_1, \ KL(\mathcal{N}(0, I)||\mathcal{N}(\boldsymbol{\mu}_2', \boldsymbol{\Sigma}_2')) \le \varepsilon_2 \tag{I.189}$$

*Now we can apply Lemma 2 on $\mathcal{N}(\boldsymbol{\mu}_1', \boldsymbol{\Sigma}_1')$, $\mathcal{N}(0, I)$ and $\mathcal{N}(\boldsymbol{\mu}_3', \boldsymbol{\Sigma}_3'))$ and get the bound of $KL(\mathcal{N}(\boldsymbol{\mu}_1', \boldsymbol{\Sigma}_1')||\mathcal{N}(\boldsymbol{\mu}_3', \boldsymbol{\Sigma}_3'))$. Finally, combining Equation* (I.188), *we can prove Theorem 4.*

$\square$

## J Proof of Theorem 5

**Proof 15** *Suppose that $\varepsilon_1, \varepsilon_2$ are sufficiently small. According to the series expanding $W_0$ and $W_1$ (Section III.17 in [7]), we have*

$$W_0(-e^{-(1+2\varepsilon)}) = -1 + 2\sqrt{\varepsilon} + O(\varepsilon) \tag{J.190}$$

$$W_{-1}(-e^{-(1+2\varepsilon)}) = -1 - 2\sqrt{\varepsilon} + O(\varepsilon) \tag{J.191}$$

*So we can obtain*

$$
\begin{aligned}
& W_{-1}(-e^{-(1+2\varepsilon_1)})W_{-1}(-e^{-(1+2\varepsilon_2)}) + W_{-1}(-e^{-(1+2\varepsilon_1)}) + W_{-1}(-e^{-(1+2\varepsilon_2)}) + 1 \\
= & (W_{-1}(-e^{-(1+2\varepsilon_1)}) + 1)(W_{-1}(-e^{-(1+2\varepsilon_2)}) + 1) \\
= & (2\sqrt{\varepsilon_1} + O(\varepsilon_1))(2\sqrt{\varepsilon_2} + O(\varepsilon_2)) \\
= & 4\sqrt{\varepsilon_1\varepsilon_2} + o(\varepsilon_1) + o(\varepsilon_2)
\end{aligned}
\tag{J.192}
$$

*and*

$$
\begin{aligned}
& -W_{-1}(-e^{-(1+2\varepsilon_2)})\left(\sqrt{2\varepsilon_1} + \sqrt{\frac{2\varepsilon_2}{-W_0(-e^{-(1+2\varepsilon_2)})}}\right)^2 \\
= & (1 + 2\sqrt{\varepsilon_2} + O(\varepsilon_2))\left(\sqrt{2\varepsilon_1} + \sqrt{\frac{2\varepsilon_2}{1 - 2\sqrt{\varepsilon_2} + O(\varepsilon_2)}}\right)^2 \\
\le & (1 + 2\sqrt{\varepsilon_2} + O(\varepsilon_2))\left(4\varepsilon_1 + \frac{4\varepsilon_2}{1 - 2\sqrt{\varepsilon_2} + O(\varepsilon_2)}\right) \\
= & 4\varepsilon_1 + o(\varepsilon_1) + o(\varepsilon_2) + \frac{4\varepsilon_2(1 + 2\sqrt{\varepsilon_2} + O(\varepsilon_2))}{1 - 2\sqrt{\varepsilon_2} + O(\varepsilon_2)} \\
= & 4\varepsilon_1 + o(\varepsilon_1) + o(\varepsilon_2) + 4\varepsilon_2 + \frac{4\varepsilon_2(4\sqrt{\varepsilon_2} + O(\varepsilon_2))}{1 - 2\sqrt{\varepsilon_2} + O(\varepsilon_2)} \\
= & 4\varepsilon_1 + 4\varepsilon_2 + o(\varepsilon_1) + o(\varepsilon_2) + O(\varepsilon_2^{1.5})
\end{aligned}
\tag{J.193}
$$

*Using Equations* (J.192) *and* (J.193), *we can rewrite the bound in Theorem 4 as*

$$KL((\mathcal{N}(\boldsymbol{\mu}_1, \boldsymbol{\Sigma}_1)||\boldsymbol{\Sigma}(\boldsymbol{\mu}_3, \boldsymbol{\Sigma}_3)) < 3\varepsilon_1 + 3\varepsilon_2 + 2\sqrt{\varepsilon_1\varepsilon_2} + o(\varepsilon_1) + o(\varepsilon_2) \tag{J.194}$$

$\square$

## K Discussion

### K.1 Comparison with Existing Inequalities

The bound in our relaxed triangle inequality is independent of the parameters of Gaussians and only related to $\varepsilon_1$ and $\varepsilon_2$. Our result is different from existing Pythagoras inequalities satisfied by KL divergence. We list them as follows.

1. The generalized Pythagoras inequality for KL divergence [5, 16] states that for a convex set of distributions $\mathcal{P}$, any distribution $Q$ not in $\mathcal{P}$, and $D_{min} = \inf_{P \in \mathcal{P}} KL(P||Q)$, there exists a distribution $P^*$ such that

$$KL(P||Q) \geq KL(P||P^*) + D_{min} \quad \text{for all} \quad P \in \mathcal{P}$$

2. Erven *et al.* generalize the Pythagoras inequality for KL divergence to Rényi divergence which includes KL divergence with order 1. See [16] for details.

3. Functional Bregman divergence also satisfies a generalized Pythagoras theorem [8]. Let $(\mathbb{R}^d, \Omega, \nu)$ be a measure space, where $d$ is a positive integer and $\nu$ is a Borel measure. Let $\mathcal{A}$ be a convex subset of $L^p(\nu)$. For any $f, g, h \in \mathcal{A}$, functional Bregman divergence $d_\phi$ satisfies

$$d_\phi[f, h] = d_\phi[f, g] + d_\phi[g, h] + \delta\phi[g; f - g] - \delta\phi[h; f - g] \tag{K.195}$$

where $\phi : L^p(\nu) \to \mathbb{R}$ is a strictly convex, twice-continuously Fréchet-differentiable functional. $\delta\phi[g; \cdot]$ is the Fréchet derivative of $\phi$ at $g$. KL divergence is a special form of functional Bregman divergence when $\phi = \int p(x) \log p(x) \, dx$ whose Fréchet derivative at $g$ is $\delta\phi[g; t] = \int (\log g(x) + 1) t(x) \, dx$. Plugging $\phi$ and $\delta\phi$ into Equation (K.195), we get

$$
\begin{aligned}
&KL(f||h) \\
=&KL(f||g) + KL(g||h) + \int (\log g(x) + 1)(f(x) - g(x)) \, dx \\
&- \int (\log h(x) + 1)(f(x) - g(x)) \, dx \\
=&KL(f||g) + \int f(x) \log \frac{g(x)}{h(x)} \, dx
\end{aligned}
\tag{K.196}
$$

All the bounds in the above inequalities are dependent on the parameters of the given distributions.

## L    Applications

### L.1    Applications of Theorem 1

**Providing Theoretical Guarantee for Continuous Gaussian policy**. Theorem 1 can extend existing theoretical guarantee in offline reinforcement learning [13] to continuous Gaussian policy. In [13], Nair *et al.* propose AWAC method to accelerate online reinforcement learning with offline datasets. They use a KL term as constraint on the policy improvement update to avoid bootstrapping on out-of-distribution actions. In the original inequalities (24) and (25) in [13], the authors use Pinsker's inequality [6] to build the relation between forward and reverse KL divergences such that minimizing the reverse KL divergence also bounds the forward KL divergence. Here we reformulate the original inequalities (24) and (25) in [13] as follows.

$$KL(\pi^*||\pi_\theta) \leq \frac{2}{\alpha_\theta} D_{TV}(\pi^*, \pi_\theta)^2 \leq \frac{1}{\alpha_\theta} KL(\pi_\theta||\pi^*) \tag{L.197}$$

where $\pi^*$ and $\pi_\theta$ are policy distributions, $D_{TV}$ is total variation, and $\alpha_\theta = \min \pi_\theta$ is the infimum of density function of $\pi_\theta$ over finite space. To make the above Inequalities (L.197) hold, it requires that $\alpha_\theta = \min \pi_\theta$. However, the above inequalities do not hold for the commonly used continuous Gaussian policy because $\alpha_\theta$ equals 0. That is why the authors of [13] only discuss discrete policy on the above inequalities in [13].

Now with the help of Theorem 1, we can extend the above guarantee to the commonly used Gaussian policies in continuous space tasks. When both $\pi^*$ and $\pi_\theta$ are Gaussian distributions, Theorem 1 builds the relation between forward and reverse KL divergence without Pinsker's inequality. *We can guarantee that minimizing the reverse KL divergence also bounds the forward KL divergence.* Note that their method AWAC can be implemented for continuous policies in practice.

**Bringing New Insights to Existing Reinforcement Learning Algorithm**. In [1], Abdolmaleki *et al.* propose the MPO algorithm for reinforcement learning. The MPO algorithm employs the powerful Expectation-Maximization (EM) to solve control problems. It introduces KL constraints controlling

the policy change in both E and M steps, aiming to yield robust learning. In the constrained E-step, they use a KL term $\mathbb{E}[KL(q||\pi)] < \varepsilon$ as constraint (see original Equation (7) in [1]). The authors also state that their method is similar to TRPO [15] algorithm for continuous control, except that being in an off-policy setting and the KL term is reversed. Now based on the theoretical guarantee provided by Theorem 2, the reverse KL constraint in the original Equation (7) in [1] can be replaced with forward KL constraint. Therefore, the KL terms used in both methods can be unified. Besides, in [1], the authors point out that in E-step they use reverse, mode-seeking KL and in M-step they use forward, moment-matching KL term. These KL constraints can greatly increase the stability of the algorithm. Theorem 1 can eliminate such difference for continuous Gaussian policies.

**Bridging Research on Sample Complexity of Learning Gaussian Distribution**. Theorem 1 can bridge existing research on sample complexity of Gaussian distribution. In [2], Ashtiani *et al.* propose a compression-based learning method and establish an optimal lower bound of sample complexity of learning Gaussian mixtures. For a fixed target Gaussian mixture distribution $P$, their learning method receives a sample set and outputs a distribution $Q$ satisfying $KL(Q||P) \leq \varepsilon$. See page 26, the inequality below Equation (17) in [2], where KL divergence is used to bound Total Variation distance. Their conclusion applies to a single Gaussian when the number of mixture components is 1. One open problem proposed in their paper is what is the sample complexity for learning Gaussian mixtures with guarantee using the reverse KL divergence $KL(P||Q) \leq \varepsilon$ (see page 35 in [2]). Our Theorem 1 on the approximate symmetry can extend existing theory and answer this open problem in the single Gaussian case. According to Theorem 1, when $KL(Q||P) \leq \varepsilon$, $KL(P||Q) \leq \varepsilon + 2\varepsilon^{1.5} + O(\varepsilon^2)$. The supremum equals $O(\varepsilon)$ when $\varepsilon$ is small. This implies that the bounds of forward and reverse KL divergence have the same order. Therefore, the optimal sample complexity for learning single Gaussian is the same when using reverse KL divergence as a guarantee. Therefore, we answer the open problem proposed in [2] in the single Gaussian case.

Similarly, in [3], the authors propose a learning method for sparse fixed-structure Gaussian Bayesian network, which can be treated as a representation of multidimensional Gaussian distributions [9]. They prove the sample complexity of their method for learning Gaussian distribution with guarantee $KL(P||Q) \leq \varepsilon$, where $P$ is the target Gaussian distribution from which samples are drawn, $Q$ is the learned Gaussian distribution. Specially, in page 9 of [3], the authors note that their theoretical result uses reverse KL divergence while [2] discussed above uses forward KL divergence. Again, Theorem 1 can extend their conclusion to forward KL divergence and eliminate the difference between forward and reverse KL divergence.

To summarize, researchers have proposed algorithms for learning a multivariate Gaussian in either KL directions separately so far and give a similar learning result. Theorem 1 can eliminate the difference between forward and reverse KL divergence in this scenario.

## L.2   Application of Theorem 4

**Extending One-step Safety Guarantee to Multiple Steps in Reinforcement Learning**. After we post the last version of this manuscript on Arxiv, the relaxed triangle inequality (Theorem 4) is applied as a critical step in the research of constrained variational policy optimization for safe reinforcement learning [11]. Liu *et al.* propose an Expectation-Maximization style approach for learning safe policy in reinforcement learning. In the original Section 3.4 in [11], the authors state that their algorithm can achieve multiple steps robustness under the commonly used Gaussian policy in continuous action space tasks. In more detail, the authors want to extend one-step robustness guarantee to multiple steps. This requires triangle inequality for consecutive updated policies. It is known that KL divergence does not have such property in general cases. However, multivariate Gaussian is commonly used as policy in continuous action space tasks. In such context, the relaxed triangle inequality (Theorem 4) can extend one-step robustness guarantee to multiple steps. In [11], the authors use the original Proposition 4 to formulate the above extension, which is presented in our Theorem 4 and 5. In particular, the authors simplify the bound[4] in Theorem 4 and 5 in case $\varepsilon_1 = \varepsilon_2$, obtaining $3\varepsilon_1 + 3\varepsilon_2 + 2\sqrt{\varepsilon_1 \varepsilon_2} + o(\varepsilon_1) + o(\varepsilon_2) = 8\varepsilon_2 + o(\varepsilon_2)$. This is why they obtain a bound "$\frac{\varepsilon_2}{8}$" in their Proposition 4. In summary, Theorem 4 is indispensable to achieve multiple steps robustness in their safe reinforcement learning algorithm. Please see the original Section 3.4 and Proposition 4 in [12] for more details about the application.

---

[4]Liu *et al.* simplify the bound after we submit the last version of this manuscript to Arxiv, in which we did not contain such simplification.