# OpenReview forum: "On the Properties of Kullback-Leibler Divergence Between Multivariate Gaussian Distributions"
_NeurIPS.cc/2023/Conference — NeurIPS 2023 poster_

### Official Review · Reviewer_8dDm · 2023-06-19

**Soundness:** 3 good
**Presentation:** 3 good
**Contribution:** 3 good
**Rating:** 6
**Confidence:** 3

**Summary:**

This paper explores and proves some properties of KL divergence between multivariate Gaussian distributions. One of the motivations is that as a statistical distance, KL divergence does not satisfy the properties of a metric, that is, symmetry and triangle inequality. In spite of these issues, this paper proposes the relaxed versions. To be specific, it proves the lower bound (resp. upper bound) for reverse KL divergence given the lower bound (resp. upper bound) of forward KL divergence, and the summation upper bound of two bounded KL divergences. Finally, the proposed techniques are applied to anomaly detection with flow based model and reinforcement learning.

**Strengths:**

(1) This paper proves the lower bound (resp. upper bound) for reverse KL divergence given the lower bound (resp. upper bound) of forward KL divergence, and the summation upper bound of two bounded KL divergences.
(2) The theoretical results can be applied to some applications in deep learning and reinforcement learning.
(3) This paper is well-written and easy to understand.


**Weaknesses:**

(1) Theorem 1 and Theorem 3 hold when two conditions are satisfied. For example, for the mean, it requires $\mu_1 = \mu_2$, which is too strong in practice.
(2) Since the KL divergence has a wide range of applications, the two applications shown in this paper are kind of limited and not convincing.

**Questions:**

The KL divergence is widely used in machine learning and statistics, etc. Can the theoretical results in this paper be used to some other tasks in machine learning?

**Limitations:**

See Weaknesses.

---

> ### Author Rebuttal · Authors · 2023-08-03
>
> Thanks for your careful and valuable comments. We explain your concerns as follows.
>
> **C1**: Theorem 1 and Theorem 3 hold when two conditions are satisfied. For example, for the mean, it requires $\mu_1=\mu_2$, which is too strong in practice.
>
> **A1**: **These strong conditions are advantages rather than disadvantages when applying our theorems**.
>
> In Theorem 1 and Theorem 3, we give the strong conditions when the supremum/infimum can be attained. We can benefit from these strong conditions in applications.
>
> As we discussed in Section 5.2, the approximate symmetry of KL divergence between Gaussians brings the following convenience to us.
>
> (1) Minimizing one of forward and reverse KL divergences also bounds another.
> (2) We can exchange forward and reverse KL divergences for small $\epsilon$.
>
> For example, when applying the approximate symmetry of KL divergence between Gaussians (Theorem 1), we know the forward KL divergence is small ($\leq \epsilon$). We want to guarantee that the reverse KL divergence is also small such that bounding forward KL divergence also bounds the reverse KL divergence. Theorem 1 states that the supremum of reverse KL divergence is $\epsilon + 2\epsilon^{1.5}+O(\epsilon^2)$, which is a small bound. Since the conditions needed to attain the supremum are strong, it is hard to reach the supremum in practice. This implies that when these strong conditions do not hold, the reverse KL divergence is even smaller than the supremum. That is just what we want in applications. In other words, the supremum describes the *worst case* in application. The stronger these conditions are, the harder to meet the worst case.
>
> To summarize, Theorem 1 has the following two meanings.
>
> (1) The supremum of reverse KL divergence $\epsilon + 2\epsilon^{1.5}+O(\epsilon^2)$ is small. This tells us the worst case itself in application is acceptable.
>
> (2) The strong conditions tell us the worst case barely happens in practice, so the reverse KL divergence is usually smaller than the supremum. This is just what we want in applications.
>
> Therefore, the strong conditions to attain the supremum are our theorem's advantage rather than disadvantage. We will add more discussion to explain this point in the revision.
>
> **C2**: Since the KL divergence has a wide range of applications, the *two* applications shown in this paper are kind of limited and not convincing. Can the theoretical results in this paper be used to some other tasks in machine learning?
>
> **A2**: **Yes!** We have discussed **four** (not two) applications ranging from deep anomaly detection to reinforcement learning in our paper. In the *common rebuttal for all reviewers*, we summarize four existing applications and discuss one new important application in sample complexity research. Please see the *common rebuttal for details*. These five applications have demonstrated the usefulness of our theory. Our theory may have other potential applications.
>
> Thanks again for your valuable comments.

---

> > ### Comment · Reviewer_8dDm · 2023-08-15
> >
> > Thank you for your clear explanations! I will update my score.

---

### Official Review · Reviewer_ZfdJ · 2023-06-22

**Soundness:** 3 good
**Presentation:** 2 fair
**Contribution:** 2 fair
**Rating:** 6
**Confidence:** 3

**Summary:**

Kullback-Leibler (KL) divergence is an important measure of distance between probability distributions with uses in statistics, information theory and many other fields. However, it is not a proper distance measure, since it is not symmetric and does not satisfy the triangle inequality in general. The authors consider the KL divergence between multivariate Gaussian distributions and show that a relaxed notion of symmetry and triangle inequality holds under certain conditions. Specifically, they formulate an upper bound on KL(N2,N1) when KL(N1,N2) < epsilon, and show that it cannot be much greater than epsilon. Similarly, they give a lower bound on KL(N2,N1) when KL(N1,N2)  > M. Finally, they upper bound KL(N1,N3) when KL(N1,N2) < epsilon1 and KL(N2,N3) < epsilon2. They conclude by discussing several applications of the results in deep learning and reinforcement learning.

**Strengths:**

The disadvantages of KL divergence as far as symmetry and triangle inequality go are well known, so finding conditions in which even a relaxed version of these properties hold is interesting and potentially useful.

**Weaknesses:**

Firstly, due to the continuity of the KL divergence around epsilon=0 (N1=N2), the results are not too surprising. The proofs are technical, lengthy and somewhat repetitive. Lemma G.5 in particular is not so digestible for readers. Secondly, the structure of the paper is unorthodox: usually you would have the Related Work section right after the Introduction instead of before Conclusions; that would also be a better place to start mentioning the applications for which your results might be relevant. The section called Lemmas and Notations has no lemmas. The Applications section should be called Discussion.

**Questions:**

- Can you please rearrange the structure of the paper to be more in line with convention?
- Can Lemma G.5 be made more edible, or could a more informative overview be given?

-- The authors have thoroughly addressed these points in the rebuttal.

**Limitations:**

There is no potential negative societal impact of the work

---

> ### Author Rebuttal · Authors · 2023-08-03
>
> Thanks for your careful and valuable comments. We explain your concerns as follows.
>
> **C1**: due to the continuity of the KL divergence around epsilon=0, the results are not too surprising.
>
> **A1**: The task of this work is to (1) *quantify* the approximate symmetry and (2) *find* a relaxed triangle inequality. Such issues have hindered researchers for a long time. See the application section and *common rebuttal for all reviewers* for details.
>
> **C2**: Can you please rearrange the structure of the paper to be more in line with convention?
>
> **A2**: We will move the related work section after the Introduction.
>
> **C3**: Can Lemma G.5 be made more edible, or could a more informative overview be given?
>
> **A3**: Proving Lemma G.5 is hard. Appendix G gives a proof sketch. Here we give a more detailed proof sketch.
>
> In the LHS (resp. RHS) of Inequality G.94, $\varepsilon_{x,2}$ and $\varepsilon_{y,2}$ lie in the second (resp. first) term. We will use Inequality G.94 to move $\varepsilon_{x,2}, \varepsilon_{y,2}$ to the first item such that $\varepsilon_{x,1}, \varepsilon_{x,2}$ and $\varepsilon_{y,1},\varepsilon_{y,2}$ are allocated in only one dimension (see Equation H.178).
>
> We construct function
> $S(\theta_x, \theta_y)=f(w_2(\varepsilon_{x,1}+\theta_x\varepsilon_{x,2})w_2(\varepsilon_{y,1}+\theta_y\varepsilon_{y,2}))+f(w_2(\varepsilon_{x,2}-\theta_x\varepsilon_{x,2})w_2(\varepsilon_{y,2}-\theta_y\varepsilon_{y,2}))$ for $-\frac{\varepsilon_{x,1}}{\varepsilon_{x,2}}\leq \theta_x\leq 1,-\frac{\varepsilon_{y,1}}{\varepsilon_{y,2}}\leq \theta_y\leq 1 $ (see Equation G.112). When $\theta_x=\theta_y=0$, $S(\theta_x, \theta_y)=S(0,0)$ equals the LHS of Inequality G.94.
> Recall that $w_2(0)=1$ and $f(1)=1$. So when $\theta_x=\theta_y=1$, $S(\theta_x, \theta_y)=S(1,1)$ equals the RHS of Inequality G.94.
> When increasing $\theta_x$ and $\theta_y$ from 0 to 1, $\varepsilon_{x,2}, \varepsilon_{y,2}$ are moved to the first item gradually and
> the LHS approaches the RHS of Inequality G.94 gradually.
> $\theta_x$ and $\theta_y$ control how $\varepsilon_{x,2}$ and $\varepsilon_{y,2}$ are allocated among the two terms.
> We call $(\theta_x, \theta_y)$ and the corresponding pairs $(\varepsilon_{x,1}+\theta_x\varepsilon_{x,2},\ \varepsilon_{x,2}-\theta_x\varepsilon_{x,2})$ and
> $(\varepsilon_{y,1}+\theta_y\varepsilon_{y,2},\ \varepsilon_{y,2}-\theta_y\varepsilon_{y,2})$
> indiscriminately as *allocations*. When $\theta_x=\theta_y=1$, we call it an *extreme allocation*.
> To prove Inequality G.94, it suffices to show $S(0,0)\leq S(1,1)$.
>
> However, it is hard to prove $S(0,0)\leq S(1,1)$ directly due to the complexity brought by Lambert $W$ function. We treat the problem as an optimization problem where $S(\theta_x, \theta_y)$ is the objective function.
> We use an analytical and variation version of coordinate ascent to solve the optimization problem. In the beginning, we start from a well-chosen point (see Equation G.113) arbitrarily close to point $(\theta_x=0, \theta_y=0)$. In each iteration, we fix one of $\theta_x$ and $\theta_y$ and make another vary. The goal is to maximize the objective function $S(\theta_x, \theta_y)$. In this way, we construct an infinite sequence of allocations and reach the supremum finally.
>
> The proof mainly consists of the following four aspects, which are much harder than a simple coordinate ascent algorithm.
>
> **Aspect 1**: We start optimization from a well-chosen point close to $(\theta_x=0, \theta_y=0)$. In each iteration, once we fix one of $\theta_x$ and $\theta_y$ and make another one vary, we prove there exists one and only one supremum. For example, once we fix $\theta_x=\theta_{x,0}$ where $\theta_{x,0}>0$ can be artitrarily small, we prove there exists one and only one $-\frac{\varepsilon_{y,1}}{\varepsilon_{y,2}}< \theta_{y,1}<1$ that maximizes $S(\theta_{x,0},\theta_y)$ (see Proposition G.2). In the next step, we fix $\theta_y=\theta_{y,1}$ and find $\theta_{x,2}$ to lift $S(\theta_x, \theta_{y,1})$ further. In this way, we find an infinite sequence of allocations $(\theta_{x,0}, \theta_{y,0}), (\theta_{x,0}, \theta_{y,1}), (\theta_{x,2}, \theta_{y,1}), (\theta_{x,2}, \theta_{y,3}),\dots$.
> In the end, we will show that in these iterations, the allocations corresponding to these local maximums are more and more *extreme*.
>
> **Aspect 2**: However, the condition describing the local maximum (e.g., $\theta_{y,1}$) is complicated. For example, Equation G.119 describes the condition that $\theta_{y,1}$ should satisfy. We cannot solve the Equation analytically. So we turn to analyze the condition to study the property of these local maximums.
> We use a crucial transformation (in Equations G.120~G.124) to obtain key Equations  G.124, G.138, G.142 which express the property of these local maximums (corresponding to allocations) implicitly.
> Equations  G.124, G.138, G.142 also characterize the relation between local maximums obtained in neighboring iterations.
>
> **Aspect 3**: Prove that the constructed sequential allocations in the above iterations are more and more *extreme*. There are two problems we need to tackle. First, how to measure the *extremeness* of one allocation. We find formulae (see Equations G.148 and G.149) to measure one allocation's extremeness indirectly. Second, how to compare the extremeness of two allocations. The Equations G.124, G.138, G.142 used to characterize the property of local maximums obtained in Aspect 2 are also used to prove these sequential allocations are more and more extreme.
>
> **Aspect 4**: Prove the limit of the allocation sequence is $(\theta_x=1,\theta_y=1)$ which can maximize the objective function.
>
> Hope you enjoy our analysis.
>
> **Q4**:  The section called Lemmas and Notations has no lemmas. The Applications section should be called Discussion.
>
> **A4**: The section ‘Lemmas and Notations’ introduces notations and Lemmas B.1. We will add subcaption to improve consistency. We will add discussion and revise the Application section name.
>
> Thanks.

---

> > ### Comment · Reviewer_ZfdJ · 2023-08-11
> > **Answer to rebuttal**
> >
> > I have read the authors response and consider my concerns addressed. The paper grading was updated accordingly.

---

### Official Review · Reviewer_odZe · 2023-07-06

**Soundness:** 3 good
**Presentation:** 3 good
**Contribution:** 3 good
**Rating:** 7
**Confidence:** 4

**Summary:**

This paper investigates the properties of KL divergence between Gaussian distributions. The main theoretical contributions include two main theorems. The first one gives the supremum of reverse KL divergence between Gaussians when the forward KL divergence is bounded. The conditions when the supremum is attained are also identified. The second theorem gives the relaxed triangle inequality of KL divergence between Gaussians. Based on these two main theorems, this paper also derives several corollaries, including the local approximations and a lower bound of reverse KL divergence. It is also notable that the bounds are dimension-free. Finally, this paper discusses several applications of the theoretical results in OOD detection with flow-based generative models and safe/robust reinforcement learning.

Overall, the research questions studied in this paper have not been answered before. The theoretical contributions of this paper are novel and solid. The proofs are carefully written and correct. Notably, the proof of Theorem 4 is rather technical. The theorems presented in this paper can be applied in various contexts involving KL divergence and Gaussian distributions.


**Strengths:**

1.	The problems studied in this paper are novel and interesting. This paper answers these research problems for the first time.
2.	The proofs, which are based on the Lambert W function, are technical.
3.	The theoretical results can be applied to various problems, including anomaly detection and reinforcement learning. These results also have other potential applications.


**Weaknesses:**

1.	It is possible to make some equations tighter by introducing notations earlier. For example, notations in Equations (G.146)-(G.150) can be introduced earlier to make Equations (G.128)-(G.144) more concise.
2.	The derivations in Equation (E.54) and (J.193) are over-detailed. These two equations can be shortened.


**Questions:**

See Weaknesses.

**Limitations:**

The authors have discussed social impacts and limitations.

---

> ### Author Rebuttal · Authors · 2023-08-03
>
> Thanks for your careful and valuable comments. We explain your concerns as follows.
>
> C1: It is possible to make some equations tighter by introducing notations earlier. For example, notations in Equations (G.146)-(G.150) can be introduced earlier to make Equations (G.128)-(G.144) more concise.
>
> A1: We will introduce these notations earlier to make Equations (G.128)-(G.144) more concise in the revision.
> We have two choices on where to introduce these notations (Equations (G.146)-(G.150)). The first choice is to introduce them earlier. This would make the proof more concise but harder to understand. The second choice is to retain the details of Equations (G.128)-(G.144) and introduce notations (i.e., Equations (G.146)-(G.150)) later. This would make the proof more detailed but a little longer. Note that these two choices do not affect the correctness of the proof.
>
> C2: The derivations in Equation (E.54) and (J.193) are over-detailed. These two equations can be shortened.
>
> A2: Thanks for this suggestion. In the revision, we will make the derivations in Equation (E.54) and (J.193) more concise.
>
> Thanks again for your valuable comments.

---

> > ### Comment · Reviewer_odZe · 2023-08-21
> >
> > I have read the response and appreciate the contributions of this paper. The theoretical contributions of this paper are novel and solid. The proofs are carefully written and correct. I keep my accept recommendation.

---

### Official Review · Reviewer_rMwu · 2023-07-06

**Soundness:** 3 good
**Presentation:** 3 good
**Contribution:** 3 good
**Rating:** 7
**Confidence:** 3

**Summary:**

In this paper, the authors look at the KL divergence between two multivarite Gaussian distributions. The KL divergence is an important distance function between two distributions. However, it lacks certain nice properties that other metric distance functions such as variation distance satisfies: namely, symmetry and triangle inequality. This paper shows that, nevertheless, KL divergence satisfies an approximate version of these two important properties. Specifically, if one of the KL divergences is small then the reverse KL divergence will also be small. Similarly, if two pairs of distributions have small KL divergences between them, then the remaining pair will also have a small KL divergence in between.

The results are derived by posing this as an optimization problem that minimizes the unknown KL divergences subject to the constraint that the known KL divergences are small. Then certain relevant functions such as  and  are analyzed to derive an upper bound for the above optimization problems.

Finally, the authors argue that such approximate symmetry and approximate triangle inequality appear in several important practical applications. In fact, they mention one such problem involving deep neural networks that led them to study this question.

One more application of this result is that learning a multivariate Gaussian in either KL gives a similar learning result for the reverse KL. So far algorithms have been derived separately for the two directions, see [arXiv:1710.05209] and [arXiv:2107.10450] for more.

I have not carefully checked the mathematical details.

**Strengths:**

The paper works on a fundamental mathematical problem of proving that KL divergence between multivariate Gaussians is almost a metric near 0. and gives a nice solution. This paper is very nicely written and a pleasure to read. This is really a beautiful paper.



**Weaknesses:**

None.

**Questions:**

None.

**Limitations:**

None.

---

> ### Author Rebuttal · Authors · 2023-08-03
>
> Thanks for your careful and valuable comments, especially for suggesting one more application of our theorems. Please see the *common rebuttal for all reviewers* for discussion about this application.
>
> There is no weakness, question, or limitation contained in the comments. Thanks again for your valuable comments!

---

### Official Review · Reviewer_azx2 · 2023-07-07

**Soundness:** 4 excellent
**Presentation:** 4 excellent
**Contribution:** 4 excellent
**Rating:** 7
**Confidence:** 4

**Summary:**

In this paper, the authors prove the following interesting mathematical properties of the Kullback-Leibler (KL) divergence between multivariate Gaussian distributions, while the KL divergence is not a proper distance (in sense that it is not symmetric) and does not satisfy the triangle inequality, but:
1. if $KL(N_2||N_1) \leq \epsilon$ then it can be shown that the supremum of $KL(N_1||N_2) $  can be upper bounded by some explicit function of $\epsilon$ that is of order $\epsilon$ for $\epsilon$ small, so that the KL is approximately symmetric in the Gaussian case when being close; and
2. an infimum of $KL(N_1||N_2) $ is also derived for $KL(N_2||N_1) \geq M$; and
2. for three Gaussian $N_1, N_2, N_3$, one has an upper bound $KL(N_1||N_3) $ that verifies the triangle inequality up to a factor of three, again when the three Gaussians are close.

The authors discuss the basic proof ideas and some possibly applications in Section 5.


**Strengths:**

This paper focuses on the fundamental theoretical properties of Kullback-Leibler divergence between multivariate Gaussian distributions, that, to the best of my knowledge, are novel, and have wide applications in ML.
I've not checked the detailed proofs, but the proof sketch looks compelling. The proof idea is very interesting and may be of independent interest.




**Weaknesses:**

The paper is in good shape, I do not have specific concern to raise.

**Questions:**

1. The authors mention that they propose an unified OOD detection algorithm KLODS, but no detail about KLODS is given, it would be great if the authors could elaborate more on this.

**Limitations:**

This paper is primarily of theoretical nature, and I do not see any potential negative societal impact of this work.

---

> ### Author Rebuttal · Authors · 2023-08-03
>
> Thanks for your careful and valuable comments. We explain your concerns as follows.
>
>
> C1: The authors mention that they propose an unified OOD detection algorithm KLODS, but no detail about KLODS is given, it would be great if the authors could elaborate more on this.
>
> A1: We will add more details about the OOD detection algorithm in the revision.
>
> The OOD detection work motivating the theoretical research in this submission is elaborated in our another manuscript named “Kullback-Leibler Divergence-Based Out-of-Distribution Detection  with Flow-Based Generative Models”. Currently, this manuscript is under review in another journal. We also append the anonymous version in the supplementary material. Please see the appended manuscript for details.
>
> Thanks again for your valuable comments.

---

> > ### Comment · Reviewer_azx2 · 2023-08-11
> >
> > Thanks for the comments, I have read the rebuttal.

---

### Author Rebuttal · Authors · 2023-08-05

Thanks for your valuable comments.

Here we address one concern from Reviewer (8dDm) on the applications of our theory. We think Reviewer (rMwu) may be also interested in this point. So we put the answer in the common rebuttal.

Reviewer (rMwu) proposes no weakness or question. We thank Reviewer rMwu for her/his suggestion of one more important application of Theorem 1 (approximate symmetry of KL divergence between Gaussians) on sample complexity research.
Reviewer (8dDm) proposes one problem as follows.
“Since the KL divergence has a wide range of applications, the *two* applications shown in this paper are kind of limited and not convincing. Can the theoretical results in this paper be used to some other tasks in machine learning?”

**A**: As pointed out by reviewers, our theory can be applied to various problems ranging from deep learning to reinforcement learning to sample complexity research.

**Firstly**, we have discussed **four** applications in Section 5 and Appendix L. They are:

(1) *The Motivation Application on OOD Detection Using Flow-Based Model*. Both the approximate symmetry (Theorem 1) and relaxed triangle inequality (Theorem 5) are applied in the motivation application. Please see Section 5.1 and the manuscript (anonymous version) “Kullback-Leibler Divergence-Based Out-of-Distribution Detection with Flow-Based Generative Models” in the supplementary materials.

(2) *Providing Theoretical Guarantee for Continuous Gaussian policy in AWAC method [arXiv:2006.09359]*. Nair et al. provide a theoretical guarantee for discrete policy distributions. The approximate symmetry (Theorem 1) can extend their guarantee to continuous Gaussian policy. Please see Appendix L.1.

(3) *Bringing New Insights to Existing Reinforcement Learning Algorithm*. In [arXiv:1806.06920, ICLR 2018], Abdolmaleki et al. propose the MPO algorithm for reinforcement learning. They use Expectation-Maximization (EM) to solve control problems and use constraints on KL terms in both E and M-steps. Theorem 1 can eliminate such a difference for continuous Gaussian policies. Please see Appendix L.1.

(4) *Extending One-step Safety Guarantee to Multiple Steps in Reinforcement Learning*. In [arXiv:2201.11927, ICML 2022], Liu et al. propose an Expectation-Maximization style approach for learning safe policy in reinforcement learning. Our relaxed triangle inequality extends their one-step robustness guarantee to multiple steps. Please see Appendix L.2 and [arXiv:2201.11927, ICML 2022] for details.

**Secondly**, as Reviewer (rMwu) pointed out, Theorem 1 (approximate symmetry of KL divergence) has one more important application. Theorem 1 can be used to **extend existing theoretical results in sample complexity research**. In “Ashtiani, et al., Near-optimal Sample Complexity Bounds for Robust Learning of Gaussian Mixtures via Compression Schemes, Journal of the ACM, 2020. arXiv:1710.05209”, the authors propose a compression-based learning method and establish an optimal lower bound of sample complexity for learning Gaussian mixtures. For a fixed target Gaussian mixture distribution $P$, the learning method receives a sample set and outputs a distribution $Q$ satisfying $KL(Q||P)\leq \epsilon$. See page 26, the inequality below Equation (17) in [arXiv:1710.05209, JACM 2020], where KL divergence is used to bound Total Variation distance. Their conclusion applies to a single Gaussian when the number of mixture components is 1. One open problem proposed in their paper is: what is the sample complexity for learning Gaussian mixtures with guarantee using the reverse KL divergence $KL(P||Q)\leq \epsilon$ (see page 35, last paragraph, in [arXiv:1710.05209, JACM 2020]). Our Theorem 1 on the approximate symmetry can extend existing theory and answer this open problem in the single Gaussian case. According to Theorem 1, when $KL(Q||P)\leq \epsilon$, $KL(P||Q) \leq \epsilon+2\epsilon^{1.5}+O(\epsilon^2)$. The supremum equals $O(\epsilon)$ when $\epsilon<1$. This implies that the bounds of forward and reverse KL divergence have the same order. Therefore, the optimal sample complexity for learning single Gaussian is the same when using reverse KL divergence as a guarantee.

Similarly, in “Bhattacharyya et al., Learning Sparse Fixed-Structure Gaussian Bayesian Networks, AISTATS 2022,
arXiv:2107.10450”, the authors propose a learning method for sparse fixed-structure Gaussian Bayesian network, which can be treated as a representation of multidimensional Gaussian distributions (see Koller's book: Probabilistic graphical models: principles and techniques). They prove the sample complexity of their method for learning Gaussian distribution with guarantee $KL(P||Q)\leq \epsilon$, where $P$ is the target Gaussian distribution from which samples are drawn, $Q$ is the learned Gaussian distribution. Specially, in page 9 of [arXiv:2107.10450, AISTATS 2022], the authors note that their theoretical result uses reverse KL divergence while the JACM paper discussed above uses forward KL divergence. Again, Theorem 1 can extend their conclusion to forward KL divergence and eliminate the difference between forward and reverse KL divergence.

Just as Reviewer (rMwu) says, "Learning a multivariate Gaussian in either KL gives a similar learning result for the reverse KL. So far algorithms have been derived separately for the two directions". Theorem 1 can eliminate the difference between forward and reverse KL divergence in this scenario. In other words, we answer the open problem proposed in [arXiv:1710.05209, JACM 2020] in single Gaussian case. We can see that the asymmetry of KL divergence between Gaussians has hindered researchers for a long time. Note that Theorem 1 allows us to exchange forward and reverse KL divergence in the derivation on-demand, which would bring more convenience. We plan to explore this direction in the future.

Since KL divergence and Gaussian are widely applied, our theory may have more potential applications.

Thanks.

---

### Decision · Program_Chairs · 2023-09-21

**Decision:**

Accept (poster)

**Comment:**

The paper studies the properties of the KL divergence between multivariate Gaussian distributions and show various properties such as approximate symmetry in KL divergence, a relaxed triangle inequality. The paper is well written and all reviewers recommend acceptance. I  encourage authors incorporate reviewer comments. I also suggest that they add a few one dimensional toy examples in the paper.